# TOPOLOGICAL SCHRÖDINGER BRIDGE MATCHING

**Maosheng Yang**
Delft University of Technology
`m.yang-2@tudelft.nl`

## ABSTRACT

Given two boundary distributions, the *Schrödinger Bridge* (SB) problem seeks the "most likely" random evolution between them with respect to a reference process. It has revealed rich connections to recent machine learning methods for generative modeling and distribution matching. While these methods perform well in Euclidean domains, they are not directly applicable to topological domains such as graphs and simplicial complexes, which are crucial for data defined over network entities, such as node signals and edge flows. In this work, we propose the *Topological Schrödinger Bridge problem* ($\mathcal{T}$SBP) for matching signal distributions on a topological domain, where we set the reference process to follow some linear tractable *topology-aware* stochastic dynamics such as topological heat diffusion. For the case of Gaussian boundary distributions, we derive a *closed-form* Gaussian topological SB in terms of its time-marginal and stochastic differential. In the general case, leveraging the well-known result, we show that the optimal process follows the forward-backward topological dynamics governed by some unknowns. Building on these results, we develop $\mathcal{T}$SB-based models for matching topological signals by parameterizing the unknowns in the optimal process as *(topological) neural networks* and learning them through *likelihood training*. We validate the theoretical results and demonstrate the practical applications of $\mathcal{T}$SB-based models on both synthetic and real-world networks, emphasizing the role of topology. Additionally, we discuss the connections of $\mathcal{T}$SB-based models to other emerging models, and outline future directions for topological signal matching.

## 1 INTRODUCTION

As a fundamental problem in statistics and optimization, *matching distributions* aims to find a map that transforms one distribution to another. It has found numerous applications in machine learning tasks, particularly in generative modeling, which often involves learning a transformation from a data distribution to a simple one (often Gaussian) for efficient sampling and inference. While various methods have been proposed, including score-based [Ho et al., 2020; Song et al., 2020b] and flow-based [Lipman et al., 2022] generative models, among others [Neklyudov et al., 2023; Albergo et al., 2024; Tong et al., 2024a], the *Schrödinger Bridge* (SB)-based methods [De Bortoli et al., 2021; Chen et al., 2022a; Liu et al., 2024] provide a principled framework for matching *arbitrary* distributions.

Inspired by Schrödinger [1931; 1932], the classical SB problem (SBP) aims to find an optimal *stochastic process* that evolves from an initial distribution to a final distribution, while minimizing the *relative entropy* (Kullback-Leibler divergence) between the measures of the optimal process and the Wiener process [Léonard, 2014]. Alternatively, the SBP can be cast as a *stochastic optimal control* (SOC) problem which minimizes the kinetic energy while matching the distributions through a nonlinear stochastic process [Dai Pra, 1991; Pavon & Wakolbinger, 1991]. The optimal solution to this problem satisfies a *Schrödinger system* of coupled *forward-backward* (FB) stochastic differential equations (SDEs) [Léonard, 2014]. Traditionally, SB problems have been solved by addressing the unknowns in this system using purely numerical methods [Fortet, 1940; Föllmer, 1988]. More recently, machine learning approaches have been proposed [Pavon et al., 2021; Vargas et al., 2021; Wang et al., 2021; De Bortoli et al., 2021; Chen et al., 2022a] where the unknowns are approximated by learnable models (e.g., Gaussian processes, neural networks) trained on data-driven objectives, such as the *likelihood* training [Chen et al., 2022a].

However, SB-based methods have primarily focused on solving tasks in Euclidean spaces, such as time series, images [Deng et al., 2024] and point clouds. Modern learning tasks often involve data supported on irregular topological domains such as graphs, simplicial and cell complexes. Arising from applications like chemical reaction networks, biological networks, power systems, social networks [Wang et al., 2022; Faskowitz et al., 2022; Bick et al., 2023], the emerging field of *topological machine learning* [Papamarkou et al., 2024] centers on signals supported on topological objects such as nodes and edges, which can represent sensor data or flow type data over network entities. A direct application of existing SB models to such topological data may fail due to their inability to account for the underlying topology. Thus, in this work, we investigate the SBP for topological signals, with a focus on node and edge signals over networks modeled as graphs and simplicial complexes. To match distributions of such topological signals, our contributions are threefold.

(i) We propose the *Topological Schrödinger Bridge problem* ($\mathcal{T}$SBP), which seeks an optimal *topological stochastic process* that minimizes the relative entropy with respect to a reference process, while respecting the initial and final distributions. To incorporate the domain knowledge, we define the reference process to follow *topology-aware* SDEs ($\mathcal{T}$SDEs) with a linear *topological convolution* drift term, admitting tractable Gaussian transition kernels. This subsumes the commonly-used stochastic heat diffusions on graphs and simplicial complexes for networked dynamics modeling.

(ii) Focusing on the case where the end distributions are Gaussian, we find the closed-form optimal Gaussian $\mathcal{T}$SB and characterize it in terms of a *stochastic interpolant* time marginal, as well as its Itô differential. This generalizes the results of Bunne et al. [2023] where the reference process is limited to SDEs *scalar*-valued linear coefficients. For the general case, we show that, upon existing results, the optimal $\mathcal{T}$SB adheres a pair of FB-$\mathcal{T}$SDEs governed by some unknown terms (also called *policies*), which in turn satisfy a *system* driven by topological dynamics.

(iii) We propose the $\mathcal{T}$SB-based model for topological signal generative modeling and matching. Specifically, we parameterize the *hard-to-solve* policies by some (topology-aware) learnable models (e.g., graph/simplicial neural networks), and train them by maximizing the likelihood of the model based on Chen et al. [2022a]. We show that $\mathcal{T}$SB-based models unify the extensions of score-based and diffusion bridge-based models [Song et al., 2020b; Zhou et al., 2024] for topological signals.

We validate the theoretical results and demonstrate the practical implications of $\mathcal{T}$SB-models on synthetic and real-world networks involved with brain signals, single-cell data, ocean currents, seismic events and traffic flows. Before concluding the paper, we extensively discuss future directions in generative modeling for topological data. Overall, our work lies in the intersection of SB theory, stochastic dynamics on topology, machine learning and generative modeling for topological signals.

**Notations.** We denote by $X$ a *stochastic process* $(X_t)_{0 \leq t \leq 1}$ as a map $X : [0,1] \times \mathcal{X} \to \mathbb{R}^n$ from the unit time interval $[0,1]$ (i.e., *index space*) and sample space $\mathcal{X}$ (e.g., Euclidean space) to $\mathbb{R}^n$ (state space). Here, $X_t$ is a random variable representing the state at $t$. The standard $n$-dim *Wiener process* (Brownian motion) is denoted by $W$. Let $\Omega = \mathcal{C}([0,1], \mathbb{R}^n)$ denote the space of all continuous $\mathbb{R}^n$-valued paths on $[0,1]$, and let $\mathcal{P}(\Omega)$ denote the space of probability measures on $\Omega$. For a *path measure* $\mathbb{P} \in \mathcal{P}(\Omega)$ describing the law of the process $X$, we denote by $\mathbb{P}_t$ its *time marginal* that describes the distribution of $X_t$, i.e., if $X \sim \mathbb{P}$, then $X_t \sim \mathbb{P}_t$. We assume distributions of random variables are associated with measures that have a *density* with respect to the Lebesgue measure. We also assume locally Lipschitz smoothness on the drift and diffusion coefficients of SDEs.

## 2 BACKGROUND

### 2.1 SCHRÖDINGER BRIDGE PROBLEM

Let $\mathbb{Q}_W$ be the path measure of a Wiener process $\mathrm{d}Y_t = \sigma \, \mathrm{d}W_t$ with variance $\sigma^2$. The *classical* SBP [Léonard, 2014] seeks an optimal path measure $\mathbb{P}$ on $\Omega$ by minimizing its relative entropy $D_{\mathsf{KL}}$ with respect to $\mathbb{Q}_W$

$$\min D_{\mathsf{KL}}(\mathbb{P} \| \mathbb{Q}_W), \quad \text{s.t. } \mathbb{P} \in \mathcal{P}(\Omega), \mathbb{P}_0 = \rho_0, \mathbb{P}_1 = \rho_1, \quad \text{(SBP)}$$

where $\rho_0$ and $\rho_1$ are the *prescribed* initial and final time marginals on $\mathbb{R}^n$. Intuitively, the SBP aims to find a stochastic process evolving from $\rho_0$ to $\rho_1$ that are "most likely" to a *reference* (a *prior*) process, here the Wiener process. This is in fact a *dynamic* formulation of the *entropic-regularized*

*optimal transport* (OT) with a quadratic transport cost [Villani, 2009]. The *static* formulation reads

$$\min_{\pi \in \Pi(\rho_0, \rho_1)} \int_{\mathbb{R}^n \times \mathbb{R}^n} \frac{1}{2} \|x_0 - x_1\|^2 \, d\pi(x_0, x_1) + \sigma^2 D_{\mathsf{KL}}(\pi \| \rho_0 \otimes \rho_1) \qquad \text{(E-OT)}$$

where $\Pi(\rho_0, \rho_1)$ is the set of couplings between $\rho_0$ and $\rho_1$, and $\rho_0 \otimes \rho_1$ denotes their product measure. In this *static* formulation, the optimization is over the coupling of $\rho_0$ and $\rho_1$, as opposed to the full path measure in the *dynamic* formulation (SBP). As $\sigma \to 0$, E-OT reduces to the typical 2-Wasserstein OT, and the associated dynamic problem is given by Benamou & Brenier [2000].

## 2.2 TOPOLOGICAL SIGNALS

In this work, we are interested in signals defined on a graph, a simplicial complex or a cell complex such a topological domain, denoted by $\mathcal{T}$. If $\mathcal{T}$ is a graph with a node set and an edge set, we may define a *node signal* $x \in \mathbb{R}^n$ as a collection of values associated to the nodes, where $n$ denotes the number of nodes. Such signals often arise from sensor measurements in sensor networks, user properties in social networks, etc [Shuman et al., 2013]. Similarly, if $\mathcal{T}$ is a simplicial 2-complex $\mathrm{SC}_2$ with the sets of nodes, edges, as well as triangular faces (or triangles), we can define an *edge flow* by associating a real value to each *oriented* edge. Here, for an edge $e = \{i, j\}$, if we choose $[i, j]$ as its positive orientation, then $[j, i]$ represents the opposite [Godsil & Royle, 2001]. The sign of the signal thus indicates the flow orientation relative to the chosen one. Such edge signals often represent flows of information or energy, such as water flows, power flows, or transaction flows [Bick et al., 2023]. Moreover, we may consider signals on general topological objects such as higher-order simplices (or cells). If $n$ is the number of simplices, we refer to $x \in \mathbb{R}^n$ as a *topological signal* where the $i$-th entry represents the signal value on the $i$-th simplex. In topology, these are called *cochains*, which are the discrete analogues to *differential forms* [Lim, 2020].

The emerging field of learning on graphs and topology [Barbarossa & Sardellitti, 2020; Papamarkou et al., 2024] concerns such topological signals, where the central idea is to leverage the underlying topological structure in $\mathcal{T}$. For example, the *graph Laplacian* or adjacency matrix (or their variants) can be used to encode the graph's structure, acting as a spectral operator for node signals [Chung, 1997]. Similarly, in a $\mathrm{SC}_2$, the *Hodge Laplacian* can be defined as the operator for edge flows, composed of the *down* and *up* parts, which encode the edge adjacency through a common node or triangle, respectively. Other variants of Hodge Laplacians can also be defined [Grady & Polimeni, 2010; Schaub et al., 2020]. Thus, for a topological signal $x \in \mathbb{R}^n$, we assume a *Laplacian-type*, positive semidefinite, *topological operator* $L \in \mathbb{R}^{n \times n}$ on $\mathcal{T}$ which encodes the topological structure.

In a probabilistic setting, a *topological signal* can be considered random, following some high-dim distribution on $\mathbb{R}^n$ associated with the topology $\mathcal{T}$. This allows for the application of probabilistic methods to topological signals, similar to Euclidean cases. Recent works [Borovitskiy et al., 2021; Yang et al., 2024; Alain et al., 2024] have modeled node signals and edge flows using *Gaussian processes* (GPs) on graphs and simplicial complexes. These GPs encode the topological structure by building their covariance matrix (kernel) $\Sigma$ as a *matrix function* of the associated Laplacian $L$. For example, a diffusion node GP uses the kernel $\Sigma = \exp(-\frac{\kappa^2}{2} L)$ with a hyperparameter $\kappa$ and the graph Laplacian $L$. Other GPs can be defined as well to model signals in specific subspaces with certain properties [Yang et al., 2024] or to jointly model the node-edge signals [Alain et al., 2024].

## 3 TOPOLOGICAL SCHRÖDINGER BRIDGE PROBLEM

In a topological domain $\mathcal{T}$, we consider a *topological stochastic process* $X$ where the index space is instead the product space of $[0, 1]$ and the set of topological objects (e.g., nodes, edges) in $\mathcal{T}$, and the state space $\mathbb{R}^n$ is the space of topological signals with $n$ the cardinality of the set. When we consider the node set in a graph (the edge set in a $\mathrm{SC}_2$), $X$ is a stochastic process of node signals (edge flows). We assume that $X$ follows some *unknown* dynamics with its law described by the path measure $\mathbb{P}$. For some prescribed initial and final time-marginals, i.e., $X_0 \sim \mathbb{P}_0 = \nu_0$ and $X_1 \sim \mathbb{P}_1 = \nu_1$, we then aim to obtain the optimal $\mathbb{P}$ by solving the *Topological Schrödinger Bridge Problem* ($\mathcal{T}$SBP):

$$\min D_{\mathsf{KL}}(\mathbb{P} \| \mathbb{Q}_{\mathcal{T}}), \quad \text{s.t. } \mathbb{P} \in \mathcal{P}(\Omega), \mathbb{P}_0 = \nu_0, \mathbb{P}_1 = \nu_1. \qquad (\mathcal{T}\text{SBP})$$

Here, $\mathbb{Q}_{\mathcal{T}}$ is the path measure of a reference process $Y$ which follows some prior *topology-aware* stochastic dynamics on $\mathcal{T}$. Effectively, the solution $\mathbb{P}$ to $\mathcal{T}$SBP describes the "most likely" process

Figure 1: Heat diffusion starting from a node over a graph *(Left)* and an edge over a $\mathrm{SC}_2$ *(Right)*, followed by intermediate states, then reaching the steady states where the heat becomes uniform for the node case whereas circulating around the cycle for the edge case.

$X$ that conforms to the prior $Y$ in the sense of minimizing relative entropy with respect to $\mathbb{Q}_\mathcal{T}$, while respecting the initial and final distributions $\nu_0$ and $\nu_1$.

**Topological stochastic dynamics.** For the reference process $Y$, given an initial topological signal condition $Y_0 = y_0$, we assume it follows a general class of topological SDEs:

$$\mathrm{d}Y_t = f(t, Y_t; L)\,\mathrm{d}t + g_t\,\mathrm{d}W_t, \tag{$\mathcal{T}$SDE}$$

where $f_t \equiv f(t, \cdot\,; L) : \mathbb{R}^n \to \mathbb{R}^n$ is a time-varying *drift* that depends on the topological structure $\mathcal{T}$ through the operator $L$, and $g_t \equiv g(t) \in \mathbb{R}$ is a scalar *diffusion* coefficient. For tractability, we consider a class of *linear dynamics* on $\mathcal{T}$ with the following drift term:

$$f(t, Y_t; L) = H_t(L)Y_t + \alpha_t, \quad \text{with } H_t(L) = \sum_{k=0}^K h_k(t)L^k \tag{1}$$

and $\alpha_t \in \mathbb{R}^n$ a bias term. Here, $H_t(L)$, denoted simply as $H_t$, is a *matrix polynomial* of $L$ with time-varying coefficients $h_{k,t} \equiv h_k(t)$, which is able to approximate any *analytic function* of $L$ for an appropriate $K$ by the *Cayley-Hamilton theorem*. The drift $f_t$ is also referred to as a *topological convolution* of the topological signal in the literature. With a graph Laplacian $L$, this returns the *graph convolution* of a node signal [Sandryhaila & Moura, 2013; 2014], and with the Hodge Laplacian for edges or general simplices, it yields the *simplicial convolution* [Yang et al., 2022b]. Various topological machine learning methods have been developed based on such convolutions for their expressivity and efficiency. We provide a few examples of linear $\mathcal{T}$SDE, which will be used later.

*Topological stochastic heat diffusion*: $\mathcal{T}$SDE gives the stochastic variant of heat equation on $\mathcal{T}$

$$\mathrm{d}Y_t = -cLY_t\,\mathrm{d}t + g_t\,\mathrm{d}W_t, \tag{$\mathcal{T}$SHeat}$$

by setting $H_t = -cL$, with $c > 0$. When $\mathcal{T}$ is a graph with the graph Laplacian $L$, this dynamics enables modeling graph-time GPs [Nikitin et al., 2022], networked dynamic system [Pereira et al., 2010; Delvenne et al., 2015; Santos et al., 2024], and social opinion dynamics [Gaitonde et al., 2021]. More importantly, in the deterministic case of $g_t = 0$, it has a harmonic steady-state, revealing the *topological features* of $\mathcal{T}$. Specifically, node diffusion converges to a state that can identify the connected components (0-dim *holes*), while edge diffusion in a $\mathrm{SC}_2$ based on the Hodge Laplacian has a converging state of circulating around cycles (1-dim *holes*). We refer to Fig. 1 for such illustrations. In line with our goal of distribution matching for topological signals, we present three examples of $\mathcal{T}$SHeat, inspired by diffusion models for generative modeling [Song et al., 2020b].

*Example* 1 ($\mathcal{T}$SHeat$_{\mathrm{BM}}$). Consider a constant $g_t = g$ in $\mathcal{T}$SHeat. This results in a mixture of a topological heat diffusion and BM with variance $g^2$, which we refer to as $\mathcal{T}$SHeat$_{\mathrm{BM}}$.

*Example* 2 ($\mathcal{T}$SHeat$_{\mathrm{VE}}$). For some noise scales $0 < \sigma_{\min} < \sigma_{\max}$, consider a time-increasing $g_t = \sqrt{\mathrm{d}\sigma^2(t)/\mathrm{d}t}$ with $\sigma(t) = \sigma_{\min}(\sigma_{\max}/\sigma_{\min})^t$, which drives the well-known *variance exploding* (VE) noising process [Song & Ermon, 2020; Song et al., 2020b]. The resulting form of $\mathcal{T}$SHeat is

$$\mathrm{d}Y_t = -cLY_t\,\mathrm{d}t + \sqrt{\mathrm{d}\sigma^2(t)/\mathrm{d}t}\,\mathrm{d}W_t. \tag{$\mathcal{T}$SHeat$_{\mathrm{VE}}$}$$

*Example* 3 ($\mathcal{T}$SHeat$_{\mathrm{VP}}$). When combined with another noising process, known as the *variance preserving* (VP) process [Sohl-Dickstein et al., 2015; Ho et al., 2020; Song et al., 2020b], we obtain

$$\mathrm{d}Y_t = -\left(\frac{1}{2}\beta(t)I + cL\right)Y_t\,\mathrm{d}t + \sqrt{\beta(t)}\,\mathrm{d}W_t, \tag{$\mathcal{T}$SHeat$_{\mathrm{VP}}$}$$

where $\beta(t) = \beta_{\min} + t(\beta_{\max} - \beta_{\min})$ with scales $0 < \beta_{\min} < \beta_{\max}$. The drift here can be considered as an instantiation of the topological convolution $H_t = -\left(\frac{1}{2}\beta(t)I + cL\right)$ in (1).

**Gaussian transition kernels.** The $\mathcal{T}$SDE, as an Itô process, is fully characterized by its *transition kernel* in a probabilistic sense. As a result of the linear drift (1) of $\mathcal{T}$SDE, the associated transition kernel $p_{t|s}(y_t|y_s)$ (i.e., conditional distribution of $Y_t|Y_s$) is Gaussian. Its mean and covariance can be computed according to Särkkä & Solin [2019, Eq. 6.7]. Let the *transition matrix* of the ODE $\mathrm{d}Y_t = H_t(L)Y_t\,\mathrm{d}t$ be denoted by $\Psi_{ts} \equiv \Psi(t, s)$, which is given by $\Psi_{ts} = \exp\left(\int_s^t H_\tau\,\mathrm{d}\tau\right)$ [cf. Lemma B.5]. For brevity, we denote $\Psi_{t0}$ as simply $\Psi_t$. We then have the following lemma.

**Lemma 4** (Statistics of transition kernels). *For the $\mathcal{T}$SDE with the linear drift ([1](#)), its Gaussian transition kernel $p_{t|0}(y_t|y_0)$ has the mean $m_t$ and the cross covariance $K_{t_1 t_2}$, at $t_1$ and $t_2$:*

$$m_t = \Psi_t y_0 + \Psi_t \int_0^t \Psi_\tau^{-1} \alpha_\tau \, \mathrm{d}\tau =: \Psi_t y_0 + \xi_t, \qquad \text{(cond. mean)}$$

$$K_{t_1 t_2} = \Psi_{t_1} \left( \int_0^{\min\{t_1, t_2\}} g_\tau^2 \Psi_\tau^{-2} \, \mathrm{d}\tau \right) \Psi_{t_2}^\top. \qquad \text{(cond. cross cov)}$$

More importantly, we may characterize them for $\mathcal{T}$SHeat$_{\text{BM}}$ and $\mathcal{T}$SHeat$_{\text{VE}}$ in closed-forms. Both have the same mean $m_t = \Psi_t y_0$ with $\Psi_t = \exp(-cLt)$, and their covariances are given by:

$$K_{t_1 t_2} = \begin{cases} \frac{g^2}{2c} \big[ \exp(-cL|t_1 - t_2|) - \exp(-cL(t_1 + t_2)) \big] L^{-1}, & \text{for } \mathcal{T}\text{SHeat}_{\text{BM}} \\ \sigma_{\min}^2 \ln\big(\frac{\sigma_{\max}}{\sigma_{\min}}\big) \exp(-cL(t_1 + t_2)) \big[ \exp(2A \min\{t_1, t_2\}) - I \big] A^{-1}, & \text{for } \mathcal{T}\text{SHeat}_{\text{VE}} \end{cases} \quad (2)$$

with $A = \ln\big(\frac{\sigma_{\max}}{\sigma_{\min}}\big) I + cL$. If $L$ is singular, we use a perturbed $L + \epsilon I$ for a small $\epsilon > 0$. We detail the derivations in [Appendix B](#). These expressions allow for tractable solutions for $\mathcal{T}$SBP and, more importantly, facilitate the construction of $\mathcal{T}$SB-based learning models, which we will discuss later.

## 3.1 Towards an Optimal Solution of $\mathcal{T}$SBP

To solve the classical SBP, early mathematical treatments [Fortet, 1940; Beurling, 1960; Jamison, 1975; Föllmer, 1988] lead to a *Schrödinger system* characterizing the SB optimality. Similarly, by Disintegration of Measures, we can convert the $\mathcal{T}$SBP to a static problem over the joint measure $\mathbb{P}_{01}$ of the initial and final states, instead of the full path measure $\mathbb{P}$

$$\min D_{\text{KL}}(\mathbb{P}_{01} \,\|\, \mathbb{Q}_{\mathcal{T}01}), \quad \text{s.t. } \mathbb{P}_{01} \in \mathcal{P}(\mathbb{R}^n \times \mathbb{R}^n), \mathbb{P}_0 = \nu_0, \mathbb{P}_1 = \nu_1 \qquad (\mathcal{T}\text{SBP}_{\text{static}})$$

where $\mathbb{Q}_{\mathcal{T}01}$ is the joint measure of the reference process $Y$ at $t = 0$ and 1. The $\mathcal{T}$SBP$_{\text{static}}$ only concerns at the boundary times, unlike the (*dynamic*) $\mathcal{T}$SBP. Using Lagrange multipliers for the linear constraints above, we can arrive at a *Schrödinger System* that is instead *driven by topological dynamics* (see [Appendix C](#)), differing from the classical case [Jamison, 1975; Léonard, 2014]. This can be also interpreted through the equivalent E-OT formulation of $\mathcal{T}$SBP$_{\text{static}}$:

$$\min_{\mathbb{P}_{01}} \int_{\mathbb{R}^n \times \mathbb{R}^n} \frac{1}{2} \|y_1 - \Psi_1 y_0 - \xi_1\|_{K_{11}^{-1}}^2 \, \mathrm{d}\mathbb{P}_{01}(y_0, y_1) + \int_{\mathbb{R}^n \times \mathbb{R}^n} \log(\mathbb{P}_{01}) \, \mathrm{d}\mathbb{P}_{01} \qquad (\mathcal{T}\text{E-OT})$$

where the transport cost is linked to the $\mathcal{T}$SDE as a $K_{11}^{-1}$-weighted norm of the difference $y_1 - m_1$.

On the other hand, this system could *also* be derived from the SOC view which makes more apparent connections to machine learning methods. Building on the variational formulations of the classical SBP by Dai Pra [1991]; Pavon & Wakolbinger [1991], Caluya & Halder [2021] extended the analysis to the case with a general nonlinear reference process and derived the corresponding optimality condition. As it is convenient to arrive at an SOC formulation for the $\mathcal{T}$SBP (see [Appendix C](#)), we readily obtain the following optimality.

**Proposition 5** ($\mathcal{T}$SBP optimality; Caluya & Halder [2021]; Chen et al. [2022a]). *The optimal solution $\mathbb{P}$ of $\mathcal{T}$SBP can be expressed as the path measure of the following forward ([3a](#)), or equivalently, backward ([3b](#)), $\mathcal{T}$SDE:*

$$\mathrm{d}X_t = [f_t + g_t Z_t] \, \mathrm{d}t + g_t \, \mathrm{d}W_t, \quad X_0 \sim \nu_0, Z_t \equiv g_t \nabla \log \varphi_t(X_t) \qquad (3a)$$

$$\mathrm{d}X_t = [f_t - g_t \hat{Z}_t] \, \mathrm{d}t + g_t \, \mathrm{d}W_t, \quad X_1 \sim \nu_1, \hat{Z}_t \equiv g_t \nabla \log \hat{\varphi}_t(X_t) \qquad (3b)$$

*where ([3a](#)) runs forward and ([3b](#)) runs backward with a backward Wiener process. Here, $\varphi_t \equiv \varphi_t(X_t)$ and $\hat{\varphi}_t \equiv \hat{\varphi}_t(X_t)$ satisfy a pair of PDEs system (forward-backward Kolmogorov equations). Using nonlinear Feynman-Kac formula (or applying Itô's formula on $\log \varphi_t$ and $\log \hat{\varphi}_t$), this PDE system admits the SDEs*

$$\mathrm{d}\log \varphi_t = \frac{1}{2}\|Z_t\|^2 \, \mathrm{d}t + Z_t^\top \, \mathrm{d}W_t, \quad \mathrm{d}\log \hat{\varphi}_t = \left( \frac{1}{2}\|\hat{Z}_t\|^2 + \nabla \cdot (g_t \hat{Z}_t - f_t) + \hat{Z}_t^\top Z_t \right) \mathrm{d}t + \hat{Z}_t^\top \, \mathrm{d}W_t \tag{4}$$

*Then, the optimal path measure has the time-marginal $\mathbb{P}_t = \varphi_t(X_t)\hat{\varphi}_t(X_t) = \mathbb{P}_t^{(3a)} = \mathbb{P}_t^{(3b)}$.*

This optimality condition adapts the result from Chen et al. [2022a] for $\mathcal{T}$SBP. From the forward-backward $\mathcal{T}$SDEs (FB-$\mathcal{T}$SDEs in (3)), we see that the optimal $Z_t$ guides the forward $\mathcal{T}$SDE to the final $\nu_1$, and likewise $\hat{Z}_t$ adjusts the reverse $\mathcal{T}$SDE to return to the initial $\nu_0$. While solving the system (4) is still highly nontrivial, we **highlight** that the FB-$\mathcal{T}$SDEs (3) and (4) pave a way for constructing generative models and efficient training algorithms, as demonstrated by the recent works, to name a few, [Pavon et al., 2021; Vargas et al., 2021; De Bortoli et al., 2021; Chen et al., 2022a]. We further discuss in detail how to build such models for topological signals in Section 5.

## 4 GAUSSIAN TOPOLOGICAL SBP

In this section, we consider the special case of $\mathcal{T}$SBP where the initial and final measures are Gaussians, to which we refer as the *Gaussian topological SBP* (G$\mathcal{T}$SBP). We show that there exists a closed-form G$\mathcal{T}$SB by following the idea in Bunne et al. [2023], which focuses on a limited class of reference SDEs with a scalar coefficient in the drift, instead of a convolution operator $H_t(L)$. We establish the first closed-form expression on the G$\mathcal{T}$SB in the following theorem.

**Theorem 6.** *Denote by $\mathbb{P}$ the solution to G$\mathcal{T}$SBP with $\nu_0 = \mathcal{N}(\mu_0, \Sigma_0)$ and $\nu_1 = \mathcal{N}(\mu_1, \Sigma_1)$. Then, $\mathbb{P}$ is the path measure of a* Markov Gaussian process *whose marginal $X_t \sim \mathcal{N}(\mu_t, \Sigma_t)$ admits an expression in terms of the initial and final variables, $X_0, X_1$, as follows*

$$X_t = \bar{R}_t X_0 + R_t X_1 + \xi_t - R_t \xi_1 + \Gamma_t Z \tag{5}$$

*where $Z \sim \mathcal{N}(0, I)$ is standard Gaussian, independent of $(X_0, X_1)$, and*

$$R_t = K_{t1} K_{11}^{-1}, \quad \bar{R}_t = \Psi_t - R_t \Psi_1, \quad \Gamma_t := \mathrm{Cov}[Y_t | (Y_0, Y_1)] = K_{tt} - K_{t1} K_{11}^{-1} K_{1t}. \tag{6}$$

*Proof.* We provide a sketch of the proof here, with the full derivations presented in Appendix D.

1. By Disintegration of Measures, we first solve the reduced *static* Gaussian $\mathcal{T}$SBP$_{\text{static}}$ (i.e., $\mathcal{T}$E-OT). We can then convert the problem into a classical Gaussian E-OT via a change-of-variables. The closed-form formula for the latter has been recently found by Janati et al. [2020, Theorem 1]. Via an inverse transform, we can then obtain the optimal coupling $\mathbb{P}_{01}$ [i.e., the optimal G$\mathcal{T}$SB$_{\text{static}}$].

2. In the disintegration of G$\mathcal{T}$SBP to its static problem, the optimum is achieved when $\mathbb{P}$ shares the *bridge* with the reference $\mathbb{Q}_\mathcal{T}$ (i.e., $\mathbb{P}$ is in the *reciprocal class* of $\mathbb{Q}_\mathcal{T}$) [Föllmer, 1988; Léonard, 2014, Proposition 1]. The $\mathbb{Q}_\mathcal{T}$-*bridge*, $\mathbb{Q}_\mathcal{T}^{xy} = \mathbb{Q}_\mathcal{T}[\cdot | Y_0 = x, Y_1 = y]$, can be constructed using the conditional Gaussian formula and Lemma 4. Upon this, together with the optimal $\mathbb{P}_{01}$, we can construct the optimal $X_t$ and the marginal $\mathbb{P}_t$. $\quad\square$

At the first sight, the construction of optimal process $X$ in (5) meets the recently proposed *stochastic interpolant* framework by Albergo et al. [2024, Definition 1], in that $X_{t=0} = X_0$ and $X_{t=1} = X_1$, and $\Gamma_0 = \Gamma_1 = 0$. Moreover, from (5), we can compute the marginal statistics $\mathbb{P}_t$ in terms of its mean $\mu_t$ and covariance $\Sigma_t$ in closed-form as well, detailed in Corollary D.1. In the following, we characterize the process $X$ under the optimal $\mathbb{P}$ in terms of its Itô differential.

**Theorem 7** (SDE representation). *Under the optimal $\mathbb{P}$, the process $X$ admits the SDE dynamics:*

$$\mathrm{d}X_t = f_\mathcal{T}(t, X_t; L)\,\mathrm{d}t + g_t\,\mathrm{d}W_t, \ \text{ where } f_\mathcal{T}(t, x; L) = S_t^\top \Sigma_t^{-1}(x - \mu_t) + \dot{\mu}_t \tag{7}$$

*with $\mu_t, \Sigma_t$ the mean and covariance of $X_t$ [cf. Corollary D.1] and we have*

$$S_t = P_t - Q_t^\top + H_t K_{tt} - K_{t1} K_{11}^{-1} \Upsilon_t^\top, \tag{8}$$

*with $P_t = (R_t \Sigma_1 + \bar{R}_t C)\dot{R}_t^\top, Q_t = -\dot{\bar{R}}_t(CR_t^\top + \Sigma_0 \bar{R}_t^\top), \Upsilon_t = H_t K_{t1} + g_t^2 \Psi_1^{-1} \Psi_1^\top$, where $C$ is the covariance of $X_0, X_1$ in the optimal $\mathbb{P}_{01}$.*

*Proof.* We detail the proof in Appendix D and outline a sketch here. From Léonard [2014]; Caluya & Halder [2021] [cf. Theorem C.2], the optimal $\mathbb{P}$ is the law of an SDE in the class of (7). To determine the drift, we first compute the associated *infinitesimal generator* by definition for some test function. Since the generator for an Itô SDE is *known* (dependent on the drift) [Särkkä & Solin, 2019, Eq. 5.9], we can then match the two expressions and find a closed-form for the drift term. $\quad\square$

Theorems 6 and 7 characterize the optimal $\mathbb{P}$ of the G$\mathcal{T}$SBP from different views. While the stochastic interpolant formula is intuitive and straightforward, it is natural to look for the associated SDE for a Markov measure. Despite the packed variables, both results [cf. Eqs. (5) and (7)] fundamentally depend on the transition matrix $\Psi_t$ and $\xi_t, K_{t_1 t_2}$ [cf. Lemma 4], which has closed-forms (2) for $\mathcal{T}$SHeat$_{\text{BM}}$ and $\mathcal{T}$SHeat$_{\text{VE}}$. From a broader perspective, Theorems 6 and 7 extended the existing results of Bunne et al. [2023], where the reference process has a limited drift $cY_t + \alpha_t$ for some *scalar c*. While Chen et al. [2016] aimed to solve for a linear drift with a matrix coefficient, their results lead to the solution of a matrix Riccati equation, which is computationally expensive.

**Solution complexity.** While the variables involved in Eqs. (5) and (7) involve many matrix operations, we remark that (i) the *underlying* $\Psi_t$ is a matrix function of $L$ and can be computed efficiently [Higham, 2008]. Given the eigen-decomposition $L = U\Lambda U^\top$, denote by $\tilde{h}_k^{t,s} = \int_s^t h_{k,\tau}\, \mathrm{d}\tau$ the integral of the scalar coefficients in $H_t$ [cf. (1)], then we have $\Psi_{ts} = U \exp\left(\sum_{k=0}^K \tilde{h}_k^{t,s}\Lambda^k\right)U^\top$, where the matrix exponential can be directly computed elementwise on each diagonal element of $\Lambda$. (ii) The other terms depending on $\Psi_t$ can be computed similarly in the eigenspectrum of $L$.

## 5 FROM $\mathcal{T}$SBP TO TOPOLOGICAL SIGNAL GENERATIVE MODELS

The recent SB-based generative modeling framework primarily relies on the *learnable* parameterizations of the $(Z_t, \hat{Z}_t)$ pair (also viewed as the FB policies) in the FB-SDEs and a trainable objective that approximates the SBP. Specifically, Vargas et al. [2021] and De Bortoli et al. [2021] use GPs and neural networks, respectively, to parameterize the policies, and alternatively train them using iterative proportional fitting (IPF) to solve the *half*-bridge problem. On the other hand, Chen et al. [2022a] derived a *likelihood* based on the SB optimality condition, generalizing the *score matching* framework [Song et al., 2020b]. Upon the proposed $\mathcal{T}$SBP, along with the above theoretical results, we now discuss how to build generative models for topological signals using the existing framework designed for Euclidean domains.

$\mathcal{T}$**SB-based model.** Consider the matching task: *In some topological domain $\mathcal{T}$, given two sets of signal samples following initial and final distributions $\nu_0, \nu_1$ on $\mathcal{T}$, we aim to learn a Topological Schrödinger Bridge between the two distributions.* From Proposition 5, the optimal $\mathcal{T}$SB follows the FB-$\mathcal{T}$SDEs in (3). Moreover, given a path sampled from the forward SDE (3a) with an initial signal $x_0$, one can obtain an unbiased estimation of the *log-likelihood* $\mathcal{L}(x_0)$ of the $\mathcal{T}$SB model driven by the optimal policies by using (4) [Chen et al., 2022a, Theorem 4]. Similarly, the log-likelihood $\mathcal{L}(x_1)$, given a final sample $x_1$, can be found. This allows us to build a $\mathcal{T}$SB-based model for topological signals, following the ideas of De Bortoli et al. [2021]; Chen et al. [2022a].

We first parameterize the policies, $Z_t$ and $\hat{Z}_t$, by two learnable models $Z_t^\theta \equiv Z(t, x; \theta)$ and $\hat{Z}_t^{\hat{\theta}} \equiv \hat{Z}(t, x; \hat{\theta})$ with parameters $\theta$ and $\hat{\theta}$, resulting in the parameterized FB-$\mathcal{T}$SDEs. Then, we can perform a *likelihood training* by minimizing the following loss functions in an alternative fashion at initial and final signal samples $x_0$ and $x_1$

$$l(x_0; \hat{\theta}) = \int_0^1 \mathbb{E}_{X_t \sim (3a)}\left[\frac{1}{2}\|\hat{Z}_t^{\hat{\theta}}\|^2 + g_t \nabla \cdot \hat{Z}_t^{\hat{\theta}} + Z_t^{\theta\top}\hat{Z}_t^{\hat{\theta}}\Big| X_0 = x_0\right]\mathrm{d}t, \qquad (9a)$$

$$l(x_1; \theta) = \int_0^1 \mathbb{E}_{X_t \sim (3b)}\left[\frac{1}{2}\|Z_t^{\theta}\|^2 + g_t \nabla \cdot Z_t^{\theta} + \hat{Z}_t^{\hat{\theta}\top} Z_t^{\theta}\Big| X_1 = x_1\right]\mathrm{d}t, \qquad (9b)$$

which are, respectively, the upper bounds of the negative log-likelihoods (after dropping the unrelated terms) of the signal samples $x_0$ and $x_1$ given paths sampled from the FB-$\mathcal{T}$SDEs.

**Other choice of reference $\mathcal{T}$SDE.** In this work, we mainly consider reference dynamics following $\mathcal{T}$SHeat$_{\text{BM}}$, $\mathcal{T}$SHeat$_{\text{VE}}$ and $\mathcal{T}$SHeat$_{\text{VP}}$. For the dynamics involved with Hodge Laplacians in a SC$_2$, we may further allow *heterogeneous* diffusion based on the down and up parts of the Laplacian. Bunne et al. [2023] proposed to better initialize the SB model using the closed-form Gaussian SB. Likewise, we can consider the G$\mathcal{T}$SB in (7) as a stronger prior process, which yet requires a GP approximation from signal samples. We also consider *fractional Laplacian* for some cases to enable a more efficient exploration of the network [Riascos & Mateos, 2014] due to its non-local nature.

**Topological neural networks (TNNs).** While De Bortoli et al. [2021]; Chen et al. [2022a] applied convolutional neural networks for the Euclidean SBP, we naturally consider parameterizing the policies using the emergent TNNs. For node signals, we could consider graph convolution networks

(GCNs) [Kipf & Welling, 2017]; and likewise, for edge flows in a $SC_2$, simplicial neural networks (SNNs) [Roddenberry et al., 2021]. These *topology-aware* models perform convolutional learning upon the topological structure, more efficient with less parameters and better in performance.

**Complexity.** Like standard SB models, $\mathcal{T}$SB-based models also require simulations of the FB-$\mathcal{T}$SDEs. The key difference is that these models operate over topological networks where the drift (1) involves a matrix-vector multiplication $H_t Y_t$. However, this is essentially a *recursive* iteration of $LY_t$, which is efficient due to the *typically sparse* structure of $L$, reflecting the underlying topological sparsity. Moreover, our TNN-parameterized policies are also efficient for the same reason.

**Connection to other models.** As discussed in Chen et al. [2022a], in the special case of $Z_t \equiv 0$ and $\hat{Z}_t$ as the *score function* (scaled by $g_t$), the likelihood of SB models reduces to that of the score-based models [Song et al., 2020b] when $\nu_1$ is a simple Gaussian and the forward process is designed to reach $\nu_1$. Furthermore, if the reference process is poorly designed. SB models can still guide the process to the target distribution through these learnable policies, thus generalizing score-based models. On the other hand, for FB-$\mathcal{T}$SDEs, we can also obtain *probability flow ODEs* [Chen et al., 2018; Song et al., 2020b] which share the same time-marginals and likelihoods, allowing for exact likelihood evaluation of the model. Training through the likelihood of these flow ODEs naturally links to flow-based models. While there are no direct score-based or flow-based models for topological signals, the above discussions apply to $\mathcal{T}$SB-based models. We refer to Appendix E for more details where we show how the variants of these models including the *diffusion bridge* models [Zhou et al., 2024] for topological processes can be constructed.

## 6 EXPERIMENTS

First, we validate the theoretical results on G$\mathcal{T}$SB using the synthetic graph in Fig. 1. Here, we aim to bridge a zero-mean graph Matérn GP $\nu_0$ with $\Sigma_0 = (I + L)^{-1.5}$ and a diffusion GP $\nu_1$ with $\Sigma_1 = \exp(-20L)$. Using the $\mathcal{T}$SHeat$_{\text{BM}}$ and $\mathcal{T}$SHeat$_{\text{VE}}$ reference dynamics, we obtain the closed-form $X_t$ in (5), from which we further compute the covariance $\Sigma_t$ [cf. Corollary D.1]. We measure the *Bures-Wasserstein* distance between $\Sigma_t$ and $\Sigma_1$. From the *right-hand-side* figure, we see that both bridges reach the target distribution. The bridges exhibit distinct behaviors depending on the

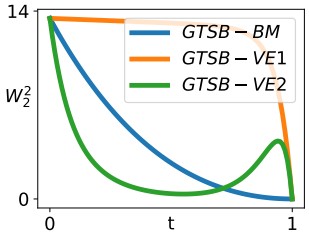

reference dynamics, as demonstrated by the disparate curves for $\mathcal{T}$SHeat$_{\text{VE}}$ with $c = 0.01$ and 10. This highlights the flexibility of $\mathcal{T}$SB-models in exploring a large space of topological bridges.

We then focus on evaluating $\mathcal{T}$SB-based models for topological signal *generation* ($\nu_1$ is a Gaussian noise) and *matching* (general $\nu_1$) in different applications, with the goal of investigating the question: ***whether $\mathcal{T}$SB-based models are beneficial for these tasks compared to the standard SB-based models?*** For this goal, we consider as the baseline SB-based models in Euclidean domains which use BM, VE and VP as reference dynamics [Chen et al., 2022a], labeled as `SB-BM`, `SB-VE` and `SB-VP`, respectively. We consider $\mathcal{T}$SB-based models using $\mathcal{T}$SHeat$_{\text{BM}}$, $\mathcal{T}$SHeat$_{\text{VE}}$ and $\mathcal{T}$SHeat$_{\text{VP}}$ as references, labeled as `TSB-BM`, `TSB-VE` and `TSB-VP`, respectively. We refer to Appendix F for the experimental details and additional results, as well as complexity analyses in Appendix F.2.5.

**Topological signal matching.** We first consider matching two sets of fMRI brain signals from the Human Connectome Project [Van Essen et al., 2013], which represent the liberal (with high energy, as the initial) and aligned (with low energy, as the final) brain activities, respectively. We use the recommended brain graph [Glasser et al., 2016] that connects 360 parcelled brain regions with edge weights denoting the connection strength. From Fig. 2, we see that a `TSB-VE` model learns to reach at a final state with low energy indicating the aligned activity, whereas `SB-VE` fails.

We then consider the single-cell embryoid body data that describes cell differentiation over 5 time-points [Moon et al., 2019]. We follow the preprocessing from Tong [2023]; Tong et al. [2024a;b]. We aim to transport the initial observations to the final state. Our method relies on the affinity graph constructed from the *entire set of observations* (∼18k). We define two normalized indicator functions as the boundary distributions, which specify the nodes corresponding to the data observed at the first and last timepoints. Fig. 3 shows the two-dim `phate` embeddings of the groundtruth and predicted data points using `TSB-BM` and `SB-BM` models. Here, `SB-BM` gives very noisy predictions, especially for intermediate ones, even when trained on the full dataset (see Table F.7).

Figure 2: Energies of true *(Left)* final state and the predictions obtained from `TSB-VE` *(Center)* and `SB-VE` *(Right)* models.

Figure 3: `Phate` embeddings of the single-cell data observations *(Left)* and predictions based on `TSB-BM` *(Center)* and `SB-BM` *(Right)* models.

Table 1: 1-Wasserstein distances for generating and matching tasks across datasets over five runs, where $\star$ indicates using `GSB-VE` and `GTSB-VE` for ocean currents.

| Method | Seismic magnitudes | Traffic flows | Brain signals | Single-cell data | Ocean currents |
|---|---|---|---|---|---|
| SB-BM | $11.73_{\pm0.05}$ | $18.69_{\pm0.02}$ | $12.08_{\pm0.08}$ | $0.33_{\pm0.01}$ | $7.21_{\pm0.00}$ |
| SB-VE | $11.49_{\pm0.04}$ | $19.04_{\pm0.02}$ | $17.46_{\pm0.14}$ | $0.33_{\pm0.01}$ | $7.17_{\pm0.02}$ |
| SB-VP | $12.61_{\pm0.06}$ | $18.22_{\pm0.03}$ | $13.41_{\pm0.05}$ | $0.33_{\pm0.01}$ | $0.83_{\pm0.01}\star$ |
| TSB-BM | $9.01_{\pm0.03}$ | $10.57_{\pm0.02}$ | $7.51_{\pm0.08}$ | $0.14_{\pm0.03}$ | $6.94_{\pm0.01}$ |
| TSB-VE | $7.69_{\pm0.04}$ | $10.51_{\pm0.02}$ | $7.59_{\pm0.05}$ | $0.14_{\pm0.02}$ | $6.89_{\pm0.00}$ |
| TSB-VP | $8.40_{\pm0.04}$ | $9.92_{\pm0.02}$ | $7.67_{\pm0.11}$ | $0.14_{\pm0.01}$ | $0.53_{\pm0.00}\star$ |

Edge flows have been used to model vector fields upon a discrete Hodge Laplacian estimate of the manifold Helmholtzian [Chen et al., 2021b]. Following the setup there, we consider the edge-based ocean current matching in a $SC_2$ ($\sim$20k edges). With an edge GP, learned by Yang et al. [2024] from drifter data, as the initial distribution modeling the currents, we synthetize a curl-free edge GP as the final one, modeling different behaviors of currents. From Fig. 4, we see that `SB-BM` fails to reach the final curl-free state, while `TSB-BM` becomes more divergent, ultimately closer to the target.

For these matching tasks, we evaluate the forward final predictions using 1-Wasserstein distances in Table 1, showing the consistent superiority of $\mathcal{T}$SB-based models over SB ones. We reasonably argue that this difference is due to the improper reference in SB-based models, which overlooks the underlying topology. This highlights the role of topology using $\mathcal{T}$SB-based models in these tasks.

**Generative modeling.** We model the magnitudes of yearly seismic events from IRIS as node signals on a mesh graph of 576 nodes based on the geodesic distance between the vertices of an icosahedral triangulated earth surface [Moresi & Mather, 2019]. We also consider the traffic flow from PeMSD4 dataset modeled as edge flows on a $SC_2$ with 340 edges [Chen et al., 2022b]. From Table 1, we see that $\mathcal{T}$SB-based models consistently outperform SB-based models also for signal generation tasks, highlighting the importance of topology-aware reference processes.

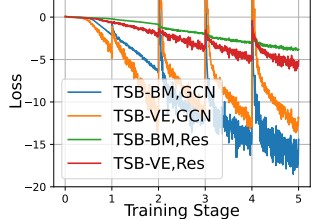

**Effect of policy models.** From the training curves of `TSB-BM/VE` for ResBlock and GCN as policy models on the right, we see that the training converges much faster and better using GCN compared to the former. This underlines the positive effect of TNNs on topological signal generative modeling. We refer to Tables F.2 and F.3 for the performance metrics of other bridge models with the two policy parameterizations on both seismic and traffic datasets.

**Effect of G$\mathcal{T}$SB prior.** Instead of using $\mathcal{T}$SHeat$_{BM}$ or $\mathcal{T}$SHeat$_{VE}$ as the reference, we here consider their corresponding closed-form SDEs (7) as the reference, imposing on the bridges a stronger prior carrying the moment information of the data samples [Bunne et al., 2023]. For ocean current matching, we show the samples from the learned FB-$\mathcal{T}$SDEs using `GTSB-BM` in Fig. 4, which arrives at a more faithful final state compared to `TSB-BM`, as also evaluated in Table 1.

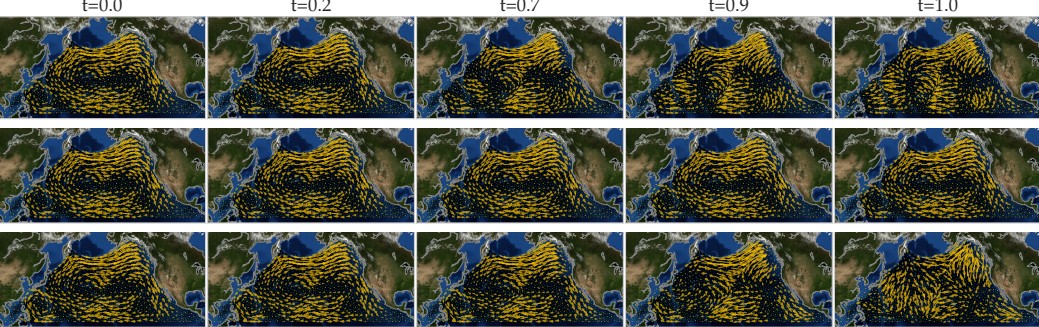

Figure 4: Forward sampled currents using `TSB-BM` *(Top)*, `SB-BM` *(Center)* and `GTSB-BM` *(Bottom)*.

## 7 DISCUSSION AND CONCLUSION

In this work, we demonstrated how to construct $\mathcal{T}$SB-based topological signal matching models within the likelihood training framework [Chen et al., 2022a]. We here discuss a few promising future directions based on emerging work and unexplored theoretical results.

**On model training.** Peluchetti [2023]; Shi et al. [2023] applied iterative Markovian fitting (IMF), as an alternative of IPF, to the classical SBP. This algorithm, trained via score matching, extends to $\mathcal{T}$SB-models with $\mathcal{T}$SHeat$_{\text{BM}}$ or $\mathcal{T}$SHeat$_{\text{VE}}$ as the reference, thanks to their closed-form transition kernels in (2). Recent work proposed (partially) *simulation-free* training of SB models. Tong et al. [2024b] learns the optimal SB by flow and score matching the forward SDE upon a heuristic E-OT. Korotin et al. [2024]; Gushchin et al. [2024] modeled Schrödinger potentials using Gaussian mixtures, enabling light training for the optimal drift, and Deng et al. [2024] linearized the forward policy. However, these methods require the reference dynamics to be either Wiener process or have scalar drifts. While training $\mathcal{T}$SB-based models remain scalable w.r.t. the topology size (see Appendix F.2.5), extending these approaches to our models is worthwhile but nontrivial.

**On model improving.** We focused on the reference dynamics driven by a topological convolution $H_t$ up to order one. It is however worthwhile to consider more involved (potentially learnable) convolutions to impose more general priors or incorporate physics knowledge of the process. The scalar diffusion coefficient $g_t$ could be extended as matrix-valued, enabling spatially correlated noising processes over the topology. On the other hand, SB models perform a kinetic energy minimization from the SOC view. [Liu et al., 2024] considered *generalized* SBP by adding a cost term which can model other knowledge of the process. This broadens the applicability of SB models and $\mathcal{T}$SB models could benefit from this, when there are external interactions with the topological process or prior knowledge on the process, such as enforcing curl-free edge flows.

**On other models.** While we showed the connections of $\mathcal{T}$SB to stochastic interpolants, flow- and score-based models, as well as diffusion bridges, we notice that the $\mathcal{T}$SB optimality can be interpreted as a *Wasserstein gradient flow* (WGF) (see Appendix C.2). For example, the $\mathcal{T}$SHeat$_{\text{BM}}$-driven $\mathcal{T}$SB is the WGF of a functional $\mathcal{F}(\nu)$ of some measure $\nu$ with $\mathcal{F}(\nu) = c \int \frac{1}{2} x^\top L x \cdot \nu(x) \, \mathrm{d}x + \frac{1}{2} g^2 \int \nu \log \nu \, \mathrm{d}x$ where $\mathcal{D}(x) := \frac{1}{2} x^\top L x$ is the *Dirichlet energy* of $x$ and the second term is the negative entropy. Thus, the $\mathcal{T}$SHeat$_{\text{BM}}$-driven forward $\mathcal{T}$SB essentially reduces the Dirichlet energy. This may not always align with the needs of real-world applications, which in turn motivates developing topological dynamics learning models via the JKO flow [Jordan et al., 1998] of a parametrized functional $\mathcal{D}(x)$ on topology, akin to approaches used in Euclidean domains [Bunne et al., 2022].

We focused on a fixed topological domain, but it is also of interest to study the case where $\mathcal{T}$ itself evolves over time. The $\mathcal{T}$SBP in this scenario may rely on a time-varying operator $L_t$ to guide the reference process. This is relevant for recent generative models for graphs, to name a few [Niu et al., 2020; Jo et al., 2022; Liu et al., 2023], where the graph structure, together with node features, are learned in the latent space based on diffusion models [Song et al., 2020b]. Lastly, we remark that discrete distributions on topological domains may be defined. For instance, nodes of a graph can represent discrete states where node $i$ is associated with a discrete probability $P_i$. This motivates the emerging geneartive models for discrete data [Austin et al., 2021; Ye et al., 2022; Haefeli et al., 2023; Campbell et al., 2024]. For a formal treatment of matching such discrete distributions on graphs, we refer to Maas [2011]; Léonard [2013]; Solomon [2018]; Chow et al. [2022].

**Conclusion.** With the goal of matching topological signal distributions beyond Euclidean domains, we introduced the $\mathcal{T}$SBP (topological Schrödinger bridge problem). We defined the reference process using an SDE driven by a topological convolution linear operator, which is tractable and includes the commonly used heat diffusion on topological domains. When the end distributions are Gaussians, we derived a closed-form $\mathcal{T}$SB, generalizing the existing results by Bunne et al. [2023]. In general cases, we showed that the optimal process satisfies a pair of FB-$\mathcal{T}$SDEs governed by some optimal policies. Building upon these results, we developed $\mathcal{T}$SB-based models where we parameterize the policies as (topological) neural networks and learn them from likelihood training, extending the framework of De Bortoli et al. [2021]; Chen et al. [2022a] to topological domains. We applied $\mathcal{T}$SB-based models for both topological signal generation and matching in various applications, demonstrating their improved performance compared to standard SB-based models. Overall, our work lies at the intersection of the SB-based distribution matching and topological machine learning, and we hope it inspires further research in this direction.

ACKNOWLEDGEMENTS

The author's research is funded by TU Delft AI Labs Programme and supported by professors Elvin Isufi and Geert Leus. The author is grateful for the financial support provided by the Multimedia computing group at TU Delft. Additionally, the author thanks the anonymous reviewers for their comments and valuable feedback, especially on the single-cell data experiments. Lastly, the author acknowledges the work of Bunne et al. [2023] for inspiring the proofs in Section 4.

REPRODUCIBILITY STATEMENT

For reproducibility of the theoretical results, the complete proofs of the claims can be found in the appendix, which is organized here. For reproducibility of the experiments, we refer to the GitHub repository at topological_SB_matching.

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

**Organizations** We include several appendices with additional details, proofs, derivations and experiments. This work concerns with the intersection of Schrödinger bridge theory, topological signal processing and learning, stochastic dynamics (on topology) and generative modeling. For that, we first introduce the necessary preliminaries on Schrödinger bridge theory in Appendix A, including needed theorems and lemmas for later. In Appendix B, we first provide an overview of topological signals and probabilistic methods. Then, we extensively discuss the topological stochastic dynamics based on the linear $\mathcal{T}$SDE, as well as its three instantiations: $\mathcal{T}$SHeat$_{\text{BM}}$, $\mathcal{T}$SHeat$_{\text{VE}}$ and $\mathcal{T}$SHeat$_{\text{VP}}$. Here, we provide the detailed derivations on how to obtain the transition kernels of the $\mathcal{T}$SDE, which are crucial for the $\mathcal{T}$SBP. In Appendix C, we discuss the optimality of $\mathcal{T}$SBP, relying on the existing results. Specifically, it includes the Schrödinger system for $\mathcal{T}$SBP, an SOC formulation of the $\mathcal{T}$SBP, the optimality, and the WGF interpretation. Appendix D proves the closed-form solution of the G$\mathcal{T}$SBP, along with the marginal and conditional statistics of the optimal path measure. In Appendices E and F, we provide more details on the $\mathcal{T}$SB-based models, their connections to other models, as well as the experiment details and additional results.

## A  PRELIMINARIES ON SCHRÖDINGER BRIDGES

### A.1  ON OPTIMAL TRANSPORT

**Theorem A.1** (Static Gaussian OT; [Janati et al., 2020])**.** *Let $\Sigma_0, \Sigma_1$ be positive definite. Given two Gaussian measures $\rho_0 \sim \mathcal{N}(\mu_0, \Sigma_0)$ and $\rho_1 \sim \mathcal{N}(\mu_1, \Sigma_1)$, the entropic-regularized optimal transport*

$$\min_{\pi \in \Pi(\mu_0, \mu_1)} \int_{\mathbb{R}^n \times \mathbb{R}^n} \frac{1}{2} \|x_0 - x_1\|^2 \, \mathrm{d}\pi(x_0, x_1) + \sigma^2 D_{\mathsf{KL}}(\pi \| \mu_0 \otimes \mu_1) \qquad \text{(E-OT)}$$

*admits a closed-form solution $\pi^\star$*

$$\pi^\star \sim \mathcal{N}\left( \begin{bmatrix} \mu_0 \\ \mu_1 \end{bmatrix}, \begin{bmatrix} \Sigma_0 & C_\sigma \\ C_\sigma^\top & \Sigma_1 \end{bmatrix} \right) \qquad (A.1)$$

*where*

$$C_\sigma = \frac{1}{2} (\Sigma_0^{\frac{1}{2}} D_\sigma \Sigma_0^{-\frac{1}{2}} - \sigma^2 I), \quad D_\sigma = (4\Sigma_0^{\frac{1}{2}} \Sigma_1 \Sigma_0^{\frac{1}{2}} + \sigma^4 I)^{\frac{1}{2}}. \qquad (A.2)$$

*Remark* 2. Note that while the above results are stated for positive definite covariance matrices (in order for $\rho_0$ and $\rho_1$ to have a Lebesgue density), the closed-form solution remains well-defined for positive semi-definite covariance matrices.

### A.2  ON SCHRÖDINGER BRIDGE

**Lemma A.3** (Léonard [2014])**.** *For a given measure $\mathbb{P}$ over the path space $\Omega$, let $\mathbb{P}^{xy}$ represent the conditioning of $\mathbb{P}$ on paths that take values $x$ and $y$ at $t = 0$ and $1$, respectively. That is, $\mathbb{P}^{xy} = \mathbb{P}[\cdot | X_0 = x, X_1 = y]$. Let $\mathbb{P}_{01}$ denote the joint probability for the values of paths at the two ends $t = 0, 1$. Then, $\mathbb{P}$ can be disintegrated into*

$$\mathbb{P}(\cdot) = \int_{\mathbb{R}^n \times \mathbb{R}^n} \mathbb{P}^{xy}(\cdot) \mathbb{P}_{01}(\mathrm{d}x \, \mathrm{d}y). \qquad \text{(Disintegration of Measures)}$$

**Static SBP** By Disintegration of Measures, for all $\mathbb{P} \in \mathcal{P}(\Omega)$, the relative entropy can be factorized as

$$D_{\mathrm{KL}}(\mathbb{P} \| \mathbb{Q}) = D_{\mathrm{KL}}(\mathbb{P}_{01} \| \mathbb{Q}_{01}) + \int_{\mathbb{R}^n \times \mathbb{R}^n} D_{\mathrm{KL}}(\mathbb{P}^{xy} \| \mathbb{Q}^{xy}) \mathbb{P}_{01}(\mathrm{d}x \, \mathrm{d}y), \qquad (A.3)$$

which implies that $D_{\mathrm{KL}}(\mathbb{P}_{01} \| \mathbb{Q}_{01}) \leq D_{\mathrm{KL}}(\mathbb{P} \| \mathbb{Q})$ with equality if and only if $\mathbb{P}^{xy} = \mathbb{Q}^{xy}$ for each $(x, y) \in \mathbb{R}^n \times \mathbb{R}^n$. This allows us to reduce the (*dynamic*) SBP to the *static* one.

$$\min D_{\mathrm{KL}}(\mathbb{P}_{01} \| \mathbb{Q}_{01}), \quad \text{s.t. } \mathbb{P} \in \mathcal{P}(\mathbb{R}^n \times \mathbb{R}^n), \mathbb{P}_0 = \rho_0, \mathbb{P}_1 = \rho_1. \qquad \text{(SBP}_{\text{static}})$$

Furthermore, it readily gives the following important theorem.

**Theorem A.4** (Föllmer [1988]; Léonard [2014])**.** *The SBP and SBP$_{static}$ admit, respectively, at most one solution. If SBP has the solution $\mathbb{P}$, then its joint-marginal at the end times $\mathbb{P}_{01}$ is the solution of SBP$_{static}$. Conversely, if $\mathbb{P}_{01}$ solves SBP$_{static}$, then the solution of SBP can be expressed as*

$$\mathbb{P}() = \int_{\mathbb{R}^n \times \mathbb{R}^n} \mathbb{Q}^{xy}() \mathbb{P}_{01}(\mathrm{d}x\,\mathrm{d}y), \tag{A.4}$$

*which means that $\mathbb{P}$ shares its bridges with $\mathbb{Q}$ (i.e., $\mathbb{P}$ is in the reciprocal class of $\mathbb{Q}$):*

$$\mathbb{P}^{xy} = \mathbb{Q}^{xy} \quad \forall (x, y) \in \mathbb{R}^n \times \mathbb{R}^n. \tag{A.5}$$

**Proposition A.5** (Léonard [2014])**.** *If the reference measure $\mathbb{Q}$ is Markov, then the solution $\mathbb{P}$ of SBP is also Markov.*

**Schrödinger System**

**Theorem A.6** (Jamison [1975]; Léonard [2014]; Chen et al. [2021a])**.** *Given two probability measures $\rho_0, \rho_1$ on $\mathbb{R}^n$ and the continuous, everywhere positive Markov kernel $p_{t|s}(y|x)$ (not necessarily associated to a scaled Brownian motion), there exists a unique pair of (up to scaling) of functions $\hat{\varphi}_0, \varphi_1$ on $\mathbb{R}^n$ such that the measure $\mathbb{P}_{01}$ on $\mathbb{R}^n \times \mathbb{R}^n$ defined by*

$$\mathbb{P}_{01} = \int_{\mathbb{R}^n \times \mathbb{R}^n} p_{1|0}(y|x) \hat{\varphi}_0(\mathrm{d}x) \varphi_1(\mathrm{d}y) \tag{A.6}$$

*has marginals $\rho_0$ and $\rho_1$. Moreover, the Schrödinger bridge from $\rho_0$ to $\rho_1$ induces the distribution flow*

$$\mathbb{P}_t = \varphi_t \hat{\varphi}_t \text{ with } \varphi_t(x) = \int p_{1|t}(y|x)\varphi_1(\mathrm{d}y), \hat{\varphi}_t(x) = \int p_{t|0}(x|y)\hat{\varphi}_0(\mathrm{d}y). \tag{A.7}$$

**SOC Formulation**   By *Girsanov's theorem*, Dai Pra [1991]; Pavon & Wakolbinger [1991] showed an equivalent SOC formulation of the SBP which aims to minimize the kinetic energy

$$\min_{u \in \mathcal{U}} \mathbb{E}\left[\int_0^1 \frac{1}{2}\|u(t, X_t)\|^2\right], \quad \text{s.t. } \begin{cases} \mathrm{d}X_t = u(t, X_t)\,\mathrm{d}t + \sigma\,\mathrm{d}W_t \\ X_0 \sim \rho_0, X_1 \sim \rho_1, \end{cases} \tag{SBP$_{soc}$}$$

where $\mathcal{U}$ is the set of finite control functions $u_t \equiv u(t, x)$. Given the SDE constraint in SBP$_{soc}$, the associated marginal density $\rho_t \equiv \rho(t, x)$ evolves according to the Fokker-Planck-Kolmogorov equation (FPK, Risken [1996]). This allows to arrive at an equivalent variational formulation

$$\min_{(\rho_t, u_t)} \int_{\mathbb{R}^n} \int_0^1 \frac{1}{2}\|u(t, x)\rho(t, x)\|^2\,\mathrm{d}t\,\mathrm{d}x \quad \text{s.t. } \begin{cases} \partial_t \rho_t + \nabla \cdot (\rho_t u_t) = \frac{\sigma^2}{2}\Delta\rho_t, \\ \rho_0 = \rho_0, \rho_1 = \rho_1. \end{cases} \tag{SBP$_{var}$}$$

Given $\rho_t = \varphi_t \hat{\varphi}_t$, the optimal control in SBP$_{soc}$ can be obtained by $u_t = \sigma^2 \nabla \log \varphi_t$.

# B   STOCHASTIC DYNAMICS ON TOPOLOGICAL DOMAINS

Compared to the Euclidean domain, the dynamics on topological domains are less studied. Here we provide some existing work on the dynamics on graphs and simplicial complexes. Note that our choice of the linear topological drift $f_t$ in (1) is analogous to the ideas in [Archambeau et al., 2007; Verma et al., 2024] which considered linear SDEs to approximate nonlinear dynamics, enabling approximations of more complex topological dynamics.

## B.1   PRELIMINARIES ON TOPOLOGICAL SIGNALS

Here we review the standard notions about topological signals, and we focus on the node signals and edge flows on graphs and simplicial complexes. *Note that we abuse some notions in this subsection.*

**Node signals**   Let $G = (V, E)$ be an unweighted graph where $V = \{1, \dots, n_0\}$ is the set of nodes and $E$ is the set of $n_1$ edges such that if nodes $i, j$ are connected, then $e = (i, j) \in E$. We can define real-valued functions on its node set $V \to \mathbb{R}$, collected into a vector $x = [x(1), \dots, x(n_0)] \in \mathbb{R}^{n_0}$, which is referred to as a node signal (or graph signal). Denote the oriented node-to-edge incidence

matrix by $B_1$ of dimension $n_0 \times n_1$. One can obtain the graph Laplacian by $L = B_1 B_1^\top$, which is a positive semi-definite linear operator on the space $\mathbb{R}^{n_0}$ of node signals.

A graph GP $x \sim \mathcal{N}(0, \Sigma)$ assumes $x$ is a random function with zero mean and a graph kernel (covariance matrix) $\Sigma$ which encodes the covariance between pairs of nodes. [Borovitskiy et al., 2021] constructed the diffusion and the Matérn graph GPs by extending the idea of deriving continuous GPs from SDEs, which have the kernels as follows

$$\Sigma_{\text{diffu}} = \exp\left(-\frac{\kappa^2}{2}L\right), \quad \Sigma_{\text{Matérn}} = \left(\frac{2\nu}{\kappa^2}I + L\right)^{-\nu} \tag{B.1}$$

where $\kappa > 0, \nu > 0$ are hyperparameters.

**Edge flows**  While it is possible to define edge flows on graphs, we consider a more general setting for a $\text{SC}_2$. A $\text{SC}_2$ generally contains $V, E, T$ three sets, where $V, E$ are the sets of nodes and edges, same as for graphs, and $T$ is the set of triangular faces (triangles) such that if $(i, j), (j, k), (i, k)$ form a closed triangle, then $t = (i, j, k) \in T$. Not all three pairwise connected edges are necessarily closed and included in $T$. For each edge and triangle, we assume the increasing order of their node labels as their reference orientation. (Note that the orientation of a general simplex is an equivalence class of permutations of its labels. Two orientations are equivalent (resp. opposite) if they differ by an even (resp. odd) permutation [Lim, 2020].) An oriented edge, denoted as $e = [i, j]$, is an ordering of $\{i, j\}$. This is not a directed edge allowing flow only from $i$ to $j$, but rather an assignment of the sign of the flow: from $i$ to $j$, it is positive and the reverse is negative. Likewise goes for the oriented triangle $t = [i, j, k]$. In a $\text{SC}_2$, we can define an edge flow by real-valued functions on its edges, collected in $x = [x(e_0), \ldots, x(e_{n_1})] \in \mathbb{R}^{n_1}$. which are required to be alternating, meaning that $x(\bar{e}) = -x(e)$ if $\bar{e} = [j, i]$ is oriented opposite to the reference $e = [i, j]$. Likewise, a triangle signal can be defined via an alternating function on triangles.

In the same spirit as graph Laplacians operating on node functions, we can define the discrete Hodge Laplacian operating on edge flows as $L = L_\text{d} + L_\text{u} := B_1^\top B_1 + B_2 B_2^\top$ where $B_2$ is the edge-to-triangle incidence matrix. The Hodge Laplacian $L$ describes the connectivity of edges where the down part $L_\text{d}$ and the up part $L_\text{u}$ encode how edges are adjacent, respectively, through nodes and via triangles. Moreover, the Hodge Laplacian $L$ is also positive semi-definite. Yang et al. [2024] generalized the graph GPs for edge flows, resulting in the diffusion and Matérn edge GPs with the kernels of the same forms as in (B.1) but with the Hodge Laplacian $L$ instead of the graph Laplacian $L$. Moreover, based on the combinatorial Hodge theory [Lim, 2020] that

$$\mathbb{R}^{n_1} = \text{im}(B_1^\top) \oplus \ker(L) \oplus \text{im}(B_2) \tag{B.2}$$

where $\text{im}(B_1^\top)$ is the gradient space, $\ker(L)$ the harmonic space and $\text{im}(B_2)$ the curl space, one can define the two types of edge GPs living in certain Hodge subspace $\square \in \{H, G, C\}$ with the kernels of the forms

$$\Sigma_{\text{diffu},\square} = \sigma_\square^2 U_\square \exp\left(-\frac{\kappa_\square^2}{2}\Lambda_\square\right)U_\square^\top, \quad \Sigma_{\text{Matérn},\square} = \sigma_\square^2 U_\square \left(\frac{2\nu_\square}{\kappa^2}I + \Lambda_\square\right)^{-\nu_\square} U_\square^\top \tag{B.3}$$

where $(U_\square, \Lambda_\square)$ are the eigenpairs of the Hodge Laplacian in the gradient (G), curl (C) and harmonic (H) subspaces, respectively [Yang et al., 2022b]. The samples from certain Hodge GP are in the corresponding Hodge subspace, which allows us to model the edge flows with different properties.

## B.2 PRELIMINARIES ON (STOCHASTIC) DIFFERENTIAL EQUATIONS

We are involved with differential equations in this work. In the following, we review some results on ordinary differential equations (ODEs) and SDEs which are required later.

### B.2.1 ON ODEs

Given an initial solution $x_0 \in \mathbb{R}^n$, consider a linear differential system of the form

$$\mathrm{d}x_t = A_t x_t \, \mathrm{d}t \tag{linear ODE}$$

with $A_t$ a time-varying matrix. To solve this system in closed-form, we require an expression for the state *transition matrix* $\Psi(t, s)$ which transforms the solution at $s$ to $t$, $x_t = \Psi(t, s)x_s$. For the general case, the closed-form of $\Psi(t, s)$ is not possible. In the following, we introduce an important class of matrices $A_t$ for which a closed-form solution is possible [Antsaklis & Michel, 1997].

**Lemma B.1** (Closed-form of the transition matrix of a linear ODE). *Given a linear ODE, if for every $s, t \geq 0$, we have*

$$A_t \left[ \int_s^t A_\tau \, \mathrm{d}\tau \right] = \left[ \int_s^t A_\tau \, \mathrm{d}\tau \right] A_t, \tag{B.4}$$

*then the transition matrix is given by*

$$\Psi(t, s) = \exp \left( \int_s^t A_\tau \, \mathrm{d}\tau \right) \triangleq I + \int_s^t A_\tau \, \mathrm{d}\tau + \frac{1}{2!} \left( \int_s^t A_\tau \, \mathrm{d}\tau \right)^2 + \dots. \tag{B.5}$$

For the scalar case, or when $A_t$ is diagonal, or for $A_t = A$, (B.4) is always true.

**Lemma B.2.** *For $A_t \in \mathbb{C}[\mathbb{R}, \mathbb{R}^{n \times n}]$, (B.4) is true if and only if $A_t A_s = A_s A_t$ for all $s, t$.*

**Lemma B.3** (Integration of matrix exponential). *For a nonsingular matrix $A$, we have*

$$\int_s^t \exp(A\tau) \, \mathrm{d}\tau = \left[ \exp(At) - \exp(As) \right] A^{-1}. \tag{B.6}$$

### B.2.2 ON SDEs

**Lemma B.4** (Itô isometry, Oksendal [2013]). *Let $W : [0, 1] \times \mathcal{X} \to \mathbb{R}$ denote the canonical real-valued Wiener process defined up to time $1$, and let $X : [0, 1] \times \mathcal{X} \to \mathbb{R}$ be a stochastic process that is adapted to the filtration generated by $W$. Then*

$$\mathbb{E} \left[ \left( \int_s^t X_t \, \mathrm{d}W_t \right)^2 \right] = \mathbb{E} \left[ \int_s^t X_t^2 \, \mathrm{d}t \right], \tag{B.7}$$

*and*

$$\mathbb{E} \left[ \left( \int_0^t X_t \, \mathrm{d}W_t \right) \left( \int_0^t Y_t \, \mathrm{d}W_t \right) \right] = \mathbb{E} \left[ \int_0^t X_t Y_t \, \mathrm{d}t \right]. \tag{B.8}$$

*This corollary allows us to compute the covariance of two stochastic processes $X_t$ and $Y_t$ that are adapted to the same filtration.*

**Transition densities of SDEs**  All Itô processes, that is, solutions to Itô SDEs, are *Markov processes*. This means that all Itô processes are, in a probabilistic sense, completely characterized by the transition densities (from $x_s$ at time $s$ to $x_t$ at time $t$, denoted by $p_{t|s}(x_t|x_s) \equiv p(x_s, s; x_t, t)$). The transition density is also a solution to the FPK equation with a degenerate (Dirac delta) initial density concentrated on $x_s$ at time $s$. We refer to Särkkä & Solin [2019, Thm 5.10].

### B.3 TRANSITION DENSITIES OF $\mathcal{T}$SDE

$\mathcal{T}$**SDE**  First, we can find the transition matrix of the associated ODE to $\mathcal{T}$SDE.

**Lemma B.5.** *For an ODE $\mathrm{d}y_t = H_t(L)y_t \, \mathrm{d}t$, the transition matrix is given by*

$$\Psi(t, s) =: \Psi_{ts} = \exp \left( \int_s^t H_\tau \, \mathrm{d}\tau \right) = I + \sum_{k=0}^\infty \frac{1}{k!} \left( \int_s^t H_\tau \, \mathrm{d}\tau \right)^k \qquad \text{(transition matrix)}$$

*with $y_t = \Psi_{ts} y_s$. Note that $\Psi_{ts}$ is symmetric since $H_t$ is a function of the symmetric $L$.*

This is a direct result from Lemmas B.1 and B.2 since $H_t H_\tau = H_\tau H_t$ for all $t, \tau$. By the definition of matrix integral, the computation of $\Psi_{ts}$ is given by

$$\Psi(t, \tau) = \exp(\tilde{H}_{t,\tau}(L)) = \exp \left( \sum_{k=0}^K \tilde{h}_k^{t,\tau} L^k \right) \tag{B.9}$$

where $\tilde{h}_k^{t,\tau} = \int_\tau^t h_{k,s} \, \mathrm{d}s$ are the integral of the scalar coefficients in $H_t$. In the following, we characterize the transition densities of the $\mathcal{T}$SDE, as well as the three concrete examples in Examples 1 to 3. Then, using the formulas in Särkkä & Solin [2019, Eq. 6.7], we can compute the statistics of the transition kernel.

The following two lemmas compose Lemma 4.

**Lemma B.6.** *The transition density $p_{t|s}(y_t|y_s)$ of the $\mathcal{T}SDE$ conditioned on $Y_s = y_s$ is Gaussian*

$$p_{t|s}(y_t|y_s) \sim \mathcal{N}(y_t; m_{t|s}, K_{t|s}) \tag{B.10}$$

*with the mean and covariance, for $t \geq s$, as follows*

$$m_{t|s} = \Psi_{ts} y_s + \Psi_t \int_s^t \Psi_\tau^{-1} \alpha_\tau \, \mathrm{d}\tau, \quad K_{t|s} = \Psi_t \left( \int_s^t g_\tau^2 \Psi_\tau^{-2} \, \mathrm{d}\tau \right) \Psi_t^\top.$$

*Proof.* Given the transition matrix, using the transition kernel formula in Särkkä & Solin [2019, Eq. 6.7], we have

$$m_{t|s} = \Psi_{ts} y_s + \int_s^t \Psi_{t\tau} \alpha_\tau \, \mathrm{d}\tau = \Psi_{ts} y_s + \Psi_t \int_s^t \Psi_\tau^{-1} \alpha_\tau \, \mathrm{d}\tau,$$

where we use the property $\Psi_{t\tau} = \Psi_t \Psi_\tau^{-1}$. Likewise, we have

$$K_{t|s} = \int_s^t g_\tau^2 \Psi_{t\tau} \Psi_{t\tau}^\top \, \mathrm{d}\tau = \Psi_t \left( \int_s^t g_\tau^2 \Psi_\tau^{-2} \, \mathrm{d}\tau \right) \Psi_t^\top.$$

$\square$

**Lemma B.7.** *Conditioned on $Y_0 = y_0$, the cross covariance $K(t_1, t_2)$ of $\mathcal{T}SDE$ at $t_1, t_2$ is given by*

$$K_{t_1, t_2} = \Psi_{t_1} \left( \int_0^{\min\{t_1, t_2\}} g_\tau^2 \Psi_\tau^{-2} \, \mathrm{d}\tau \right) \Psi_{t_2}^\top.$$

*Proof.* By applying the cross covariance function in Särkkä & Solin [2019, Sec 6.4], we have

$$K_{t_1, t_2} = \mathrm{Cov}[Y_{t_1}, Y_{t_2} | Y_0] = \mathbb{E}\left[ \left( \int_0^{t_1} g_{\tau_1} \Psi_{t_1, \tau_1} \, \mathrm{d}W_{\tau_1} \right) \left( \int_0^{t_2} g_{\tau_2} \Psi_{t_2, \tau_2} \, \mathrm{d}W_{\tau_2} \right)^\top \right]$$

$$= \int_0^{\min\{t_1, t_2\}} g_\tau^2 \Psi_{t_1, \tau} \Psi_{t_2, \tau}^\top \, \mathrm{d}\tau = \Psi_{t_1} \left( \int_0^{\min\{t_1, t_2\}} g_\tau^2 \Psi_\tau^{-2} \, \mathrm{d}\tau \right) \Psi_{t_2}^\top. \quad \text{(by Lemma B.4)}$$

$\square$

**$\mathcal{T}$SHeat$_{\text{BM}}$** Given an initial sample $y_0$ of the random topological signal $Y_0$, consider the SDE:

$$\mathrm{d}Y_t = -cLY_t \, \mathrm{d}t + g \, \mathrm{d}W_t. \tag{$\mathcal{T}$SHeat$_{\text{BM}}$}$$

Note that when $L$ is singular, we consider a perturbed version $L + \epsilon I$ with a small constant $\epsilon > 0$. Its steady-state distribution has zero mean and covariance matrix $\Sigma = \frac{g^2}{2c} L^{-1}$. The transition matrix of the associated ODE $\mathrm{d}Y_t = -cLY_t \, \mathrm{d}t$ is given by

$$\Psi_{ts} = \exp\left(-c(t-s)L\right). \tag{B.11}$$

**Lemma B.8.** *The transition density $p_{t|s}(y_t|y_s)$ of the $\mathcal{T}SHeat_{BM}$ conditioned on $Y_s = y_s$ is Gaussian with the mean and covariance, for $t \geq s$, as follows*

$$m_{t|s} = \Psi_{ts} y_s, \quad K_{t|s} = \frac{g^2}{2c} \left[ I - \exp(-2cL(t-s)) \right] L^{-1}.$$

*Moreover, the conditional dynamics $Y_t | Y_0 = y_0$ has the covariance process at $t_1, t_2$ as*

$$K_{t_1, t_2} = \frac{g^2}{2c} \left[ \exp(-cL|t_2 - t_1|) - \exp(-cL(t_1 + t_2)) \right] L^{-1}. \tag{B.12}$$

*Proof.* For the conditional mean, we can obtain it directly from the transition matrix of the associated ODE. For the conditional covariance, we have

$$K_{t|s} = \int_s^t \Psi(t, \tau) g^2 \Psi(t, \tau)^\top \, \mathrm{d}\tau = \Psi_t \left( \int_s^t \Psi_\tau^{-2} g^2 \, \mathrm{d}\tau \right) \Psi_t^\top$$

$$= g^2 \Psi_t \left( \int_s^t \exp(2cL\tau) \, \mathrm{d}\tau \right) \Psi_t^\top \qquad \text{(by } \Psi_\tau^{-2} = \exp(2cL\tau)\text{)}$$

$$= \frac{g^2}{2c} \Psi_t \left[ \exp(2cLt) - \exp(2cLs) \right] L^{-1} \Psi_t^\top \qquad \text{(by Lemma B.3)}$$

$$= \frac{g^2}{2c} \left[ I - \exp(-2cL(t-s)) \right] L^{-1}. \qquad \text{(} \Psi_t = \exp(-cLt)\text{)}$$

To compute the covariance process of the conditional dynamics $Y_t|Y_0 = y_0$, by definition, we have, for $t_1 \leq t_2$

$$
\begin{aligned}
K_{t_1,t_2} = \mathrm{Cov}[Y_{t_1}, Y_{t_2}|Y_0] &= \Psi_{t_1}\left(\int_0^{t_1} g^2\Psi_\tau^{-2}\,\mathrm{d}\tau\right)\Psi_{t_2}^\top && \text{(by Lemma 4)} \\
&= g^2\Psi_{t_1}\left[\int_0^{t_1}\exp(2cL\tau)\,\mathrm{d}\tau\right]\Psi_{t_2}^\top && \text{(by } \Psi_\tau^{-2} = \exp(2cL\tau)) \\
&= g^2\Psi_{t_1}\left[\frac{1}{2c}\big[\exp(2cLt_1) - I\big]\right]L^{-1}\Psi_{t_2}^\top && \text{(by Lemma B.3)} \\
&= \frac{g^2}{2c}\big[\exp(-cL(t_2 - t_1)) - \exp(-cL(t_1 + t_2))\big]L^{-1}.
\end{aligned}
$$

The case of $t_1 > t_2$ can be similarly derived, which completes the proof. $\qquad\square$

### $\mathcal{T}\text{SHeat}_{\text{VE}}$

**Lemma B.9.** *The Gaussian transition kernel $p(t|s)$ of $\mathcal{T}\text{SHeat}_{\text{VE}}$ has the mean and covariance*

$$
m_{t|s} = \Psi_{ts}y_s, \quad K_{t|s} = \sigma_{\min}^2\ln\left(\frac{\sigma_{\max}}{\sigma_{\min}}\right)\exp(-2cLt)\big[\exp(2At) - \exp(2As)\big]A^{-1} \quad \text{(B.13)}
$$

*where $\Psi_{ts}$ is the same as (B.11) and $A = \ln\left(\frac{\sigma_{\max}}{\sigma_{\min}}\right)I + cL$. The cross covariance between $Y_t$ and $Y_s$, conditioned on $Y_0$, is given by*

$$
K_{t_1,t_2} = \sigma_{\min}^2\ln\left(\frac{\sigma_{\max}}{\sigma_{\min}}\right)\exp(-cL(t_1 + t_2))[\exp(2A\min\{t_1, t_2\}) - I]A^{-1}. \quad \text{(B.14)}
$$

*Proof.* As the associated ODE of $\mathcal{T}\text{SHeat}_{\text{VE}}$ is also a topological heat diffusion, the transition matrix is the same as $\Psi_{ts} = \exp(-cL(t - s))$ for $\mathcal{T}\text{SHeat}_{\text{BM}}$. By substituting $\sigma(t)$ into $g_t$, we can find that

$$
g_t = \sigma_{\min}\left(\frac{\sigma_{\max}}{\sigma_{\min}}\right)^t\sqrt{2\ln\left(\frac{\sigma_{\max}}{\sigma_{\min}}\right)}. \quad \text{(B.15)}
$$

For the covariance of the transition density, we have

$$
K_{t|s} = \Psi_t\left(\int_s^t g_\tau^2\Psi_\tau^{-2}\,\mathrm{d}\tau\right)\Psi_t^\top
$$

for which, we need to compute the integral

$$
\begin{aligned}
\int_s^t g_\tau^2\Psi_\tau^{-2}\,\mathrm{d}\tau &= \int_s^t \sigma_{\min}^2\left(\frac{\sigma_{\max}}{\sigma_{\min}}\right)^{2\tau}2\ln\left(\frac{\sigma_{\max}}{\sigma_{\min}}\right)\exp(2cL\tau)\,\mathrm{d}\tau \\
&= 2\sigma_{\min}^2\ln\left(\frac{\sigma_{\max}}{\sigma_{\min}}\right)\left[\int_s^t\left(\frac{\sigma_{\max}}{\sigma_{\min}}\right)^{2\tau}\exp(2cL\tau)\,\mathrm{d}\tau\right] && \text{(factor out the constant)} \\
&= 2\sigma_{\min}^2\ln\left(\frac{\sigma_{\max}}{\sigma_{\min}}\right)\int_s^t\exp\left[2\tau\ln\left(\frac{\sigma_{\max}}{\sigma_{\min}}\right)\right]\exp(2cL\tau)\,\mathrm{d}\tau && \text{(by the identity } \exp(\ln x) = x) \\
&= 2\sigma_{\min}^2\ln\left(\frac{\sigma_{\max}}{\sigma_{\min}}\right)\int_s^t\exp(2\tau A)\,\mathrm{d}\tau && \text{(by } A = \ln(\sigma_{\max}/\sigma_{\min})I + cL) \\
&= \sigma_{\min}^2\ln\left(\frac{\sigma_{\max}}{\sigma_{\min}}\right)(\exp(2At) - \exp(2As))A^{-1}. && \text{(by Lemma B.3)}
\end{aligned}
$$

Thus, we have

$$
K_{t|s} = \sigma_{\min}^2\ln\left(\frac{\sigma_{\max}}{\sigma_{\min}}\right)\exp(-2cLt)\big[\exp(2At) - \exp(2As)\big]A^{-1}.
$$

For the cross covariance, assuming $t_1 \leq t_2$, then we can find the covariance kernel as

$$K_{t_1,t_2} = \text{Cov}[Y_{t_1}, Y_{t_2}|Y_0] = \Psi_{t_1}\left[\int_0^{t_1} g_\tau^2 \Psi_\tau^{-2}\, d\tau\right]\Psi_{t_2}^\top \qquad \text{(by Lemma 4)}$$

$$= \Psi_{t_1}\left[\int_0^{t_1} \sigma_{\min}^2\left(\frac{\sigma_{\max}}{\sigma_{\min}}\right)^{2\tau} 2\ln\left(\frac{\sigma_{\max}}{\sigma_{\min}}\right)\exp(2cL\tau)\, d\tau\right]\Psi_{t_2}^\top \quad \text{(since } \Psi_\tau^{-2} = \exp(2cL\tau))$$

$$= \sigma_{\min}^2 \ln\left(\frac{\sigma_{\max}}{\sigma_{\min}}\right)\Psi_{t_1}\left[\exp(2At_1) - I\right]A^{-1}\Psi_{t_2}^\top \qquad \text{(using the same steps as above)}$$

$$= \sigma_{\min}^2 \ln\left(\frac{\sigma_{\max}}{\sigma_{\min}}\right)\exp(-cL(t_1 + t_2))[\exp(2At_1) - I]A^{-1}.$$

The similar steps can be followed for $t_1 > t_2$, which completes the proof. $\qquad \square$

$\mathcal{T}\text{SHeat}_{\text{VP}}$  For this stochastic process, a closed-form transition kernel cannot be found. Yet, we could proceed the following for numerical computations. First, we can find the closed-form transition matrix of the associated ODE as

$$\Psi_{ts} = \exp\left(\int_s^t -\left(\frac{1}{2}\beta(\tau)I + cL\right)d\tau\right) = \exp\left(-cL(t - s) - \frac{1}{2}\int_s^t \beta(\tau)\, d\tau\right) \qquad \text{(B.16)}$$

where the integral can be easily obtained as

$$\int_s^t \beta(\tau)\, d\tau = \left[\frac{1}{2}\tau^2(\beta_{\max} - \beta_{\min}) + \tau\beta_{\min}\right]_s^t =: \tilde{\beta}_{ts}. \qquad \text{(B.17)}$$

This allows to compute the mean of the transition kernel $m_{t|s}$ given an initial solution $y_s$. For the covariance kernel, we have

$$K_{t|s} = \Psi_t\left(\int_s^t \beta(\tau)\Psi_\tau^{-2}\, d\tau\right)\Psi_t^\top$$

where the integral can be expressed as

$$\int_s^t \beta(\tau)\Psi_\tau^{-2}\, d\tau = \int_s^t \left(\tau(\beta_{\max} - \beta_{\min}) + \beta_{\min}\right)\exp\left(2cL\tau + \tilde{\beta}_{\tau 0}\right)d\tau$$

$$= \left[\left(\tau(\beta_{\max} - \beta_{\min}) + \beta_{\min}\right)v(\tau)\right]\Big|_s^t - (\beta_{\max} - \beta_{\min})\int_0^s v(\tau)\, d\tau. \qquad \text{(integration by parts)}$$

Here, we denote $v(\tau) = \int \exp\left(2cL\tau + \tilde{\beta}_{\tau 0}\right)d\tau$, thus $v'(\tau) := \exp\left(2cL\tau + \tilde{\beta}_{\tau 0}\right)$, which does not have a simple closed-form, we need to compute it numerically. This gives the covariance kernel. Following the similar procedures, we can compute the cross covariance $K_{t_1,t_2}$ of the conditional process $Y_t|Y_0 = y_0$.

## B.4 Other Topological Dynamics

We may consider *fractional Laplacian* in $\mathcal{T}\text{SHeat}$ which allows for a more efficient exploration of the network [Riascos & Mateos, 2014] due to its non-local nature.

For $\mathcal{T}\text{SHeat}$, we can further allow *heterogeneous heat diffusion* on the edge space as follows

$$dY_t = -(c_1 L_d + c_2 L_u)Y_t\, dt + g_t\, dW_t \qquad \text{(B.18)}$$

by setting $H_t = -(c_1 L_d + c_2 L_u)$, with $c_1, c_2 > 0$. Here, the diffusion rates are different for the different edge-adjacency types encoded in $L_d$ and $L_u$. This in fact can be generalized to using a more general topological convolution operator, if $L := L_d + L_u$ consists of the down and up parts,

$$H_t = \sum_{k=0}^{K_1} h_k^1(t)L_d^k + \sum_{k=0}^{K_2} h_k^2(t)L_u^k \qquad \text{(B.19)}$$

where $h_k^1(t), h_k^2(t)$ are the coefficients of the topological convolution. We refer to Yang et al. [2022b] for more details on its expressive power compared to $\mathcal{T}\text{SDE}$.

## C  TOWARDS THE OPTIMALITY OF TOPOLOGICAL SBP

**Proposition C.1** ($\mathcal{T}$-Schrödinger System; [Chen et al., 2016; Jamison, 1975]). *The optimal solution of* $\mathcal{T}SBP_{static}$ *has the form* $\mathbb{P}_{01} = \int_{\mathbb{R}^n \times \mathbb{R}^n} \hat{\varphi}_0(x_0) p_{1|0}(x_1|x_0) \varphi_1(x_1) \, \mathrm{d}x_0 \, \mathrm{d}x_1$ *with* $\varphi$ *and* $\hat{\varphi}$ *satisfying the system*

$$\varphi_t(x_t) = \int_{\mathbb{R}^n} p_{1|t}(x_1|x_t) \varphi_1(x_1) \, \mathrm{d}x_1, \quad \hat{\varphi}_t(x_t) = \int_{\mathbb{R}^n} p_{t|0}(x_t|x_0) \hat{\varphi}_0(x_0) \, \mathrm{d}x_0 \qquad \text{(C.1)}$$

*where* $p_{t|s}(y|x) = \mathcal{N}(y; \mu_{t|s}, K_{t|s})$ *is the Gaussian transition density [cf.* Lemma B.6*] of* $\mathcal{T}SDE$ *with drift in (1). Moreover, the time-marginal at* $t$ *can be factored as* $\mathbb{P}_t(x) = \varphi_t(x) \hat{\varphi}_t(x)$.

*Proof.* This is a direct result of the Schrödinger system in Theorem A.6 by replacing the Markov kernel by that [cf. Lemma B.6] of the $\mathcal{T}$SDE. □

From this system, we see that the optimal path measure has its marginal $\mathbb{P}_t$ factorized into two time-marginals $\varphi_t$ and $\hat{\varphi}_t$, which are both governed by the $\mathcal{T}$SDE.

### C.1  VARIATIONAL FORMULATIONS OF $\mathcal{T}$SBP

By Girsanov's theorem, the $\mathcal{T}$SBP can be formulated as the minimum energy SOC problem:

$$\min_{b_t} \ \mathbb{E}\left[ \frac{1}{2} \int_0^1 \|b(t, X_t)\|^2 \, \mathrm{d}t \right], \quad \text{s.t.} \begin{cases} \mathrm{d}X_t = [f_t + g_t b(t, X_t)] \, \mathrm{d}t + g_t \, \mathrm{d}W_t \\ X_0 \sim \nu_0, X_0 \sim \nu_1 \end{cases} \qquad (\mathcal{T}\text{SBP}_{\text{soc}})$$

where $b_t \equiv b(t, X_t)$ is the control function. The SDE constraint in $\mathcal{T}\text{SBP}_{\text{soc}}$ is also known as the *controlled* SDE, in comparison to the *uncontrolled* reference $\mathcal{T}$SDE. It further leads to the variational problem

$$\min_{(b_t, \nu_t)} \ \frac{1}{2} \int_0^1 \int_{\mathbb{R}^n} \|b_t\|^2 \nu(t, x) \, \mathrm{d}x \, \mathrm{d}t, \ \text{s.t.} \begin{cases} \partial_t \nu_t + \nabla \cdot [\nu_t(f_t + g_t b_t)] = \frac{1}{2} g_t^2 \Delta \nu_t \\ \nu(0, x) = \nu_0, \nu(1, x) = \nu_1 \end{cases} \qquad (\mathcal{T}\text{SBP}_{\text{var}})$$

where $\nu_t \equiv \nu(t, x) \equiv \mathbb{P}_t$ is the time-marginal of $\mathbb{P}$ and follows some PDE constraint, which is the FPK equation of the SDE constraint in $\mathcal{T}\text{SBP}_{\text{soc}}$.

**Theorem C.2** ($\mathcal{T}$SBP Optimality; Léonard [2014]; Caluya & Halder [2021]). *Let* $\varphi_t \equiv \varphi(t, x)$ *and* $\hat{\varphi}_t \equiv \hat{\varphi}(t, x)$ *be the solutions to the pair of PDEs*

$$\begin{cases} \partial_t \varphi_t = -\nabla \varphi_t^\top f_t - \frac{1}{2} g_t^2 \Delta \varphi_t \\ \partial_t \hat{\varphi}_t = -\nabla \cdot (\hat{\varphi}_t f_t) + \frac{1}{2} g_t^2 \Delta \hat{\varphi}_t, \end{cases} \quad \text{s.t. } \varphi(0, \cdot) \hat{\varphi}(0, \cdot) = \nu_0, \varphi(1, \cdot) \hat{\varphi}(1, \cdot) = \nu_1. \qquad \text{(C.2)}$$

*Then, the optimal control in* $\mathcal{T}SBP_{var}$ *is* $b_t^\star = g_t^2 \nabla \log \varphi_t$ *and the optimal path measure is* $\nu_t = \mathbb{P}_t = \varphi_t \hat{\varphi}_t$. *Moreover, the solution to* $\mathcal{T}SBP$ *can be represented by the path measure of the following coupled (*forward-backward*)* $\mathcal{T}SDEs$

$$\mathrm{d}X_t = [f_t + g_t^2 \nabla \log \varphi(t, X_t)] \, \mathrm{d}t + g_t \, \mathrm{d}W_t, \quad X_0 \sim \nu_0, \qquad \text{(C.3a)}$$

$$\mathrm{d}X_t = [f_t - g_t^2 \nabla \log \hat{\varphi}(t, X_t)] \, \mathrm{d}t + g_t \, \mathrm{d}W_t, \quad X_1 \sim \nu_1, \qquad \text{(C.3b)}$$

*where* $\nabla \log \varphi(t, X_t)$ *and* $\nabla \log \hat{\varphi}(t, X_t)$ *are the forward and backward optimal drifts, respectively.*

*Proof.* The proof is an adaption of Caluya & Halder [2021] to the topological setting. First, we make the assumptions[1] that $g(t)$ is uniformly lower-bounded and $f(t, x; L)$ satisfies Lipschitz conditions

---

[1] The nonexplosive Lipschitz condition on $f_t$ rules out the finite-time blow up of the sample paths of the SDE, and ensures the existance and uniqueness. It, together with the uniformly lower-bounded diffusion $g_t$, guarantees the transition kernel $p_{t|s}$ is positive and everwhere continuous.

with at most linear growth in $x$. From the first oprder optimality conditions for the SOC formulation $\mathcal{T}\text{SBP}_{var}$, we can obtain a coupled system of nonlinear PDEs for $\psi_t$ (the potential function of $b_t$, i.e., $b_t = \nabla\psi_t$) and $\nu_t$, which are known as the Hamilton-Jacobi-Bellman (HJB) and FPK equations, respectively, as well as the optimal control $b_t^\star = g_t^2 \nabla \log \varphi_t$. Via the Hopf-Cole transform, this system returns (C.2) [Caluya & Halder, 2021, Thm 2]. Then, by substituting the optimal control into the constraint in $\mathcal{T}\text{SBP}_{soc}$, one can obtain the forward SDE, and the backward SDE can be derived from the time-reversal of the forward SDE [Anderson, 1982; Nelson, 2020]. $\qquad\square$

Using nonlinear Feynman-Kac formula (or applying Itô's formula on $\log\varphi_t$ and $\log\hat{\varphi}_t$), the PDE system (C.2) admits the SDEs [Chen et al., 2022a]

$$d\log\varphi_t = \frac{1}{2}\|Z_t\|^2\,dt + Z_t^\top\,dW_t, \tag{C.4}$$

$$d\log\hat{\varphi}_t = \left(\frac{1}{2}\|\hat{Z}_t\|^2 + \nabla\cdot(g_t\hat{Z}_t - f_t) + \hat{Z}_t^\top Z_t\right)dt + \hat{Z}_t^\top\,dW_t \tag{C.5}$$

where $Z_t \equiv g_t\nabla\log\varphi_t(X_t)$ and $\hat{Z}_t \equiv g_t\nabla\log\hat{\varphi}_t(X_t)$. This results in Proposition 5.

## C.2 WASSERSTEIN GRADIENT FLOW INTERPRETATION

The gradient flow of a funicional over the space of probability measures with Wasserstein metric, i.e., the Wasserstein gradient flow (WGF), is fundamentally linked to FPK equations [Otto, 2001; Ambrosio et al., 2008]. In the following, we show that solving the $\mathcal{T}\text{SBP}$ with $\mathcal{T}\text{SHeat}_{BM}$ reference amounts to solving the WGF of some functional on a probability measure $\nu$.

**Theorem C.3.** *Consider the $\mathcal{T}\text{SBP}_{var}$ with the reference process $\mathcal{T}\text{SHeat}$. The SB optimality [cf. (C.2)] respects a pair of FPK equations of the form*

$$\partial_t\hat{\varphi}_t(x) = \nabla\cdot(cLx\hat{\varphi}_t(x)) + \frac{1}{2}g^2\Delta\hat{\varphi}_t(x), \quad \hat{\varphi}_0(x) = \hat{\varphi}_0(x), \tag{C.6a}$$

$$\partial_t\rho_t(x) = \nabla\cdot(cLx\rho_t(x)) + \frac{1}{2}g^2\Delta\rho_t(x), \quad \rho_0(x) = \varphi_1(x)\exp(2cx^\top Lx/g^2). \tag{C.6b}$$

*Therefore, the Wasserstein gradient flow of $\mathcal{F}(v)$ recovers the paired PDE in solving $\mathcal{T}\text{SBP}_{var}$*

$$\mathcal{F}(\nu) = c\int_{\mathbb{R}^n}\frac{1}{2}x^\top Lx\cdot\nu(x)\,dx + \frac{1}{2}g^2\int_{\mathbb{R}^n}\nu\log\nu\,dx := c\,\mathbb{E}_\nu[\mathcal{D}(x)] + \frac{1}{2}g^2\mathcal{S}(\nu) \tag{C.7}$$

*where $\mathcal{D}(x) = \frac{1}{2}x^\top Lx$ is the Dirichlet energy of $x$ and $\mathcal{S}(\nu)$ is the negative differential entropy.*

*Proof.* First, from Theorem C.2, we have the PDE system in (C.2) which can be rewritten as the pair of PDEs in (C.6) by applying Caluya & Halder [2021, Thm 3]. Both PDEs are of the following FPK form with $V(x) := \frac{1}{2}cx^\top Lx$ on some density $p_t$

$$\partial_t p_t(x) = \nabla\cdot(p_t(x)\nabla V(x)) + \frac{1}{2}g^2\Delta p_t(x), \tag{C.8}$$

for some initial condition. We can view $V(x)$ as the potential energy of some function $f_t(x)$. Here, we have $f_t(x) = -cLx_t = -\nabla V(x)$. Then, from the seminal work Jordan et al. [1998], the flows generated by the PDEs in (C.6) (both of the FPK form) can be seen as the gradient descent of the Lyapunov functional $\mathcal{F}(\cdot)$ in the following form

$$\mathcal{F}(\cdot) = c\int_{\mathbb{R}^n}\frac{1}{2}x^\top Lx\cdot(\cdot)\,dx + \frac{1}{2}g^2\int_{\mathbb{R}^n}(\cdot)\log(\cdot)\,dx := c\,\mathbb{E}_{(\cdot)}[\mathcal{D}(x)] + \frac{1}{2}g^2\mathcal{S}(\cdot) \tag{C.9}$$

with respect to the 2-Wasserstein distance in the space $\mathcal{P}_2(\mathbb{R}^n)$ of probability measures on $\mathbb{R}^n$ with finite second moments. Here, $(\cdot)$ can be $\hat{\varphi}_t$ or $p_t$. $\qquad\square$

We note that it is also possible to obtain the associated functional for the $\mathcal{T}\text{SBP}$ with more general reference $\mathcal{T}\text{SDE}$ based on the similar argument, but it would be more involved and lead to a time-dependent functional [Ferreira & Valencia-Guevara, 2018].

## D    THE CLOSED-FORM OF GAUSSIAN TOPOLOGICAL SCHRÖDINGER BRIDGES [THEOREMS 6 AND 7] (PROOFS AND OTHERS)

For convenience, we state the Gaussian $\mathcal{T}$SBP

$$\min D_{\mathsf{KL}}(\mathbb{P} \,\|\, \mathbb{Q}_{\mathcal{T}}), \quad \text{s.t. } \mathbb{P} \in \mathcal{P}(\Omega), \nu_0 = \mathcal{N}(\mu_0, \Sigma_0), \nu_1 = \mathcal{N}(\mu_1, \Sigma_1) \qquad \text{(G}\mathcal{T}\text{SBP)}$$

and its *static* problem

$$\min D_{\mathsf{KL}}(\mathbb{P}_{01} \,\|\, \mathbb{Q}_{\mathcal{T}01}), \quad \text{s.t. } \mathbb{P}_{01} \in \mathcal{P}(\mathbb{R}^n \times \mathbb{R}^n), \mathbb{P}_{0\cdot} = \nu_0, \mathbb{P}_{\cdot 1} = \nu_1. \qquad \text{(G}\mathcal{T}\text{SBP}_{\text{static}}\text{)}$$

We also restate the main results of the Gaussian $\mathcal{T}$SBP.

**Theorem 6.** *Denote by $\mathbb{P}$ the solution to GTSBP with $\nu_0 = \mathcal{N}(\mu_0, \Sigma_0)$ and $\nu_1 = \mathcal{N}(\mu_1, \Sigma_1)$. Then, $\mathbb{P}$ is the path measure of a* Markov Gaussian process *whose marginal $X_t \sim \mathcal{N}(\mu_t, \Sigma_t)$ admits an expression in terms of the initial and final variables, $X_0, X_1$, as follows*

$$X_t = \bar{R}_t X_0 + R_t X_1 + \xi_t - R_t \xi_1 + \Gamma_t Z \qquad (5)$$

*where $Z \sim \mathcal{N}(0, I)$ is standard Gaussian, independent of $(X_0, X_1)$, and*

$$R_t = K_{t1} K_{11}^{-1}, \quad \bar{R}_t = \Psi_t - R_t \Psi_1, \quad \Gamma_t := \text{Cov}[Y_t | (Y_0, Y_1)] = K_{tt} - K_{t1} K_{11}^{-1} K_{1t}. \qquad (6)$$

**Corollary D.1** (Marginal Statistics). *The time marginal variable $X_t$ in (5) of the optimal solution to GTSBP has the mean and covariance as follows*

$$\mu_t = \bar{R}_t \mu_0 + R_t \mu_1 + \xi_t - R_t \xi_1, \qquad \text{(D.1a)}$$

$$\Sigma_t = \bar{R}_t \Sigma_0 \bar{R}_t^\top + R_t \Sigma_1 R_t^\top + \bar{R}_t C R_t^\top + R_t C^\top \bar{R}_t^\top + \Gamma_t, \qquad \text{(D.1b)}$$

*where $C = \Psi_1^{-1} K_{11}^{1/2} \tilde{C} K_{11}^{1/2}$ with*

$$\tilde{C} = \frac{1}{2}(\tilde{\Sigma}_0^{1/2} \tilde{D} \tilde{\Sigma}_0^{-1/2} - I), \qquad \tilde{D} = (4\tilde{\Sigma}_0^{1/2} \tilde{\Sigma}_1 \tilde{\Sigma}_0^{1/2} + I)^{1/2},$$
$$\tilde{\Sigma}_0 = K_{11}^{-1/2} \Psi_1 \Sigma_0 \Psi_1^\top K_{11}^{-1/2}, \quad \tilde{\Sigma}_1 = K_{11}^{-1/2} \Sigma_1 K_{11}^{-1/2}. \qquad \text{(D.2)}$$

**Theorem 7** (SDE representation). *Under the optimal $\mathbb{P}$, the process $X$ admits the SDE dynamics:*

$$\mathrm{d}X_t = f_{\mathcal{T}}(t, X_t; L)\,\mathrm{d}t + g_t\,\mathrm{d}W_t, \ \text{where } f_{\mathcal{T}}(t, x; L) = S_t^\top \Sigma_t^{-1}(x - \mu_t) + \dot{\mu}_t \qquad (7)$$

*with $\mu_t, \Sigma_t$ the mean and covariance of $X_t$ [cf. Corollary D.1] and we have*

$$S_t = P_t - Q_t^\top + H_t K_{tt} - K_{t1} K_{11}^{-1} \Upsilon_t^\top, \qquad (8)$$

*with $P_t = (R_t \Sigma_1 + \bar{R}_t C)\dot{R}_t^\top, Q_t = -\dot{\bar{R}}_t(C R_t^\top + \Sigma_0 \bar{R}_t^\top), \Upsilon_t = H_t K_{t1} + g_t^2 \Psi_t^{-1} \Psi_1^\top$, where $C$ is the covariance of $X_0, X_1$ in the optimal $\mathbb{P}_{01}$.*

### D.1    PRELIMINARIES FOR THE PROOF

We first introduce the following three lemmas and the definition of infinitesimal generators.

**Lemma D.2** (Central identity of Quantum Field Theory [Zee, 2010]). *For all matrix $M \succ 0$ and all sufficiently regular analytical function $v$ (e.g., polynomials or $v \in \mathcal{C}^\infty(\mathbb{R}^d)$ with compact support), we have*

$$(2\pi)^{-\frac{d}{2}} (\det M)^{-\frac{1}{2}} \int_{\mathbb{R}^d} v(x) \exp\left(-\frac{1}{2} x^\top M x\right) \mathrm{d}x = \exp\left(\frac{1}{2} \partial_x^\top M^{-1} \partial_x\right) v(x)\Big|_{x=0} \qquad \text{(D.3)}$$

*where $\exp(D) = I + D + \frac{1}{2}D^2 + \cdots$, for a differential operator $D$.*

**Lemma D.3** (Conditional Gaussians). *Let $(Y_0, Y_1) \sim \mathcal{N}\left(\begin{bmatrix} \mu_0 \\ \mu_1 \end{bmatrix}, \begin{bmatrix} \Sigma_{00} & \Sigma_{01} \\ \Sigma_{10} & \Sigma_{11} \end{bmatrix}\right)$. Then, $Y_0 | Y_1 = y$ is Gaussian with*

$$\mathbb{E}[Y_0 | Y_1 = y] = \mu_0 + \Sigma_{01} \Sigma_{11}^{-1}(y - \mu_1), \text{ and } \text{Cov}(Y_0 | Y_1 = y) = \Sigma_{00} - \Sigma_{01} \Sigma_{11}^{-1} \Sigma_{10}. \qquad \text{(D.4)}$$

**Definition 4** (Infinitesimal generator of a stochastic process). For a sufficiently regular time-dependent function $\phi(t,x) \in \mathbb{R}^+ \times \mathbb{R}^n \to \mathbb{R}$, the infinitesimal generator of a stochastic process $X_t$ for $\phi(t,x)$ can be defined as

$$\mathcal{A}_t\phi(t,x) = \lim_{h \to 0} \frac{\mathbb{E}[\phi(t+h, X_{t+h})|X_t = x] - \phi(t,x)}{h}. \tag{D.5}$$

For an Itô process defined as the solution to the SDE

$$\mathrm{d}X_t = f(t, X_t)\,\mathrm{d}t + g(t, X_t)\,\mathrm{d}W_t, \tag{D.6}$$

with $f(t,x), g(t,x): \mathbb{R}^+ \times \mathbb{R}^n \to \mathbb{R}^n$, the generator is given as

$$\mathcal{A}_t\phi(t,x) = \partial_t\phi(t,x) + f(t,x)^\top\nabla_x\phi(t,x) + \frac{1}{2}\operatorname{Tr}[g(t,x)g(t,x)^\top\Delta\phi(t,x)] \tag{D.7}$$

where $\Delta := \nabla_x^2$ is the Euclidean Laplacian operator.

## D.2 OUTLINE OF THE PROOF

Our proofs follow the idea from Bunne et al. [2023, Theorem 3].

**For Theorem 6.** We follow the following two steps:

1. We first solve the associated *static* G$\mathcal{T}$SBP. Specifically, we formulate the equivalent E-OT problem, which has the transport cost dependent on the transition kernel of the $\mathcal{T}$SDE. By introducing new variables, we can convert this involved transport cost to a quadratic cost over new variables, thus, converting the G$\mathcal{T}$SBP$_{\text{static}}$ to a classical Gaussian E-OT. Based on the existing results [Janati et al., 2020; Mallasto et al., 2022], we can then obtain the optimal coupling over the transformed variables. The coupling over the original variables can be recovered via an inverse transform.
2. From Theorem A.4, we can obtain the solution of G$\mathcal{T}$SBP based on that the optimal $\mathbb{P}$ is in the reciprocal class of $\mathbb{Q}_\mathcal{T}$, specifically, by composing the static solution with the $\mathbb{Q}_\mathcal{T}$-bridge. This is an optimality condition obtained from the reduction to the static problem. From that, we know that the solution $\mathbb{P}$ is a Markov Gaussian process and shares the same bridge as $\mathbb{Q}_\mathcal{T}$ [cf. Proposition A.5]. This further allows us to characterize the mean and covariance of the time-marginal.

**For Theorem 7.** Let $\mathbb{P} \in \mathcal{P}(\Omega)$ be a finite-energy diffusion [Föllmer & Wakolbinger, 1986]; that is, under $\mathbb{P}$, the canonical process $X$ has a (forward) Itô differential. Furthermore, since $\mathbb{P}$ is in the reciprocal class of $\mathbb{Q}_\mathcal{T}$, it has the SDE representation in the class of (7) from the SOC formulation where the drift $f_\mathcal{T}(t, x : L)$ is to be determined. We then proceed the following two steps:

1. For the SDE (7) of the optimal process $X_t$, we first compute its *infinitesimal generator* [Protter, 2005] for a test function $\phi(t,x) \in \mathbb{R}^+ \times \mathbb{R}^n \to \mathbb{R}$ by definition using (D.5).
2. Second, we express the generator in terms of its given solution in (D.7) for the SDE (7)

$$\mathcal{A}_t\phi(t,x) = \partial_t\phi(t,x) + f_\mathcal{T}^\top\nabla_x\phi(t,x) + \frac{1}{2}g_t^2\nabla_x^2\phi(t,x). \tag{D.8}$$

By matching the generators computed in both ways, we then obtain the closed-form of the drift term.

## D.3 DETAILED PROOF OF THEOREM 6

### D.3.1 STEP 1: SOLVE G$\mathcal{T}$SBP$_{\text{STATIC}}$

First, recall that $Y_t|Y_0 = y_0$ is a Gaussian process with mean $m_t := \mathbb{E}(Y_t|y_0)$ and covariance $K_{tt}$ [cf. cond. mean and cond. cross cov], respectively. Thus, we have the transition probability density

$$\mathbb{Q}_{\mathcal{T}1|0}(y_1|y_0) \propto \exp\left(-\frac{1}{2}(y_1 - \Psi_1 y_0 - \xi_1)^\top K_{11}^{-1}(y_1 - \Psi_1 y_0 - \xi_1)\right)$$
$$\propto \exp\left(-\frac{1}{2}(y_1 - m_1)^\top K_{11}^{-1}(y_1 - m_1)\right). \tag{D.9}$$

By introducing the variables $\tilde{Y}_0 = K_{11}^{-\frac{1}{2}}(\Psi_1 Y_0 + \xi_1)$ and $\tilde{Y}_1 = K_{11}^{-\frac{1}{2}}Y_1$, we have

$$\mathbb{Q}_{\mathcal{T}1|0}(y_1|y_0) \propto \exp\left(-\frac{1}{2}\|\tilde{y}_1 - \tilde{y}_0\|^2\right). \tag{D.10}$$

Furthermore, if the joint distribution $\mathbb{P}_{01}$ has marginals $Y_0 \sim \nu_0$ and $Y_1 \sim \nu_1$, then after the change of variables ($Y_0 \to \tilde{Y}_0$ and $Y_1 \to \tilde{Y}_1$), it gives rise to a joint distribution $\tilde{\mathbb{P}}_{01}$ with marginals $\tilde{Y}_0 \sim \tilde{\nu}_0 = \mathcal{N}(\tilde{\mu}_0, \tilde{\Sigma}_0)$ and $\tilde{Y}_1 \sim \tilde{\nu}_1 = \mathcal{N}(\tilde{\mu}_1, \tilde{\Sigma}_1)$, where

$$
\begin{aligned}
\tilde{\mu}_0 &= K_{11}^{-\frac{1}{2}}(\Psi_1 \mu_0 + \xi_1), & \tilde{\Sigma}_0 &= K_{11}^{-\frac{1}{2}} \Psi_1 \Sigma_0 \Psi_1^\top K_{11}^{-\frac{1}{2}}, \\
\tilde{\mu}_1 &= K_{11}^{-\frac{1}{2}} \mu_1, & \tilde{\Sigma}_1 &= K_{11}^{-\frac{1}{2}} \Sigma_1 K_{11}^{-\frac{1}{2}}.
\end{aligned}
\tag{D.11}
$$

That is, there is an one-to-one correspondence between $\mathbb{P}_{01}$ and $\tilde{\mathbb{P}}_{01}$. This allows us to expand the objective of $\mathsf{G}\mathcal{T}\mathrm{SBP}_{\mathrm{static}}$ in terms of *minimization* as follows

$$
\begin{aligned}
D_{\mathsf{KL}}(\mathbb{P}_{01} \| \mathbb{Q}_{\mathcal{T}01}) &= \int_{\mathbb{R}^n \times \mathbb{R}^n} \mathbb{P}_{01}(y_0, y_1) \log\left(\frac{\mathbb{P}_{01}(y_0, y_1)}{\mathbb{Q}_{\mathcal{T}01}(y_0, y_1)}\right) \mathrm{d}y_0\, \mathrm{d}y_1 \\
&= -\int \log(\mathbb{Q}_{\mathcal{T}01}(y_0, y_1))\, \mathrm{d}\mathbb{P}_{01}(y_0, y_1) + \int \log(\mathbb{P}_{01})\, \mathrm{d}\mathbb{P}_{01} \\
&= \frac{1}{2}\int \|\tilde{y}_1 - \tilde{y}_0\|^2\, \mathrm{d}\mathbb{P}_{01}(y_0, y_1) + \int \log(\mathbb{P}_{01})\, \mathrm{d}\mathbb{P}_{01} + \text{const. 1} \qquad (\star) \\
&= \frac{1}{2}\int \|\tilde{y}_1 - \tilde{y}_0\|^2\, \mathrm{d}\tilde{\mathbb{P}}_{01}(\tilde{y}_0, \tilde{y}_1) + \int \log(\tilde{\mathbb{P}}_{01})\, \mathrm{d}\tilde{\mathbb{P}}_{01} + \text{const. 2} \\
&\equiv D_{\mathsf{KL}}(\tilde{\mathbb{P}}_{01} \| \mathbb{Q}_{\mathcal{T}01})
\end{aligned}
$$

where the second last step results from $\int \log(\tilde{\mathbb{P}}_{01})\, \mathrm{d}\tilde{\mathbb{P}}_{01} = \int \log(\mathbb{P}_{01})\, \mathrm{d}\mathbb{P}_{01} + \text{const.}$. To obtain $(\star)$, we notice that $\mathbb{Q}_{\mathcal{T}01}(y_0, y_1) = \mathbb{Q}_{\mathcal{T}1|0}(y_1|y_0)\mathbb{Q}_{\mathcal{T}0}(y_0)$, and we have

$$
\begin{aligned}
&-\int \log(\mathbb{Q}_{\mathcal{T}01}(y_0, y_1))\, \mathrm{d}\mathbb{P}_{01}(y_0, y_1) \\
&= -\int \log(\mathbb{Q}_{\mathcal{T}1|0}(y_1|y_0))\, \mathrm{d}\mathbb{P}_{01}(y_0, y_1) - \int \log(\mathbb{Q}_{\mathcal{T}0}(y_0))\, \mathrm{d}\mathbb{P}_{01}(y_0, y_1) \\
&= -\int \log(\mathbb{Q}_{\mathcal{T}1|0}(y_1|y_0))\, \mathrm{d}\mathbb{P}_{01}(y_0, y_1) - \int \log \mathbb{Q}_{\mathcal{T}0}(y_0)\, \mathrm{d}\mathbb{P}_0(y_0)
\end{aligned}
$$

where the last equality holds since we can remove the dependence on $y_1$ in the second term by integrating over $y_1$, thus, appearing as a constant in the optimization over $\mathbb{P}_{01}$. Moreover, the expression $(\star)$ is in fact the equivalent *E-OT* associated to the $\mathcal{T}\mathrm{SBP}$

$$
\min_{\mathbb{P}_{01}} \frac{1}{2}\int \|y_1 - \Psi_1 y_0 - \xi_1\|^2_{K_{11}^{-1}}\, \mathrm{d}\mathbb{P}_{01}(y_0, y_1) + \int \log(\mathbb{P}_{01})\, \mathrm{d}\mathbb{P}_{01}. \tag{$\mathcal{T}$E-OT}
$$

Note that by definition we have $D_{\mathsf{KL}}(\mathbb{P}_{01} \| \nu_0 \otimes \nu_1) = \int \log(\mathbb{P}_{01})\, \mathrm{d}\mathbb{P}_{01} - \int \log(\nu_0 \otimes \nu_1)\, \mathrm{d}\mathbb{P}_{01} = \int \log(\mathbb{P}_{01})\, \mathrm{d}\mathbb{P}_{01} - \int \log \nu_0\, \mathrm{d}\nu_0 - \int \log \nu_1\, \mathrm{d}\nu_1$ where the last two terms are constants.

Thus, solving $\mathsf{G}\mathcal{T}\mathrm{SBP}_{\mathrm{static}}$ is equivalent to solving the following problem

$$
\min_{\tilde{\mathbb{P}}_{01} \in \mathcal{P}(\mathbb{R}^n \times \mathbb{R}^n)} D_{\mathsf{KL}}(\tilde{\mathbb{P}}_{01} \| \mathbb{Q}_{\mathcal{T}01}) \equiv \int \frac{1}{2}\|\tilde{y}_1 - \tilde{y}_0\|^2\, \mathrm{d}\tilde{\mathbb{P}}_{01}(\tilde{y}_0, \tilde{y}_1) + \int \log(\tilde{\mathbb{P}}_{01})\, \mathrm{d}\tilde{\mathbb{P}}_{01} \tag{D.12}
$$

with $\tilde{\mathbb{P}}_0 = \tilde{\nu}_0$ and $\tilde{\mathbb{P}}_1 = \tilde{\nu}_1$. This is a classical static Gaussian E-OT between $\tilde{\nu}_0$ and $\tilde{\nu}_1$ with $\sigma = 1$. The closed-form solution is given by the joint Gaussian [cf. Theorem A.1]

$$
\tilde{\mathbb{P}}_{01}^\star = \mathcal{N}\left(\begin{bmatrix} \tilde{\mu}_0 \\ \tilde{\mu}_1 \end{bmatrix}, \begin{bmatrix} \tilde{\Sigma}_0 & \tilde{C} \\ \tilde{C}^\top & \tilde{\Sigma}_1 \end{bmatrix}\right) \tag{D.13}
$$

where

$$
\tilde{C} = \frac{1}{2}(\tilde{\Sigma}_0^{1/2} \tilde{D} \tilde{\Sigma}_0^{-1/2} - I), \quad \tilde{D} = (4\tilde{\Sigma}_0^{1/2} \tilde{\Sigma}_1 \tilde{\Sigma}_0^{1/2} + I)^{1/2}. \tag{D.14}
$$

Finally, via the inverse transforms $Y_0 = \Psi_1^{-1}(K_{11}^{\frac{1}{2}} \tilde{y}_0 - \xi_1)$ and $Y_1 = K_{11}^{\frac{1}{2}} \tilde{y}_1$, we can obtain the solution to the original problem $\mathsf{G}\mathcal{T}\mathrm{SBP}_{\mathrm{static}}$ as

$$
\mathbb{P}_{01}^\star \sim \mathcal{N}\left(\begin{bmatrix} \mu_0 \\ \mu_1 \end{bmatrix}, \begin{bmatrix} \Sigma_0 & C \\ C^\top & \Sigma_1 \end{bmatrix}\right) \tag{$\mathsf{G}\mathcal{T}\mathrm{SB}_{\mathrm{static}}$}
$$

where $C = \Psi_1^{-1} K_{11}^{\frac{1}{2}} \tilde{C} K_{11}^{\frac{1}{2}}$.

### D.3.2 STEP 2: FROM STATIC TO DYNAMIC VIA DISINTEGRATION FORMULA

From Theorem A.4, we know that the solution $\mathbb{P}$ to G$\mathcal{T}$SBP shares its bridges with the reference $\mathbb{Q}_{\mathcal{T}}$. We denote by $\mathbb{Q}_{\mathcal{T}}^{y_0 y_1}$ the process $Y$ conditioning on $Y_0 = y_0$ and $Y_1 = y_1$ under $\mathbb{Q}_{\mathcal{T}}$, i.e., $Y|y_0, y_1 \sim \mathbb{Q}_{\mathcal{T}}^{y_0 y_1} = \mathbb{Q}_{\mathcal{T}}[Y_0 = y_0, Y_1 = y_1]$. It is the bridge of $\mathbb{Q}_{\mathcal{T}}$, following

$$\mathbb{Q}_{\mathcal{T}}(\cdot) = \int_{\mathbb{R}^n \times \mathbb{R}^n} \mathbb{Q}_{\mathcal{T}}^{y_0 y_1}(\cdot) \mathbb{Q}_{\mathcal{T}01}(\mathrm{d}y_0 \, \mathrm{d}y_1). \tag{D.15}$$

In the classical case of Brownian motion $Y = W \sim \mathbb{Q}_W$, $\mathbb{Q}_W^{y_0 y_1}$ is often referred to as the *Brownian bridge*. Here, we aim to first find the $\mathbb{Q}_{\mathcal{T}}^{y_0 y_1}$-bridge, and then construct the optimal solution $\mathbb{P}^\star$ by composing the static solution $\mathbb{P}_{01}^\star$ with the $\mathbb{Q}_{\mathcal{T}}^{y_0 y_1}$-bridge [cf. (A.4) in Theorem A.4].

From the transition kernel in Lemma 4, we have the conditional distributions $Y_t|y_0 \sim \mathcal{N}(m_t, K_{tt})$ and $Y_1|y_0 \sim \mathcal{N}(m_1, K_{11})$. Thus, the joint distribution of $Y_t$ and $Y_1$ given $y_0$ follows

$$Y_t, Y_1|y_0 \sim \mathcal{N}\left(\begin{bmatrix} m_t \\ m_1 \end{bmatrix}, \begin{bmatrix} K_{tt} & K_{t1} \\ K_{1t} & K_{11} \end{bmatrix}\right). \tag{D.16}$$

Applying Lemma D.3, we know that $Y_t|y_0, Y_1 = y_1$ is Gaussian with mean

$$\begin{aligned}
\mathbb{E}(Y_t|y_0, Y_1 = y_1) &= m_t + K_{t1}K_{11}^{-1}(y_1 - m_1) \\
&= \Psi_t y_0 + \xi_t + K_{t1}K_{11}^{-1}(y_1 - \Psi_1 y_0 - \xi_1) \\
&= (\Psi_t - K_{t1}K_{11}^{-1}\Psi_1)y_0 + K_{t1}K_{11}^{-1}y_1 + \xi_t - K_{t1}K_{11}^{-1}\xi_1 \\
&\triangleq \bar{R}_t y_0 + R_t y_1 + \xi_t - R_t \xi_1
\end{aligned} \tag{D.17}$$

where we recall the definitions of $R_t$ and $\bar{R}_t$ in (8), and covariance

$$\Gamma_t := \mathrm{Cov}(Y_t|Y_0 = y_0, Y_1 = y_1) = K_{tt} - K_{t1}K_{11}^{-1}K_{1t}. \tag{D.18}$$

Since a Gaussian process is completely determined by its mean and covariance, we have

$$Y_t|Y_0, Y_1 \stackrel{\text{law}}{=} \bar{R}_t Y_0 + R_t Y_1 + \xi_t - R_t \xi_1 + \Gamma_t Z \sim \mathbb{Q}_{\mathcal{T}t}^{y_0 y_1} \tag{D.19}$$

where $Z \sim \mathcal{N}(0, I)$ is independent of $Y_t$. Now, the Disintegration of Measures and Theorem A.4 allow us to construct the solution to G$\mathcal{T}$SBP by first generating $(X_0, X_1) \sim \mathbb{P}_{01}^\star$ in G$\mathcal{T}$SB$_{\text{static}}$, then connecting $X_0$ and $X_1$ using the $\mathbb{Q}_{\mathcal{T}}^{y_0 y_1}$-bridge. This is equivalent to, for $X_0 \sim \nu_0, X_1 \sim \nu_1$ and $Z \sim \mathcal{N}(0, I), Z \perp (X_0, X_1)$, building a process as

$$X_t \stackrel{\text{law}}{=} \bar{R}_t X_0 + R_t X_1 + \xi_t - R_t \xi_1 + \Gamma_t Z \sim \mathbb{P}_t^\star, \tag{D.20}$$

which in fact is a *stochastic interpolant* for stochastic processes over topological domains, generalizing the same notion in Euclidean domains in Albergo et al. [2024, Definition 1]. *Note that since $\mathbb{Q}_{\mathcal{T}}$ is a stochastic process following an Itô SDE, which is a Markov process, the solution $\mathbb{P}$ is also a Markov process* [cf. Proposition A.5]. Finally, we obtain the mean and covariance of the time-marginal $X_t$ as

$$\begin{aligned}
\mu_t &= \bar{R}_t \mu_0 + R_t \mu_1 + \xi_t - R_t \xi_1, \\
\Sigma_t &= \bar{R}_t \Sigma_0 \bar{R}_t^\top + R_t \Sigma_1 R_t^\top + \bar{R}_t C R_t^\top + R_t C^\top \bar{R}_t^\top + K_{tt} - K_{t1}K_{11}^{-1}K_{1t}.
\end{aligned} \tag{D.21}$$

This concludes the proofs of Theorem 6 and Corollary D.1.

### D.4 DETAILED PROOF OF THEOREM 7

### D.4.1 STEP 1: COMPUTE THE INFINITESIMAL GENERATOR OF $X_t$ BY DEFINITION

For some time-varying function $\phi(t, x)$, by definition, the infinitesimal generator of $X_t$ is given by (D.5). Since $X_t$ is a Gaussian process, we could express the conditional expectation using Lemma D.3. As we are only interested in the terms that are of order $O(h)$, we then ignore the higher-order terms. First, we compute the first-order approximation of $\Sigma_t$ in (D.1)

$$\begin{aligned}
\dot{\Sigma}_t &= \dot{\bar{R}}_t \Sigma_0 \bar{R}_t^\top + \bar{R}_t \Sigma_0 \dot{\bar{R}}_t^\top + \dot{R}_t \Sigma_1 R_t^\top + R_t \Sigma_1 \dot{R}_t^\top \\
&\quad + \dot{\bar{R}}_t C R_t^\top + \bar{R}_t C \dot{R}_t^\top + \dot{R}_t C^\top \bar{R}_t^\top + R_t C^\top \dot{\bar{R}}_t^\top \\
&\quad + \partial_t K_{tt} - (\partial_t K_{t1})K_{11}^{-1}K_{1t} - (K_{t1}K_{11}^{-1})\partial_t K_{1t} \\
&\triangleq (P_t^\top + P_t) - (Q_t + Q_t^\top) + \partial_t K_{tt} - (\partial_t K_{t1})K_{11}^{-1}K_{1t} - (K_{t1}K_{11}^{-1})\partial_t K_{1t}
\end{aligned} \tag{D.22}$$

where at the last equality we recall the definitions of $P_t$ and $Q_t$ in (8). Next, denote by $\Sigma_{t,t+h}$ the covariance process of $X_t$ evaluated at $t$ and $t+h$. We can estimate $\Sigma_{t,t+h}$ up to the first order of $o(h)$ as

$$
\begin{aligned}
\Sigma_{t,t+h} &:= \mathbb{E}[(X_t - \mu_t)(X_{t+h} - \mu_{t+h})^\top] \\
&= \bar{R}_t \Sigma_0 \bar{R}_{t+h}^\top + R_t \Sigma_1 R_{t+h}^\top + \bar{R}_t C R_{t+h}^\top + R_t C^\top \bar{R}_{t+h}^\top + K_{t,t+h} - K_{t1} K_{11}^{-1} K_{1,t+h} \\
&= \Sigma_t + \bar{R}_t \Sigma_0 (\bar{R}_{t+h} - \bar{R}_t)^\top + R_t \Sigma_1 (R_{t+h} - R_t)^\top + \bar{R}_t C (R_{t+h} - R_t)^\top \\
&\quad + R_t C^\top (\bar{R}_{t+h} - \bar{R}_t)^\top + (K_{t,t+h} - K_{tt}) - K_{t1} K_{11}^{-1}(K_{1,t+h} - K_{1t}) \\
&\overset{(a)}{=} \Sigma_t + h(\bar{R}_t \Sigma_0 \dot{\bar{R}}_t^\top + R_t \Sigma_1 \dot{R}_t^\top + \bar{R}_t C \dot{R}_t^\top + R_t C^\top \dot{\bar{R}}_t^\top) + o(h) \\
&\quad + (K_{t,t+h} - K_{tt}) - K_{t1} K_{11}^{-1}(K_{1,t+h} - K_{1t}) \\
&\overset{(b)}{=} \Sigma_t + h(P_t - Q_t^\top + \partial_{t_2} K_{t_1,t_2}|_{t_1=t,t_2=t} - K_{t1} K_{11}^{-1} \partial_{t_2} K_{t_1,t_2}|_{t_1=1,t_2=t}) + o(h)
\end{aligned}
$$
$$\text{(D.23)}$$

where we obtain $(a)$ by plugging in $\lim_{h \to 0} \frac{1}{h}(R_{t+h} - R_t) = \dot{R}_t$ and $\lim_{h \to 0} \frac{1}{h}(\bar{R}_{t+h} - \bar{R}_t)$; and likewise, we obtain $(b)$ by recognizing the definitions of $P_t$ and $Q_t^\top$ in (8) and using the partial derivatives

$$
\begin{aligned}
\lim_{h \to 0} \frac{1}{h}(K_{t,t+h} - K_{t,t}) &= \partial_{t_2} K_{t_1,t_2}|_{t_1=t,t_2=t}, \quad t_1 \leq t_2 \\
\lim_{h \to 0} \frac{1}{h}(K_{1,t+h} - K_{1,t}) &= \partial_{t_2} K_{t_1,t_2}|_{t_1=1,t_2=t} = \partial_t K_{1t}, \quad t_1 > t_2.
\end{aligned}
$$
$$\text{(D.24)}$$

Following the similar procedure, we can obtain a first-order approximation of $\Sigma_{t+h,t}$ as

$$
\begin{aligned}
\Sigma_{t+h,t} &:= \mathbb{E}[(X_{t+h} - \mu_{t+h})(X_t - \mu_t)^\top] \\
&= \Sigma_t + h(\dot{\bar{R}}_t \Sigma_0 \bar{R}_t^\top + \dot{R}_t \Sigma_1 R_t^\top + \dot{\bar{R}}_t C R_t^\top + \dot{R}_t C \bar{R}_t^\top) + o(h) \\
&\quad + (K_{t+h,t} - K_{tt}) - (K_{t+h,1} - K_{t1}) K_{11}^{-1} K_{1t} \\
&= \Sigma_t + h(P_t^\top - Q_t + \partial_{t_1} K_{t_1,t_2}|_{t_1=t,t_2=t} - \partial_{t_1} K_{t_1,t_2}|_{t_1=t,t_2=1} K_{11}^{-1} K_{1t}) + o(h),
\end{aligned}
$$
$$\text{(D.25)}$$

where the partial derivatives should be understood as

$$
\begin{aligned}
\lim_{h \to 0} \frac{1}{h}(K_{t+h,t} - K_{t,t}) &= \partial_{t_1} K_{t_1,t_2}|_{t_1=t,t_2=t}, \quad t_1 > t_2 \\
\lim_{h \to 0} \frac{1}{h}(K_{t+h,1} - K_{t,1}) &= \partial_{t_1} K_{t_1,t_2}|_{t_1=t,t_2=1} = \partial_t K_{t1}, \quad t_1 \leq t_2.
\end{aligned}
$$
$$\text{(D.26)}$$

Since Eqs. (D.22), (D.23) and (D.25) are all involved with the partial derivatives of $K_{t_1,t_2}$, we can compute them by the closed-form of the transition matrix as

$$
\begin{aligned}
\partial_{t_1} K_{t_1,t_2} &= \partial_{t_1} \left\{ \Psi_{t_1} \left[ \int_0^{t_1} g_s^2 \Psi_s^{-2} \, ds \right] \Psi_{t_2}^\top \right\}, \quad \text{for } t_1 \leq t_2 \\
&\overset{(a)}{=} \Psi_{t_1} H_{t_1} \left[ \int_0^{t_1} g_s^2 \Psi_s^{-2} \, ds \right] \Psi_{t_2}^\top + g_{t_1}^2 \Psi_{t_1}^{-1} \Psi_{t_2}^\top \\
&\overset{(b)}{=} H_{t_1} K_{t_1,t_2} + g_{t_1}^2 \Psi_{t_1}^{-1} \Psi_{t_2}^\top, \quad \text{for } t_1 \leq t_2,
\end{aligned}
$$
$$\text{(D.27)}$$

where we use the symmetry of $\Psi_t$. At $(a)$ we use

$$
\partial_t \Psi_t = \partial_t \exp\left( \int_0^t H_s \, ds \right) = \exp\left( \int_0^t H_s \, ds \right) H_t = \Psi_t H_t,
$$
$$\text{(D.28)}$$

and at $(b)$ we use the commutativity of $H_t$ and $\Psi_t$ [cf. Lemmas B.1 and B.5]. Similarly, we have

$$
\begin{aligned}
\partial_{t_2} K_{t_1,t_2} &= \partial_{t_2} \left\{ \Psi_{t_1} \left[ \int_0^{t_1} g_s^2 \Psi_s^{-2} \, ds \right] \Psi_{t_2}^\top \right\} = K_{t_1,t_2} H_{t_2}^\top, \quad \text{for } t_1 \leq t_2 \\
\partial_{t_1} K_{t_1,t_2} &= \partial_{t_1} \left\{ \Psi_{t_1} \left[ \int_0^{t_2} g_s^2 \Psi_s^{-2} \, ds \right] \Psi_{t_2}^\top \right\} = H_{t_1} K_{t_1,t_2}, \quad \text{for } t_1 > t_2 \\
\partial_{t_2} K_{t_1,t_2} &= \partial_{t_2} \left\{ \Psi_{t_1} \left[ \int_0^{t_2} g_s^2 \Psi_s^{-2} \, ds \right] \Psi_{t_2}^\top \right\} = g_{t_2}^2 \Psi_{t_1} \Psi_{t_2}^{-1} + K_{t_1,t_2} H_{t_2}^\top, \quad \text{for } t_1 > t_2.
\end{aligned}
$$
$$\text{(D.29)}$$

We notice that $(\partial_t K_{1t})^\top = \partial_t K_{t1} = g_t^2 \Psi_t^{-1} \Psi_1^\top + H_t K_{t1} K_{11}^{-1} K_{1t}$. Now, by introducing in (D.23) the variable

$$
\begin{aligned}
S &\triangleq P_t - Q_t^\top + \partial_{t_2} K_{t_1,t_2}|_{t_1=t,t_2=t} - K_{t1} K_{11}^{-1} \partial_{t_2} K_{t_1,t_2}|_{t_1=1,t_2=t} \\
&= P_t - Q_t^\top + K_{tt} H_t^\top - K_{t1} K_{11}^{-1} (g_t^2 \Psi_1 \Psi_t^{-1} + K_{1t} H_t^\top) \\
&= P_t - Q_t^\top + K_{tt} H_t^\top - K_{t1} K_{11}^{-1} (\partial_t K_{1t}),
\end{aligned} \tag{D.30}
$$

we can then express the covariance process as

$$
\Sigma_{t,t+h} = \Sigma_t + h S_t + o(h), \tag{D.31}
$$

and

$$
\begin{aligned}
\Sigma_{t+h,t} &= \Sigma_t + h(P_t^\top - Q_t + H_t^\top K_{tt} - (g_t^2 \Psi_t^{-1} \Psi_1^\top + H_t K_{t1}) K_{11}^{-1} K_{1t}) + o(h) \\
&= \Sigma_t + h(P_t^\top - Q_t + H_t^\top K_{tt} - (\partial_t K_{t1}) K_{11}^{-1} K_{1t}) + o(h) \\
&= \Sigma_t + h S_t^\top + o(h).
\end{aligned} \tag{D.32}
$$

Lastly, using Lemma D.3, we see tha variable $X_{t+h}|X_t = x$ is a Gaussian process with mean

$$
\begin{aligned}
\check{\mu}_{t+h} := &= \mu_{t+h} + \Sigma_{t+h,t} \Sigma_t^{-1}(x - \mu_t) \\
&\overset{(a)}{=} \mu_t + h\dot{\mu}_t + (\Sigma_t + h S_t^\top) \Sigma^{-1}(x - \mu_t) + o(h) \\
&= \mu_t + h\dot{\mu}_t + (I + h S_t^\top \Sigma_t^{-1})(x - \mu_t) + o(h) \\
&= x + h(\dot{\mu}_t + S_t^\top \Sigma_t^{-1}(x - \mu_t)) + o(h)
\end{aligned} \tag{D.33}
$$

where in $(a)$ we used $\Sigma_t = \Sigma_t^\top$, and covariance

$$
\begin{aligned}
\check{\Sigma}_{t+h} &= \Sigma_{t+h} - \Sigma_{t+h,t} \Sigma_t^{-1} \Sigma_{t,t+h} \\
&= \Sigma_t + h\dot{\Sigma}_t - (\Sigma_t + h S_t^\top) \Sigma_t^{-1} (\Sigma_t + h S_t) + o(h) \\
&= \Sigma_t + h\dot{\Sigma}_t - (\Sigma_t + h S_t^\top + h S_t) + o(h) \\
&\overset{(b)}{=} h(\dot{\Sigma}_t - S_t - S_t^\top) + o(h).
\end{aligned} \tag{D.34}
$$

By seeing $K_{tt}$ as a matrix function of $t$, we have

$$
\partial_t K_{tt} = \partial_t \left\{ \Psi_t \left[ \int_0^t g_s^2 \Psi_s^{-2} \, ds \right] \Psi_t^\top \right\} = H_t K_{tt} + g_t^2 I + K_{tt} H_t^\top. \tag{D.35}
$$

This reduces (D.22) to

$$
\dot{\Sigma}_t = (P_t^\top + P_t) - (Q_t + Q_t^\top) + \partial_t K_{tt} - (\partial_t K_{t1}) K_{11}^{-1} K_{1t} - (K_{t1} K_{11}^{-1}) \partial_t K_{1t} \tag{D.36}
$$

where the last two items appear in $S_t^\top$ and $S_t$, respectively. Thus, we obtain

$$
\begin{aligned}
\check{\Sigma}_{t+h} &= h(\dot{\Sigma}_t - S_t - S_t^\top) + o(h) \\
&= h \Big\{ \big[ (P_t^\top + P_t) - (Q_t + Q_t^\top) + \partial_t K_{tt} - (\partial_t K_{t1}) K_{11}^{-1} K_{1t} - (K_{t1} K_{11}^{-1}) \partial_t K_{1t} \big] \\
&\quad - \big[ P_t - Q_t^\top + K_{tt} H_t^\top - K_{t1} K_{11}^{-1} (\partial_t K_{1t}) \big] \\
&\quad - \big[ P_t^\top - Q_t + H_t^\top K_{tt} - (\partial_t K_{t1}) K_{11}^{-1} K_{1t} \big] \Big\} + o(h) \\
&= h g_t^2 I + o(h).
\end{aligned} \tag{D.37}
$$

We can now compute $\mathbb{E}[\phi(t+h, X_{t+h})|X_t = x]$ as follows

$$
\begin{aligned}
&\mathbb{E}[\phi(t+h, X_{t+h})|X_t = x] \\
&= (2\pi)^{\frac{d}{2}} (\det \check{\Sigma}_{t+h})^{-\frac{1}{2}} \int_{\mathbb{R}^n} \phi(t+h, x') \cdot \exp\Big( -\frac{1}{2} (x' - \check{\mu}_{t+h})^\top \check{\Sigma}_{t+h}^{-1} (x' - \check{\mu}_{t+h}) \Big) \, dx' \\
&\overset{(a)}{=} (2\pi)^{\frac{d}{2}} (\det \check{\Sigma}_{t+h})^{-\frac{1}{2}} \int_{\mathbb{R}^n} \phi(t+h, \tilde{x} + \check{\mu}_{t+h}) \cdot \exp\Big( -\frac{1}{2} \tilde{x}^\top \check{\Sigma}_{t+h}^{-1} \tilde{x} \Big) \, d\tilde{x}
\end{aligned} \tag{D.38}
$$

where in $(a)$ we apply a change-of-variable $\tilde{x} := x' - \check{\mu}_{t+h}$. We further apply Lemma D.2 and arrive at

$$
\begin{aligned}
\mathbb{E}[\phi(t+h, X_{t+h})|X_t = x] &= \exp\Big(\frac{1}{2}\partial_{\tilde{x}}^\top \check{\Sigma}_{t+h}\partial_{\tilde{x}}\Big)\phi(t+h, \tilde{x}+\check{\mu}_{t+h})\Big|_{\tilde{x}=0} \\
&\overset{(a)}{=} \Big(I + \frac{1}{2}\partial_{\tilde{x}}^\top \check{\Sigma}_{t+h}\partial_{\tilde{x}} + o(\text{h.o.t.})\Big)\phi(t+h, \tilde{x}+\check{\mu}_{t+h})|_{\tilde{x}=0} \quad \text{(D.39)} \\
&\overset{(b)}{=} \phi(t+h, \check{\mu}_{t+h}) + \frac{1}{2}hg_t^2\Delta\phi(t+h, \check{\mu}_{t+h}) + o(h),
\end{aligned}
$$

where we expand the power series of $\exp(\frac{1}{2}\partial_{\tilde{x}}^\top \check{\Sigma}_{t+h}\partial_{\tilde{x}})$ and ignore the higher-order-terms in $(a)$, and plug in (D.37) in $(b)$. Recalling $\check{\mu}_{t+h}$ in (D.33), we can expand the Taylor series of $\phi(t+h, \check{\mu}_{t+h})$ in the second variabel at $x$ as

$$
\begin{aligned}
\phi(t+h, \check{\mu}_{t+h}) &= \phi\big(t+h, x + h(\dot{\mu}_t + S_t^\top \Sigma_t^{-1}(x - \mu_t))\big) \\
&= \phi(t+h, x) + h\langle\nabla\phi(t+h, x), \dot{\mu}_t + S^\top \Sigma_t^{-1}(x - \mu_t)\rangle + o(h).
\end{aligned} \quad \text{(D.40)}
$$

Therefore, we have

$$
\begin{aligned}
&\mathbb{E}[\phi(t+h, X_{t+h})|X_t = x] = \\
&\phi(t+h, x) + h\langle\nabla\phi(t+h, x), \dot{\mu}_t + S^\top \Sigma_t^{-1}(x - \mu_t)\rangle + \frac{1}{2}hg_t^2\Delta\phi(t+h, x) + o(h).
\end{aligned} \quad \text{(D.41)}
$$

Now we can express the infinitesimal generator of $X_t$ as

$$
\begin{aligned}
&\lim_{h\to 0}\frac{\mathbb{E}[\phi(t+h, X_{t+h})|X_t = x] - \phi(t, x)}{h} \\
&= \lim_{h\to 0}\frac{u(t+h, x) - u(t, x)}{h} + \langle\nabla\phi(t, x), \dot{\mu}_t + S^\top \Sigma_t^{-1}(x - \mu_t)\rangle + \frac{1}{2}g_t^2\Delta\phi(t, x) \quad \text{(D.42)} \\
&= \partial_t u(t, x) + \langle\nabla\phi(t, x), \dot{\mu}_t + S^\top \Sigma_t^{-1}(x - \mu_t)\rangle + \frac{1}{2}g_t^2\Delta\phi(t, x).
\end{aligned}
$$

### D.4.2 STEP 2: MATCH THE SOLUTION OF GENERATOR FOR AN ITÔ SDE

From Léonard [2014]; Caluya & Halder [2021], we search for the optimal solution to $\mathcal{T}\text{SBP}$ within the class of stochastic processes following an SDE:

$$
dX_t = f_{\mathcal{T}}(t, X_t)\,dt + g_t\,dW_t. \quad \text{(D.43)}
$$

Recalling the solution of an infinitesimal generator in (D.7) for this SDE

$$
\mathcal{A}_t\phi(t, x) = \partial_t\phi(t, x) + f_{\mathcal{T}}(t, x)^\top\nabla_x\phi(t, x) + \frac{1}{2}g_t^2\Delta\phi(t, x), \quad \text{(D.44)}
$$

we then match it with the generator obtained by definition in (D.42). We observe that the two are equivalent if we set

$$
f_{\mathcal{T}}(t, x) = \dot{\mu}_t + S^\top \Sigma_t^{-1}(x - \mu_t). \quad \text{(D.45)}
$$

This concludes the proof of Theorem 7.

### D.5 CONDITIONAL DISTRIBUTION OF $X_t|X_0$

**Corollary D.5** (Conditional distribution of $X_t|X_0$)**.** *Let $X_t$ be the stochastic process associated to the solution $\mathbb{P}$ to $G\mathcal{T}\text{SBP}$. Given an initial sample $x_0 \sim \nu_0$, the conditional distribution $\nu(X_t|X_0 = x_0) \sim \mathcal{N}(\mu_{t|0}, \Sigma_{t|0})$ is Gaussian with*

$$
\begin{aligned}
\mu_{t|0} &= \bar{R}_t x_0 + R_t\mu_1 + R_t C^\top \Sigma_0^{-1}(x_0 - \mu_0) + \xi_t - R_t\xi_1, \\
\Sigma_{t|0} &= R_t\Sigma_1 R_t^\top - R_t C^\top \Sigma_0^{-1} C R_t^\top + K_{tt} - K_{t1}K_{11}^{-1}K_{1t}.
\end{aligned} \quad \text{(D.46)}
$$

*Similarly, given a final sample $x_1$, the conditional distribution $\nu(X_t|X_1 = x_1) \sim \mathcal{N}(\mu_{t|1}, \Sigma_{t|1})$ is Gaussian with*

$$
\begin{aligned}
\mu_{t|1} &= R_t x_1 + \bar{R}_t\mu_0 + \bar{R}_t C\Sigma_1^{-1}(x_1 - \mu_1) + \xi_t - R_t\xi_1, \\
\Sigma_{t|1} &= \bar{R}_t\Sigma_0\bar{R}_t^\top - \bar{R}_t C\Sigma_1^{-1}C^\top \bar{R}_t^\top + K_{tt} - K_{t1}K_{11}^{-1}K_{1t}.
\end{aligned} \quad \text{(D.47)}
$$

*Proof.* First, recall the stochastic interpolant expression in (5) of the solution to G$\mathcal{T}$SBP and its mean $\mu_t$ and covariance $\Sigma_t$ in (D.1). Due to the Gaussian nature of the process, we can write the joint distribution of $X_t$ and $X_0$ as

$$\begin{bmatrix} X_t \\ X_0 \end{bmatrix} \sim \mathcal{N}\left( \begin{bmatrix} \mu_t \\ \mu_0 \end{bmatrix}, \begin{bmatrix} \Sigma_t & \Sigma_{t,0} \\ \Sigma_{0,t} & \Sigma_0 \end{bmatrix} \right) \tag{D.48}$$

where we work out the covariance between $X_t$ and $X_0$ below

$$\begin{aligned}
\Sigma_{t,0} &= \mathbb{E}[(X_t - \mu_t)(X_0 - \mu_0)^\top] \\
&= \bar{R}_t \Sigma_0 + R_t \mathrm{Cov}(X_1, X_0) + \mathrm{Cov}(\xi_t, X_0) - R_t \mathrm{Cov}(\xi_1, X_0) + \mathrm{Cov}(\zeta_t, X_0) \\
&= \bar{R}_t \Sigma_0 + R_t \mathrm{Cov}(X_1, X_0) && \text{(since } \xi_t \text{ is deterministic, } \zeta_t \perp X_0) \\
&= \bar{R}_t \Sigma_0 + R_t C^\top. && \text{(by } \mathbb{P}_{01}^* \text{ in G}\mathcal{T}\text{SB}_{\text{static}})
\end{aligned}$$

Based on Lemma D.3, we know that $X_t | X_0 = x_0$ is Gaussian with mean $\mu_{t|0}$ and covariance $\Sigma_{t|0}$ given by

$$\begin{aligned}
\mu_{t|0} &= \mu_t + \Sigma_{t,0} \Sigma_0^{-1}(x_0 - \mu_0) \\
\Sigma_{t|0} &= \Sigma_t - \Sigma_{t,0} \Sigma_0^{-1} \Sigma_{0,t}.
\end{aligned} \tag{D.49}$$

By substituting the expressions of $\mu_t$ and $\Sigma_t$ from (D.1) and canceling out terms, we complete the proof for $X_t | X_0 = x_0$. The proof for $X_t | X_1 = x_1$ follows similarly. □

# E  TOPOLOGICAL SB GENERATIVE MODELS

Here, we provide more details on the $\mathcal{T}$SB-based models. First, we give the likelihood of the model which allows for the training objective in (9), and the probability flow ODEs corresponding to the FB-$\mathcal{T}$SDEs in (3). Then, we discuss the variants of score-based and diffusion bridges models for topological signals, as well as their training objectives, with the goal of illustrating how $\mathcal{T}$SB-based models connect to these models.

## E.1  LIKELIHOOD TRAINING FOR TOPOLOGICAL SBP

The likelihood for the Euclidean SBP by Chen et al. [2022a] extends to the topological case.

**Corollary E.1** (Likelihood for $\mathcal{T}$SB models; Chen et al. [2022a]). *Given the optimal solution of $\mathcal{T}$SBP satisfying the FB-$\mathcal{T}$SDE system in (3), the log-likelihood of the $\mathcal{T}$SB model at an initial signal sample $x_0$ can be expressed as*

$$\mathcal{L}_{\mathcal{T}SB}(x_0) = \mathbb{E}[\log \nu_1(X_1)] - \int_0^1 \mathbb{E}\left[ \frac{1}{2}\|Z_t\|^2 + \frac{1}{2}\|\hat{Z}_t\|^2 + \nabla \cdot (g_t \hat{Z}_t - f_t) + \hat{Z}_t^\top Z_t \right] \mathrm{d}t \tag{E.1}$$

*where the expectation is taken over the forward SDE in (3) with the initial condition $X_0 = x_0$.*

**Corollary E.2** (Probability flow ODE for $\mathcal{T}$SB). *The following ODE characterizes the probability flow of the optimal processes of $\mathcal{T}$SB in (3)*

$$\mathrm{d}X_t = \left[ f_t + g_t Z_t - \frac{1}{2}g_t(Z_t + \hat{Z}_t) \right] \mathrm{d}t \tag{E.2}$$

*and we have that for all $t$, $\mathbb{P}_t = p_t^{(E.2)}$, i.e., the time marginal of the path measure $\mathbb{P}$ is equal to the probability flow $p_t$ of this ODE.*

This is a direct result from the probability flow for general SB [Chen et al., 2022a], which extends the probability flow for score-based models [Song et al., 2020b], and relates to the flow-based training.

## E.2  SCORE MATCHING FOR TOPOLOGICAL SIGNALS

As discussed in Section 5 and by Chen et al. [2022a], SB-based models generalize the score-based models [Song et al., 2020b]. Here, we provide a detailed derivation on how a score-based model can be built for topological signals, specifically, on the score matching objective, since there is no direct literature on this. First, we show in detail that the likelihood training based on (9b) returns a score

matching objective for topological signals when $Z_t = 0$ and the final $\nu_1$ is a simple Gaussian. The backward training objective in this case becomes

$$l(x_0; \hat{\theta}) = \int_0^1 \mathbb{E}_{X_t \sim (3a)}\left[\frac{1}{2}\|\hat{Z}_t^{\hat{\theta}}\|^2 + g_t \nabla \cdot \hat{Z}_t^{\hat{\theta}}\Big| X_0 = x_0\right] \mathrm{d}t$$

$$= \int_0^1 \mathbb{E}_{X_t \sim (3a)}\left[\frac{1}{2}g_t^2\|s_t(\hat{\theta})\|^2 + g_t^2 \nabla \cdot s_t(\hat{\theta})\Big| X_0 = x_0\right] \mathrm{d}t$$

where we introduce a score function $s_t(\hat{\theta})$ to approximate $\nabla \log p_{t|0}(X_t|X_0 = x_0)$, following $\hat{Z}_t^{\hat{\theta}} = g_t s_t(\hat{\theta})$. Here, $p_{t|0}$ is the transition kernel of the $\mathcal{T}$SDE [cf. Lemma 4]. By using the trace estimator Hutchinson [1989] to compute the divergence, i.e.,

$$\nabla \cdot s_t(\hat{\theta}) = \mathbb{E}_{u \sim \mathcal{N}(0,I)}[u^\top s_t(\hat{\theta})u], \tag{E.3}$$

and setting the weighting function $\lambda(t) := g_t^2$, we then obtain the *sliced score matching* objective [Song et al., 2020a;b, Eq. 19] for topological signals based on $\mathcal{T}$SDE, which has the form

$$\hat{\theta} = \arg\min \mathbb{E}_{t \sim \mathcal{U}(0,1)}\left\{\lambda(t)\mathbb{E}_{x_0}\mathbb{E}_{x_t}\mathbb{E}_{u \sim \mathcal{N}(0,I)}\left[\frac{1}{2}\|s_t(\hat{\theta})\|^2 + u^\top s_t(\hat{\theta})u\right]\right\}, \tag{E.4}$$

and is equivalent to $l(x_0; \hat{\theta})$. This does not require a closed-form solution for the true score function $\nabla \log p_{t|0}$. The associated FB-$\mathcal{T}$SDEs now become the forward-backward processes for the score-based models

$$\mathrm{d}X_t = f_t \, \mathrm{d}t + g_t \, \mathrm{d}W_t, \tag{E.5}$$

$$\mathrm{d}X_t = (f_t - g_t^2 s_t(\hat{\theta})) \, \mathrm{d}t + g_t \, \mathrm{d}W_t. \tag{E.6}$$

**Closed-form score matching** For $\mathcal{T}$SHeat$_{\mathrm{BM}}$ and $\mathcal{T}$SHeat$_{\mathrm{VE}}$, since we have their closed-form transition kernels in (2), we can use the direct score matching objective [Song et al., 2020b, Eq. 7] to train a score-based model for topological signals

$$\hat{\theta} = \arg\min \mathbb{E}_{t \sim \mathcal{U}(0,1)}\left\{\lambda(t)\mathbb{E}_{x_0}\mathbb{E}_{x_t|x_0}\left[\|s_t(\hat{\theta}) - \nabla \log p_{t|0}(x_t|x_0)\|^2\right]\right\} \tag{E.7}$$

where $\nabla \log p_{t|0}$ can be readily obtained based on (2).

### E.3 DIFFUSION BRIDGES FOR TOPOLOGICAL SIGNALS

As discussed earlier, SB models are closely related to stochastic interpolants, flow- and score-based models. We further remark that from the $\mathcal{T}$SDE, we can build the *topological diffusion bridge* to directly construct transport models between any topological distributions via *Doob's h-transform* [Särkkä & Solin, 2019]. This has been evidenced in Euclidean domains by converting existing diffusion processes (BM, VE, VP) to diffusion bridges so to arrive at arbitrary distributions, and training upon score matching [Heng et al., 2021; Liu et al., 2022; Delbracio & Milanfar, 2023; Li et al., 2023; Zhou et al., 2024].

Specifically, consider the $\mathcal{T}$SDE. To let it arrive at a final sample $x_1$, the Doob's h-transform gives

$$\mathrm{d}X_t = \left[f_t + g_t^2 \nabla \log p_{1|t}(x_1|X_t)\right] \mathrm{d}t + g_t \, \mathrm{d}W_t, \quad X_1 = x_1, \ x_0 \sim \nu_0 \tag{E.8}$$

where $p_{1|t}(x_1|x_t)$ is the transition kernel of the $\mathcal{T}$SDE satisfying the associated backward FPK, given by Lemma B.6 (cf. Lemma 4). We can further find the time-reversal process for (E.8) [Zhou et al., 2024, Theorem 1]

$$\mathrm{d}X_t = \left[f_t - g_t^2(\nabla \log q_{t|1}(x_t|x_1) - \nabla \log p_{1|t}(x_1|x_t))\right] \mathrm{d}t + g_t \, \mathrm{d}W_t, \quad X_1 = x_1 \tag{E.9}$$

where $q_{t|1}(x_t|x_1)$ is the transition kernel of the new SDE in (E.8) (instead of $\mathcal{T}$SDE) conditioned on $Y_1 = x_1$. The goal is to learn this new score function $\nabla \log q_{t|1}(x_t|x_1)$, which can be achieved by applying the score matching [Song et al., 2020b].

Given paired training samples $(x_0, x_1) \sim q_{01}$, Zhou et al. [2024, Theorem 2] considered a score matching objective to learn the new score function

$$\hat{\theta} = \arg\min \mathbb{E}_{x_t,x_0,x_1,t}\left[\lambda(t)\|\tilde{s}_t(\hat{\theta}) - \nabla \log q_{t|1}(x_t|x_1)\|^2\right]. \tag{E.10}$$

However, we cannot directly find closed-form $q_{t|1}$ in the topological case, we need to use sliced score matching for this case.

Note that we can view that the underlying topological process $X$ now follows the new SDE pair (E.8) and (E.9) as the forward and backward processes, respectively. In this sense, the topological diffusion bridge is a special case of the $\mathcal{T}$SB when setting the policies as $Z_t = g_t \nabla \log p_{1|t}(x_1|x_t)$ and $\hat{Z}_t = g_t \nabla [\log q_{t|1}(x_t|x_1) - \nabla \log p_{1|t}(x_1|x_t)]$. When performing learning on these policies, since $Z_t$ is fixed once $\mathcal{T}$SDE is given, the learning boils down to training the parameterized $\hat{Z}_t^{\hat{\theta}}$.

## F  ADDITIONAL EXPERIMENTS AND DETAILS

We first describe the synthetic experiment on matching Gaussian topological signal distributions based on the closed-form G$\mathcal{T}$SB. Then, we detail the generative modeling experiments conducted on real-world datasets based on $\mathcal{T}$SB-models.

### F.1  CLOSED-FORM G$\mathcal{T}$SB CORROBORATION

*Graph GP matching:* We build a synthetic graph with 30 nodes and 67 edges, as shown in Fig. 1 *(Left)*. From its graph Laplacian $L$, we construct the initial distribution of node signals as a Matérn GP with zero mean and the kernel $\Sigma_0 = (I + L)^{-1.5}$, and the final distribution as a diffusion GP with zero mean and the kernel $\Sigma_1 = \exp(-20L)$. We consider G$\mathcal{T}$SB closed forms $X_t$ in (5) driven by both $\mathcal{T}$SHeat$_{\text{BM}}$ and $\mathcal{T}$SHeat$_{\text{VE}}$. For the former, we set $c = 0.5$ and $g = 0.01$, labeled as GTSB-BM. For the latter, we consider $\sigma_{\min} = 0.01$ and $\sigma_{\max} = 1$ with $c = 0.01$ and $c = 10$, labeled as GTSB-VE1 and GTSB-VE2, respectively.

We then compute the covariances $\Sigma_t$ of the time marginals, which has a closed-form given by Corollary D.1, and obtain the samples based on the closed-form conditional distribution [cf. Corollary D.5] given an initial sample, illustrated in Fig. F.1. We also measure the Bures-Wasserstein (BW) distance [Bures, 1969] of $\Sigma_t$ and $\Sigma_1$ to evaluate the bridge quality, shown in Section 6.

*Edge GP matching:* We also consider matching two edge GPs which are able to model the discretized edge flows in a $\text{SC}_2$ of the vector fields defined on a 2D plane [Chen et al., 2021b]. The initial edge GP has a zero mean and a divergence-free diffusion kernel with $\kappa_C = 10$, while the final edge GP has a zero mean and a curl-free diffusion kernel with $\kappa_G = 10$ [cf. Appendix B.1]. We construct the closed-form G$\mathcal{T}$SB $X_t$ with the $\mathcal{T}$SHeat$_{\text{BM}}$ as the reference dynamics, where $c = 1$ and $g = 0.01$. We obtain the samples from the closed-form SDE representation (7), shown in Fig. F.2. We can see that the forward samples are able to reach the final state, and the backward samples are able to reach the initial state, despite some noise due to numerical simulation.

### F.2  $\mathcal{T}$SB-BASED GENERATIVE MODELING AND MATCHING

#### F.2.1  DATA

*Heat flows:* We use the heatflow dataset from Southeastern Australia from Mather et al. [2018], which collects the heatflow measurements with coordinates in total 294 from 1982 to 2016. Here we split the data into two parts, before and after 2010 (there is a significant change in the heat flow pattern), to understand the evolution of the heat flow by modeling them as initial and terminal data. That is, for this dataset, we consider the *signal matching* task.

*Seismic magnitudes:* We use the seismic event catalogue for M5.5+ (from 1990 to 2018) from IRIS which consists of 12,940 recorded earthquake events with magnitudes greater than 5.5. To process these events, we use the `stripy` toolbox to obtain the *icosahedral* triangulated mesh of the Earth surface [Moresi & Mather, 2019]. Using the refinement of level three, this spherical mesh has 1,922 vertices. We refer to Fig. F.3 for a visualization of such a mesh of level one for better clarity.

Upon this mesh, we first associate each earthquake event to the nearest mesh vertex based on its longitude and latitude. All events are located on 576 unique vertices of the mesh. Using these unique vertices, we then construct a *10-nearest neighbour* graph based on the geodesic distance between the vertices, and we use the symmetric normalized graph Laplacian. Lastly, on top of this graph, we associate the yearly earthquake events to its vertices and take the magnitudes as node

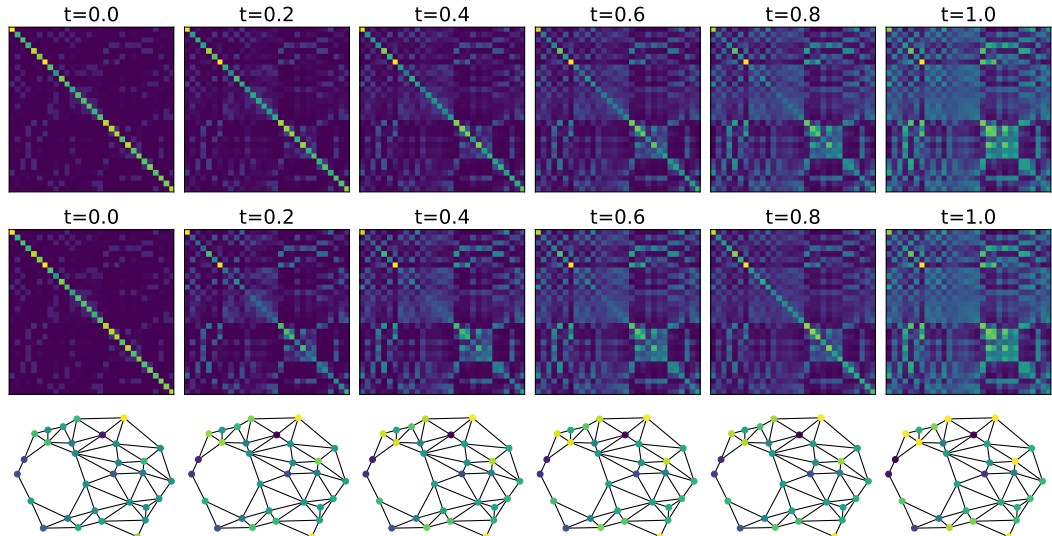

Figure F.1: Covariances of the time marginals of the G$\mathcal{T}$SB driven by $\mathcal{T}$SHeat$_{\text{BM}}$ *(Top)* and $\mathcal{T}$SHeat$_{\text{VE}}$ *(Center)*, as well as the samples conditioned on the inisital signal *(Bottom)*.

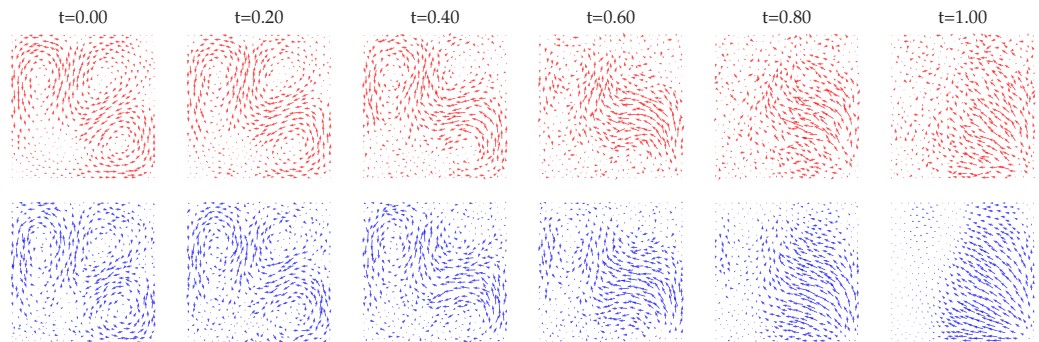

Figure F.2: Forward and backward samples based on the closed-form SDE in (7) with respect to $\mathcal{T}$SHeat$_{\text{BM}}$.

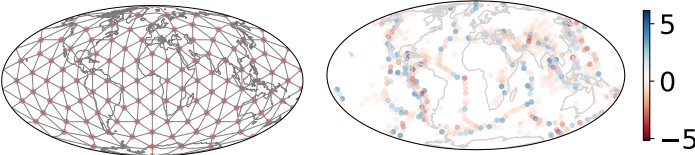

Figure F.3: Earth mesh *(Left)* and a node signal sample of the earthquake magnitudes *(Right)*.

signals, resulting in 29 such signals. Followed by this, we preprocess the magnitudes by removing the mean over the years. For this dataset, we consider the *signal generation* task.

*Traffic flows:* We consider the PeMSD4 dataset which contains traffic flow in California from 01-01-2018 to 28-02-2018 over 307 sensors. We convert the node data into edge flows over a SC$_2$ with 307 nodes, 340 edges and 29 triangles, following Chen et al. [2022b], and use the normalized Hodge Laplacian. For this dataset, we consider the *signal generation* task.

*Ocean currents:* We consider the *Global Lagrangian Drifter Data*, which was collected by NOAA Atlantic Oceanographic and Meteorological Laboratory. The dataset itself is a 3D point cloud after converting the locations of buoys to the *earth-centered, earth-fixed* (ECEF) coordinate system. We follow the procedure in Chen & Meila [2021]; Chen et al. [2021b] to first sample 1,500 buoys furthest from each other, then construct a weighted SC$_2$ as a Vietoris-Rips (VR) complex with around

20k edges and around 90k triangles. For this dataset, we consider the *signal matching* task Upon the weighted Hodge Laplacian, we use edge GP learned by Yang et al. [2024] from the drifter collected data as the initial distribution and synthetize a curl-free edge GP as the final distribution. These two GPs have rather different behaviors, able to model ocean currents with different behaviors and make the matching task challenging. From these GPs, we can generate the samples for training and testing in an efficient way based on eigenpairs associated to the 500 largest eigenvalues, analogous to using Karhunen-Loéve type-decomposition for continuous GPs.

*Brain fMRI signals:* We consider the Human Connectome Project (HCP) [Van Essen et al., 2013] Young Adult dataset where we model the human brain network as a graph and perform the *matching* task on the measured fMRI signals recorded when the subject performed different tasks. We use the HCP recommended brain atlas [Glasser et al., 2016] where each hemisphere is divided into 180 cortical parcels. This results in a total of 360 brain regions. We then build a graph based on the physical conenction patterns between these regions where the edge weights measure the strength of the axonal connections between two regions, i.e., proportional to the inverse of the square distance [Perinelli et al., 2019]. We use the symmetric normalized graph Laplacian. In our experiments, the two sets of the fMRI signals, respectively, correspond to the liberal and aligned brain activities. The former is associated with brain regions involved in high-level cognition, like decision making and memory, whereas the latter is associated with the sensory regions, like visual and auditory, meaning that functional signals are aligned with anatomical brain structure, as shown in Fig. F.4.

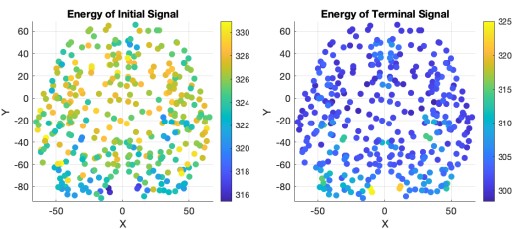
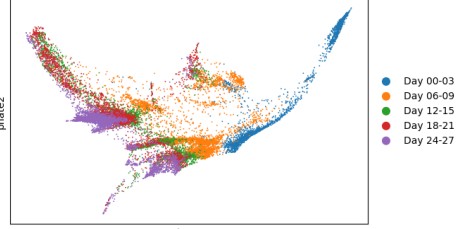

Figure F.4: The energies of the initial (liberal) *(Left)* and final (aligned) brain signals *(Right)*.

Figure F.5: Two-dim `phate` embedding of the single-cell data [Moon et al., 2019].

*Single-cell data:* We consider the single-cell embroid body data from [Moon et al., 2019], which describes the differentiation of human embryonic stem cells grown as embroid bodies into diverse cell lineages over a period of 27 days. These cell data, $X_1, X_2, \ldots, X_5$, are collected at 5 timepoints (day 0–3, day 6–9, day 12–15, day 18–21, day 24–27, indexed by $t \in \{1, 2, 3, 4, 5\}$), resulting in total 18,203 observations. We followed the preprocessing steps provided by `TorchCFM` [Tong et al., 2024a;b]. Please refer to this link for the direct use of preprocessed data [Tong, 2023]. Followed by this, we consider the two-dimensional `phate` embedding for the data [Moon et al., 2019], resulting in the data coordinates of dimension $18,203 \times 2$, as illustrated in Fig. F.5. From the preprocessing, we can build a sparse $k$-nearest neighbouring graph over *the entire set of data observations*. That is, we have an adjacency matrix of dimension $18,203 \times 18,203$.

In our experiment, we aim to transport the observed data from day 0–3 to day 24–27, i.e., from $t = 1$ to $t = 5$. Thus, we build the two boundary distributions based on the normalized indicator functions, which indicate the associated nodes of the data points observed at these two timepoints. That is,

$$\nu_0 := \mathbf{1}_{X_1} / \sum_{j \in X_1} \mathbf{1}_{X_1}(j) \tag{F.1}$$

where the sum is over the nodes associated to the first-timepoint observations in $X_1$, as the initial distribution, and similarly, $\nu_1 := \mathbf{1}_{X_5} / \sum_{j \in X_5} \mathbf{1}_{X_5}(j)$ as the final one. After training the models, using the final sample $\hat{X}_{t=5}$ obtained from the learned $\mathcal{T}\text{SB}$ given the initial observations, we can obtain the predictions at the five timepoints based on the sorting (from large to small) of $\hat{X}_{t=5}$. Specifically, given the indices after sorting, $\text{idx} = \arg\text{sort}(\hat{X}_{t=5})$, we partition them into the disjoint sets, $\text{idx} = \mathcal{S}_1 \cup \mathcal{S}_2 \cup \cdots \cup \mathcal{S}_5$ with $|\mathcal{S}_t| = n_t$ the number of observations at timepoint $t$ for $t = 1, \ldots, 5$. We then have the prediction labels given by $\mathcal{S}_t$ that indicate the nodes supporting the data points predicted at timepoint $t$. The disjointed indices in $\mathcal{S}_t$ essentially provide a labeling of the whole predictions for the five timepoints. We found that using adjacency matrix as the convolution operator in the reference dynamics performs better in practice.

#### F.2.2 MODEL

**Models.** We consider the following two sets of methods:

- Euclidean SB-based models with BM, VE and VP reference processes [Chen et al., 2022a], which we refer to as `SB-BM`, `SB-VE` and `SB-VP`, respectively.
- Topological SB-based model with $\mathcal{T}\text{SHeat}_{\text{BM}}$, $\mathcal{T}\text{SHeat}_{\text{VE}}$ and $\mathcal{T}\text{SHeat}_{\text{VP}}$ as the reference processes, which we refer to as `TSB-BM`, `TSB-VE` and `TSB-VP`, respectively.

For some datasets, we also apply the Gaussian SB SDE solution as the reference dynamics: Euclidean SB-based models with the closed-form GSB SDEs (under BM and VE reference processes) as the reference [Bunne et al., 2023], which we refer to as `GSB-BM` and `GSB-VE`, respectively; and, topological SB-based model with the closed-form G$\mathcal{T}$SB SDEs (7) (under $\mathcal{T}\text{SHeat}_{\text{BM}}$ and $\mathcal{T}\text{SHeat}_{\text{VE}}$) as the reference, which we refer to as `GTSB-BM` and `GTSB-VE`, respectively.

**Improving reference dynamics.** Our proposed three types of reference dynamics have fixed diffusion rates $c$. This may limit the model flexibility in capturing the dynamics of the data, which we found especially in matching ocean current data. Thus, for this task, we allow the time-varying diffusion rate $c_t$. Specifically, we set it to be linearly increasing as $c_t = c_{\min} + t(c_{\max} - c_{\min})$ for some $c_{\min}, c_{\max}$. Moreover, due to the nonlinearity of the underlying process (from a non-curl-free GP to a curl-free GP), we also consider the heterogeneous heat diffusion, as dicussed in (B.18) where the down and up diffusion rates are different.

**Policy models.** For the parameterization of the optimal policies $(Z_t^\theta, \hat{Z}_t^{\hat{\theta}})$, we first obtain the time and signal embeddings individually. To obtain the signal embedding from the input, we consider the following two sets of models as the signal module:

- ResBlock model: one multi-layer perceptron (MLP) followed by a number of residual block modules where each block has three MLPs with sigmoid linear unit (SiLU) activations.
- Topological neural network (TNN) model: For node signals, we consider two-layer GCNs [Kipf & Welling, 2017] followed by one MLP; For edge flows in a $\text{SC}_2$, we consider two-layer SNNs [Roddenberry et al., 2021; Yang et al., 2022a] followed by one MLP, where each SNN layer has the linear convolution $X \leftarrow L_u X W_2 + X W_1 + L_d X W_0$ with the down and up Laplacians $L_d, L_u$ and the learnable weights $W_0, W_1, W_2$.

To obtain the time embedding, we pass the *sinusoidal* positional encoding of the discretized time-point through a two-layer MLP module with SiLU activations. We then sum the two embeddings and pass it through a two-layer MLP output module with SiLU to obtain the final parameterization.

#### F.2.3 IMPLEMENTATION DETAILS

Our implementation is built upon the SB-framework by Chen et al. [2022a]. We use `AdamW` optimizer with a learning rate of $10^{-4}$ and Exponential Moving Average (EMA) with the decay rate of 0.99. For the reference processes with BM involved, we treat the noise scale $g$ as a hyperparameter and optimize it by grid search. For the reference processes with VE and VP involved, we grid search the noise scales $\sigma_{\min}, \sigma_{\max}$ and $\beta_{\min}, \beta_{\max}$. For $\mathcal{T}$SB-based models, we grid search the optimal diffusion rate $c$ and the noise scales involved in the $\mathcal{T}$SHeat.

In computing the likelihood in (9) during training, we use the trace estimator following [Hutchinson, 1989] to compute the divergence. In generative procedures, we apply the *predictor-corrector* sampling [Song et al., 2020b] to improve performance. To evaluate the models, we compute the *negative log-likelihoods* (NLLs) in (E.1) for generation tasks. For both generation and matching tasks, we assess the 1- and (square rooted) 2-*Wasserstein distances* between the predicted and true signals.

#### F.2.4 RESULTS

*Heat flows:* For matching the two types of heat flows, we observe from Table F.1 that: (i) $\mathcal{T}$SB-based models are consistently better than SB-based models; and (ii) using GCNs for policy models increases the performance by a large margin for both sets of models.

*Seismic magnitudes:* In generative modeling for seismic magnitudes, while we have the similar observations as for the previous datasets, from Table F.3, we also observe that the G$\mathcal{T}$SB-based models are able to achieve the best performance, and likewise GSB-based models also increase the

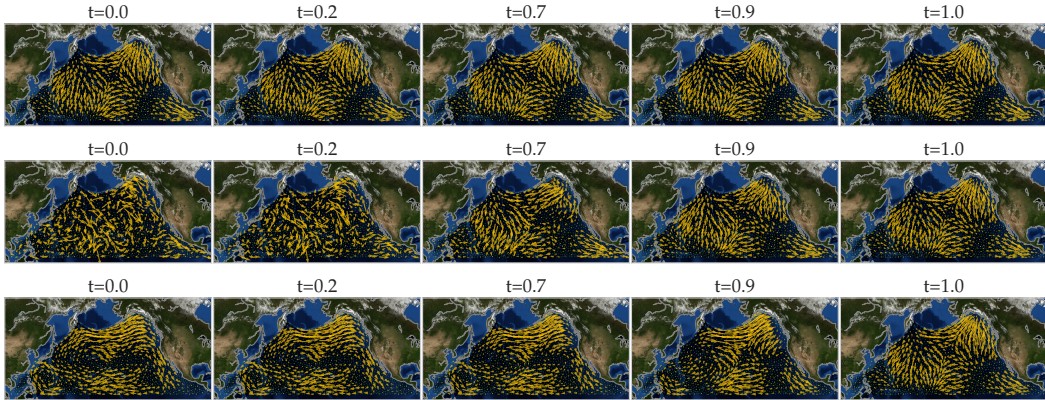

Figure F.6: Backward sampled ocean currents using `TSB-BM` *(Top)*, `SB-BM` *(Center)* and `GTSB-BM` *(Bottom)*.

Table F.1: Heat flow matching results.

| Method | ResBlock | | GCN | |
|---|---|---|---|---|
| | Forward | Backward | Forward | Backward |
| SB-BM | -0.10 ± 0.02 | -0.08 ± 0.03 | -0.74 ± 0.05 | -0.72 ± 0.05 |
| SB-VE | -0.12 ± 0.02 | -0.10 ± 0.01 | -1.20 ± 0.07 | -0.95 ± 0.07 |
| SB-VP | -0.09 ± 0.02 | -0.08 ± 0.02 | -0.83 ± 0.04 | -0.66 ± 0.11 |
| TSB-BM | -0.29 ± 0.02 | -0.27 ± 0.02 | -0.83 ± 0.05 | -0.81 ± 0.05 |
| TSB-VE | -0.31 ± 0.02 | -0.29 ± 0.01 | -1.26 ± 0.05 | -0.97 ± 0.08 |
| TSB-VP | -0.57 ± 0.02 | -0.55 ± 0.02 | -1.01 ± 0.03 | -0.92 ± 0.03 |

Table F.2: Traffic flow results.

| Method | ResBlock | SNN |
|---|---|---|
| SB-BM | 0.82 ± 0.00 | 0.18 ± 0.02 |
| SB-VE | 0.77 ± 0.00 | -0.42 ± 0.01 |
| SB-VP | 0.79 ± 0.00 | -0.09 ± 0.01 |
| TSB-BM | 0.40 ± 0.00 | 0.02 ± 0.03 |
| TSB-VE | 0.01 ± 0.00 | -0.89 ± 0.02 |
| TSB-VP | 0.02 ± 0.00 | -0.32 ± 0.01 |

Table F.3: Seismic magnitudes results.

| Method | ResBlock | GCN |
|---|---|---|
| SB-BM | 2.78 ± 0.01 | 2.71 ± 0.03 |
| SB-VE | 2.97 ± 0.03 | 2.73 ± 0.05 |
| SB-VP | 2.28 ± 0.02 | 2.01 ± 0.03 |
| GSB-BM | 1.86 ± 0.02 | 1.83 ± 0.05 |
| GSB-VE | 1.68 ± 0.03 | 1.46 ± 0.07 |
| TSB-BM | 2.13 ± 0.01 | 1.82 ± 0.02 |
| TSB-VE | 2.22 ± 0.02 | 1.53 ± 0.03 |
| TSB-VP | 2.00 ± 0.02 | 1.51 ± 0.02 |
| GTSB-BM | 1.58 ± 0.01 | 1.43 ± 0.04 |
| GTSB-VE | 1.49 ± 0.02 | 1.06 ± 0.04 |

Table F.4: Ocean current matching results.

| Method | Foward | Backward |
|---|---|---|
| SB-BM | 7.21 ± 0.00 | 7.21 ± 0.00 |
| SB-VE | 7.17 ± 0.02 | 7.17 ± 0.02 |
| GSB-BM | 1.09 ± 0.01 | 0.97 ± 0.00 |
| GSB-VE | 0.83 ± 0.01 | 0.49 ± 0.00 |
| TSB-BM | 6.94 ± 0.01 | 3.70 ± 0.00 |
| TSB-VE | 6.89 ± 0.00 | 3.60 ± 0.00 |
| GTSB-BM | 1.09 ± 0.01 | 0.97 ± 0.00 |
| GTSB-VE | 0.53 ± 0.00 | 0.47 ± 0.00 |

performance of SB-models. In Table F.5, we report the 1- and 2-Wasserstein distances between the generated samples and the true ones.

*Traffic flows:* In geneartive modeling of traffic flows, we observe from Table F.2 that: $\mathcal{T}$SB-based models achieve smaller NLLs and using SNNs for both models improves the performance. This observation is consistent with the Wasserstein metrics reported in Table F.5.

*Ocean currents:* In matching ocean currents with two types of different-behaving edge GPs, *note that the initial sample in the forward process in Fig. 4 and the final sample in the backward process in Fig. F.6 are the true samples*. From these two figures, we first observe that SB-based models fail to learn the dynamics to reach the expected end states, as shown in Figs. 4 and F.6. On the other hand, $\mathcal{T}$SB-based models are able to reach an end state with small curl component in the forward process, yet with some discrepancy from the true one. Moreover, the learned backward dynamics remains noisy and does not completely return to the initial state. This implies that the underlying dynamics cannot be fully captured by the $\mathcal{T}$SHeat-type reference processes. This can be largely alleviated by using G$\mathcal{T}$SB-based models, where both the forward and backward processes reach the expected states with high fidelity. This is however because we have the initial and final distributions

Table F.5: Overall 1- and (square rooted) 2-Wasserstein distances for generating and matching.

| Method | Seismic magnitudes | | Traffic flows | | Brain signals | | Single-cell data | |
|---|---|---|---|---|---|---|---|---|
| | $W_1$ | $W_2$ | $W_1$ | $W_2$ | $W_1$ | $W_2$ | $W_1$ | $W_2$ |
| SB-BM | $11.73_{\pm0.05}$ | $8.29_{\pm0.04}$ | $18.69_{\pm0.02}$ | $13.36_{\pm0.01}$ | $12.08_{\pm0.08}$ | $8.58_{\pm0.05}$ | $0.33_{\pm0.01}$ | $0.40_{\pm0.01}$ |
| SB-VE | $11.49_{\pm0.04}$ | $8.13_{\pm0.03}$ | $19.04_{\pm0.02}$ | $13.61_{\pm0.02}$ | $17.46_{\pm0.14}$ | $12.42_{\pm0.09}$ | $0.33_{\pm0.01}$ | $0.39_{\pm0.01}$ |
| SB-VP | $12.61_{\pm0.06}$ | $8.92_{\pm0.04}$ | $18.22_{\pm0.03}$ | $13.02_{\pm0.02}$ | $13.41_{\pm0.05}$ | $9.54_{\pm0.04}$ | $0.33_{\pm0.01}$ | $0.40_{\pm0.00}$ |
| TSB-BM | $9.01_{\pm0.03}$ | $6.37_{\pm0.03}$ | $10.57_{\pm0.02}$ | $7.62_{\pm0.01}$ | $7.51_{\pm0.08}$ | $5.51_{\pm0.06}$ | $0.14_{\pm0.03}$ | $0.28_{\pm0.05}$ |
| TSB-VE | $7.69_{\pm0.04}$ | $5.44_{\pm0.03}$ | $10.51_{\pm0.02}$ | $7.58_{\pm0.01}$ | $7.59_{\pm0.05}$ | $5.55_{\pm0.04}$ | $0.14_{\pm0.02}$ | $0.27_{\pm0.04}$ |
| TSB-VP | $8.40_{\pm0.04}$ | $5.95_{\pm0.03}$ | $9.92_{\pm0.02}$ | $7.16_{\pm0.01}$ | $7.67_{\pm0.11}$ | $5.64_{\pm0.09}$ | $0.14_{\pm0.01}$ | $0.22_{\pm0.03}$ |

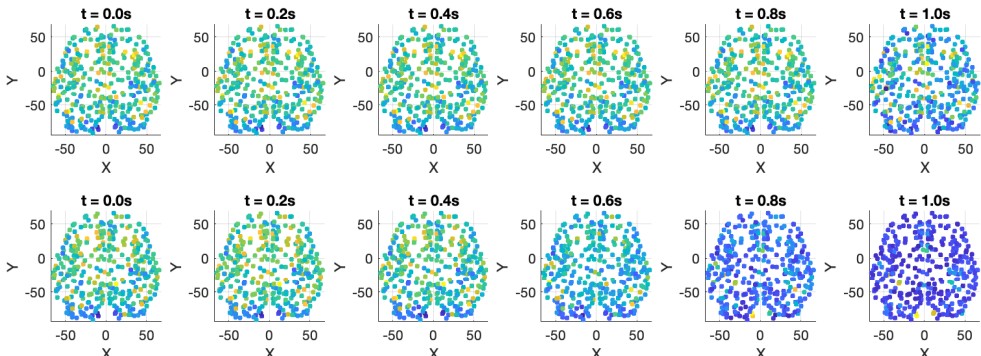

Figure F.7: Intermediate samples of brain signals learned using SB-VE *(Top)* and TSB-VE *(Bottom)*.

modeled by GPs, allowing for G$\mathcal{T}$SB to capture the underlying dynamics better. These observations are reflected in the square-rooted 2-Wasserstein distance results in Table F.4 between the samples given by the learned forward process and the true ones, as well as for the backward ones.

*Brain fMRI signals:* In matching the two brain fMRI signals, we observe from Fig. F.7 that TSB-VE-based model reaches the final state where the signals have lower energy over the brain, indicating aligned activities, whereas SB-VE fails to do so. This is quantatively reflected in terms of the Wasserstein metrics between the generated final samples and the groundtruth ones in Table F.5.

**Ablation study on graph normalizations.** Here, we compare the performance of the TSB-based models using different ways of graph Laplacian normalizations. Specifically, we consider the random walk $L_{\mathrm{RW}}$ and the combinatorial $L_{\mathrm{comb}}$ graph Laplacians. For the latter, we normalize it by dividing the maximal eigenvalue of the Laplacian for stability. From Table F.6, we notice that using the random walk has comparable performance with using the symmetric normalized Laplacian $L_{\mathrm{norm}}^{\mathrm{sym}}$, and using the combinato-

Table F.6: Ablation study results on graph normalizations for brain signal matching.

| Method | TSB-BM | TSB-VE | TSB-VP |
|---|---|---|---|
| $W_1, L_{\mathrm{norm}}^{\mathrm{sym}}$ | $7.51 \pm 0.08$ | $7.59 \pm 0.05$ | $7.67 \pm 0.11$ |
| $W_2, L_{\mathrm{norm}}^{\mathrm{sym}}$ | $5.51 \pm 0.06$ | $5.55 \pm 0.04$ | $5.64 \pm 0.09$ |
| $W_1, L_{\mathrm{RW}}$ | $7.51 \pm 0.08$ | $7.62 \pm 0.09$ | $7.65 \pm 0.09$ |
| $W_2, L_{\mathrm{RW}}$ | $5.52 \pm 0.06$ | $5.58 \pm 0.07$ | $5.62 \pm 0.06$ |
| $W_1, L_{\mathrm{comb}}$ | $8.06 \pm 0.05$ | $9.21 \pm 0.06$ | $9.29 \pm 0.05$ |
| $W_2, L_{\mathrm{comb}}$ | $5.80 \pm 0.04$ | $6.62 \pm 0.05$ | $6.73 \pm 0.03$ |

rial one is worse than the other two. This is not surprising since the combinatorial one does not encode the connection strength between brain regions.

*Single-cell data:* We first measure the Wasserstein distances between the predicted single-cells and the groundtruth ones at the final timepoint, as reported in Table F.5. We here provide the predictions in the two-dim phate embedding space for the SB-BM and TSB-BM models in Fig. F.8 and the latter has a much better prediction. Moreover, from the final sample, we evaluate the predictions at the intermediate timepoints (see Appendix F.2.1) in Table F.7 where the performance of TSB-BM is consistently better than SB-BM. Since our method relies on a graph constructed from the entire data points, we also provide the leave-one-out accuracy for SB-BM by training on the entire data points leaving out the to-be-predicted timepoint. We see that while the accuracy for the final timepoint is perfect, the intermediate predictions remain poor. In contrast, TSB-BM, by making use of the topology, captures the underlying dynamics and predicts the intermediate states better.

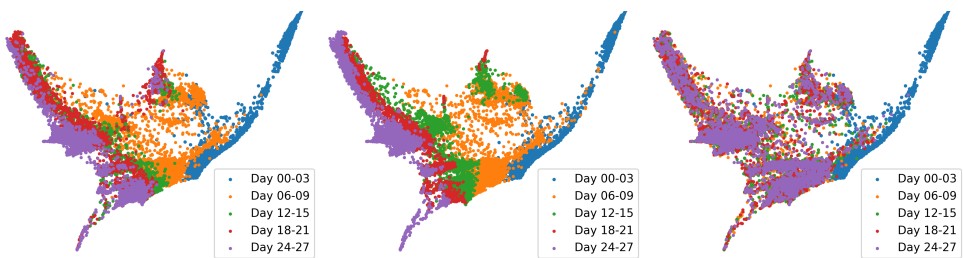

Figure F.8: Single-cell two-dim phate embeddings of the observations *(Left)* and the predictions using `TSB-BM` *(Center)* and `SB-BM` *(Right)*.

Table F.7: Intermediate prediction performance on single-cell data using `TSB-BM` and `SB-BM`.

| Timepoint | TSB-BM | | | SB-BM | | | |
|---|---|---|---|---|---|---|---|
| | $W_1$ | $W_2$ | accuracy | $W_1$ | $W_2$ | accuracy | leave-one-out |
| 2 | $0.03_{\pm 0.00}$ | $0.09_{\pm 0.00}$ | 0.80 | $0.52_{\pm 0.01}$ | $0.59_{\pm 0.01}$ | 0.28 | 0.24 |
| 3 | $0.09_{\pm 0.00}$ | $0.22_{\pm 0.01}$ | 0.42 | $0.12_{\pm 0.00}$ | $0.21_{\pm 0.00}$ | 0.23 | 0.20 |
| 4 | $0.08_{\pm 0.00}$ | $0.16_{\pm 0.01}$ | 0.45 | $0.19_{\pm 0.00}$ | $0.34_{\pm 0.00}$ | 0.26 | 0.26 |
| 5 | $0.14_{\pm 0.03}$ | $0.28_{\pm 0.05}$ | 0.70 | $0.33_{\pm 0.01}$ | $0.40_{\pm 0.01}$ | 0.24 | 1 |

### F.2.5 COMPUTATIONAL COMPLEXITY

Compared to SB-based models, the $\mathcal{T}$SB-based models introduce an additional topological convolution [cf. (1)] overhead, which however admits an efficient computation, as discussed in Section 5. We here provide a quantative comparison of the compelxity in terms of the training time and memory consumption. We measure them using `SB-VE` and `TSB-VE` models on different-sized *10-nearest neighbour* graphs built from Swiss roll point clouds. This comparison is done in a single training stage with 2,000 iterations, running on a single NVIDIA RTX 3080 GPU. As shown in Fig. F.9, we observe that in the moderate scale ($\leq 10,000$) region, the training time and memory consumption of `TSB-VE` are only slightly higher than `SB-VE`, with negligible difference. While this overhead becomes more significant as the scale further increases, both training time and memory can be reduced by exploiting the *sparse* structure (here implemented using `torch.tensor.to_sparse`) in the graph topology such that the computational overheads for for SB and TSB models remain comparable. Under the same settings, Table F.8 compares `SB-BM` and `TSB-BM` models across all datasets. The additional memory and training time introduced by `TSB-BM` remain below 4%.

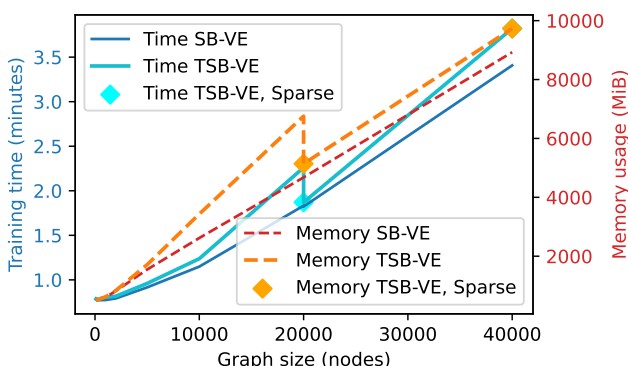

Figure F.9: Training time and memory comparison when training `SB-VE` and `TSB-VE` w.r.t. different-sized graphs.

| Dataset | TSB-BM | SB-BM |
|---|---|---|
| Seismic | 516 | 512 |
| | 50.17 | 51.48 |
| Traffic | 510 | 504 |
| | 52.25 | 50.62 |
| Ocean | 5976 | 5892 |
| | 106.68 | 102.67 |
| Brain | 486 | 468 |
| | 49.62 | 48.97 |
| Single-cell | 4446 | 4294 |
| | 94.30 | 92.54 |

Table F.8: Complexity (first row: memory (in MiB), and second row: training time (in seconds)).

