# OpenReview forum: "Topological Schrödinger Bridge Matching"
_ICLR.cc/2025/Conference — ICLR 2025 Spotlight_

### Official Review · Reviewer_waSt · 2024-10-30

**Soundness:** 3
**Presentation:** 2
**Contribution:** 3
**Rating:** 6
**Confidence:** 2

**Summary:**

The paper introduces the topological Schrödinger bridge problem (TSBP), extending Schrödinger bridge (SB) methods to non-Euclidean domains, specifically graphs and simplicial complexes. Traditional SB approaches face limitations on topological domains. TSBP is proposed to address these by defining a topologically aware reference process based on topological stochastic differential equations (TSDEs). For cases with Gaussian boundary conditions, the authors derive closed-form solutions, then construct TSB-based models for generative modeling and signal matching.

**Strengths:**

- The derivation of closed-form solutions for Gaussian boundaries is rigorous, and the detailed handling of topological SDEs adds significant depth to the method.
- The paper is well-structured, guiding readers through the theoretical framework, derivations, and experimental applications, though some dense mathematical notation may hinder readability.
- By addressing topological signal matching, this work is relevant to fields where data resides on complex networks, such as biological and social networks.

**Weaknesses:**

- Replicability is a major limitation. The lack of code or implementation details limits reproducibility, as replicating the results would require substantial independent effort, particularly in replicating TSDE parameterization and model training. Key training processes (e.g., likelihood optimization for neural networks on topological data) and parameter settings are not transparent, which might impede others from achieving comparable results.
- While synthetic and specific real-world network tasks are covered, the experimental scope could be broadened. Testing across a broader range of real-world network types (e.g., biological, transportation, or communication networks) would better demonstrate the broad applicability of TSBP and strengthen the claims of generalizability. If extending the experimental coverage is impractical, the paper should provide code to enable other researchers to explore the applicability and limitations of TSBP across different types of network-structured data.
- The primary focus on Gaussian boundary distributions restricts practical generalizability, as many real-world networked data scenarios involve non-Gaussian or mixed-distribution boundary conditions.
-  The method’s computational demands are high, particularly in large-scale networks. Although some efficiency measures are discussed, handling matrix exponentials in large or complex network structures remains challenging.

**Questions:**

- Can the authors provide further insight into adapting TSBP for boundary conditions that are not Gaussian?
- Could the authors expand on strategies to manage computational complexity for larger networks, especially in real-world scenarios?

---

> ### Author Response · Authors · 2024-11-19
>
> Dear reviewer, thank you for taking the time to review our work. We appreciate your feedback. Please find our response to your questions:
>
> **on code implementation:** We confirm that the code will be released upon publication.
>
> **on broadening the experimental scope:**
> We included additional experiments on human brain networks and a single-cell dataset for matching tasks. They are conducted on real-world data and make no assumptions of Gaussian boundaries. Please see our [_general response_](https://openreview.net/forum?id=WzCEiBILHu&noteId=7fX8aKiH6b) for the details and results.
>
> We hope they, together with those on real-world seismic, traffic and ocean flow datasets, show the practical generalizability of the method.
>
> **on the Gaussian boundary case**: If we may, we want to clarify the main focus of the paper is in terms of _general_ boundaries. In Section 4, we derived the closed-form for the Gaussian case. This study is however of relevance for both theory and practice, offering insights beyond the Gaussian case. Specifically:
> 1. _understanding topological SB for general cases_. This is particularly manifested by the two-step procedure of in optimizing Gaussian TSBP (i.e., proving Thm 6): first, reducing the original problem to a static problem, then building the optimal process based on that the optimal solution shares the bridge with the reference process.
>
>    Importantly, this procedure applies to general non-Gaussian boundary cases, offering a blueprint for solving general cases by first optimizing the joint distributions between boundaries, and then constructing a bridge based on the reference dynamics. This principle underlines IPF based training methods for SB models, corresponding to the alternating forward and backward training.
>
> 2. _explaining and inspire methods_. The Gaussian TSB time-marginal in eq. 5 aligns with the _stochastic interpolant (SI)_ framework by Albergo et al. (2023), which serves as an alternative approach to generative modeling. So we can really view this framework as a bridge model generalized from the closed-form Gaussian SB, potentially enabling the development of SI models for topological signals..
> 3. _advancing theoretical results for SBPs_. The latest work on the Gaussian SB solution is Bunne et al. 2023 which considered the limited case where the reference process has _scalar_ drift coefficients. Our results in fact showed the closed-form SB solution now can be extended for a broader class of reference processes. For even more general reference processes, this is still an open question.
>
> **on insights to adopting TSB for non-Gaussian boundaries**: Thank you for this question.
> - Our first point above in fact showed how solving TSBP for the non-Gaussian case can be approached.
> - On the other hand, to further see the role of topology, we want to refer reviewer to [this response](https://openreview.net/forum?id=WzCEiBILHu&noteId=qxx5z0bRXh) where we explain why topological approach is needed. We hope these two points provide further clarity and insights.
>
> **on computational complexity:** Two types of computational costs are discussed in this work.
> 1. _Learning TSB vs. SB Models_: The computational demands for learning the two models are comparable. In general, we do not need to handle matrix exponentials in learning TSB models.
> Our analysis (please refer to our [_general response_](https://openreview.net/forum?id=WzCEiBILHu&noteId=7fX8aKiH6b) for the details) compared the computational costs of two models when the graph size is up to 40k. _We included plots for this comparison in [appendix F.2.5](https://openreview.net/pdf?id=WzCEiBILHu#page=42) on p42. Moreover, across all datasets, the _additional memory_ and the _increase in training time_ introduced by TSB remain ```below 4%```, shown in table F.8._
>
> 2. _closed-form Gaussian TSB solution_: For this, we need to handle operators involved with matrix exponentials.
> In our practice for the large dimension of 20k, we applied fast solver (LOBPCG, available in pytorch, scipy.sparse, and large-scale learning libraries like megaman) to pre-compute the eigenbasis of the Laplacian. Note that this is a __one-time-only__ computation, taking 868.07 secs on an NVIDIA rtx 3080 GPU using ```torch.linalg.eig``` (without using sparse computations).
> In moderate-sized cases, we can apply scalable Padé approximations (e.g., scipy.linalg.expm ) for this, requiring only 5.7 secs for dimension 5k, or Chebyshev approximation (please see this [anonymized tutorial](https://anonymous.4open.science/r/topological_SB_matching-387D/GTSB_ClosedForm/readme.md)), as suggested by reviewer 1.
>
> To improve the presentation, we plan to add a notation table to mitigate the dense symbols. We made updates accordingly in the revision. Please refer to the [revision summary](https://openreview.net/forum?id=WzCEiBILHu&noteId=o7PEfrdNne).
>
> Thank you again for your thoughtful feedback. We hope our responses addressed your concerns.

---

> > ### Comment · Reviewer_waSt · 2024-11-27
> >
> > Thank you very much for addressing my concerns. After reviewing the authors' comments, I have decided to raise my score.

---

> ### Author Response · Authors · 2024-11-27
>
> Dear reviewer,
>
> Thank you for appreciating our rebuttal and the updated rating. Your suggestions and questions have helped our work. We appreciate them very much, as well as your time and effort in reviewing our work. In case you have any further questions or feedback, please feel free to leave a comment.

---

### Official Review · Reviewer_fveF · 2024-11-02

**Soundness:** 4
**Presentation:** 4
**Contribution:** 3
**Rating:** 8
**Confidence:** 3

**Summary:**

This work aims to generalize diffusion Schrodinger Bridge generative models from Euclidean domains to general topological spaces, in order to perform topological signal matching and generation. The new method is called topological Schrodinger Bridge Matching (TSBM). The authors develop a topological version of the Schrodinger bridge using a stochastic differential equation driven by a topological convolution. In the case of graphs, this can be driven by the graph heat equation with graph Laplacian. In the case where both distributions to be matched are Gaussian distributions, they give closed form solutions that generalize previous work on Schrodinger bridges. Using this framework, the authors developed a TSB-based method for topological signal generation and matching, based on parameterizing the policy with a topological neural network. The authors also provided the results of some numerical experiments on ocean current matching and modeling seismic events, along with some synthetic experiments.

A couple of typos I noticed: " a diffusion ndoe GP", "Vairous topological", "the spacial case", "we showed that the optimal process satisfy a pair"

**Strengths:**

Deep learning on topological spaces, like graphs, is an important area of research. The paper provides an original and mathematically well-grounded method for topological signal matching and generation. The mathematics is very well-developed and clearly laid out, and the paper is very well-written.

**Weaknesses:**

The numerical experiments are rather limited. I would have liked to see more large scale numerical experiments to validate the method. It is not clear to me if the methods scale up to large data sets well. Some of the future work problems (i.e., "On model training") indicate that it might not.

**Questions:**

On a graph, there are various normalizations of the graph Laplacian. The unnormalized (combinatorial), random walk and symmetric normalized are the three most common, but there are a wide range of others depending on a continuous family of parameters. Could the authors please clarify which graph Laplacian they are using? There is a substantial difference in terms of the diffusion process generated on the graph if one uses the unnormalized versus the random walk Laplacian (with the latter being the natural choice for diffusion on graphs). I think the paper would benefit from a discussion about the type of graph Laplacian that is used, and possibly numerical experiments with different graph Laplacian normalizations.

---

> ### Author Response · Authors · 2024-11-19
>
> Dear reviewer, we appreciate your effort in reviewing our work and thank you for the feedback. Please find our response to your concerns:
>
> **On experiments**: We added new applications on human brain networks and single-cell dataset for matching tasks. We hope, together with the previous ones, they are sufficient to show the generalizability of the work.
> Please see our [_general response_](https://openreview.net/forum?id=WzCEiBILHu&noteId=7fX8aKiH6b) for the details and results, and [_revision summary_](https://openreview.net/forum?id=WzCEiBILHu&noteId=o7PEfrdNne) for the according updates in the revision. Or, more conveniently, please refer to this [anonymized repo](https://anonymous.4open.science/r/topological_SB_matching-387D/TSBLearning/readme.md) where we put the new experiments together, and tabular results related to the following two points.
>
> **On scalability of the methods:**
> Please allow us to clarify that the ocean current is performed over the space of around ```20k``` edges in a simplicial 2-complex. Similarly, the newly added single-cell data is performed over a graph with around ```18k``` nodes. In both large-scale networks, our method scales well.
> Our experimental analyses showed, compared to the SB-models, the _additional memory_ and the _increase in training time_ introduced by TSB remain ```below 4%``` for these two large-scale experiments.
>
> We also reported the training time and memory with respect to the size of the graph up to 40k in the [general response](https://openreview.net/forum?id=WzCEiBILHu&noteId=7fX8aKiH6b). In [appendix F.2.5](https://openreview.net/pdf?id=WzCEiBILHu#page=42) on p42, we now included an illustrative comparison plot, as well as the comparisons of two models across all datasets.
>
> - Analytically, as the TSB-based learning is built upon the SB-based framework, the additional computational cost is only due to the extra drift term in the SDE and it exhibits the same complexity as the SB-based methods for the rest.
> Let us consider a graph with $n$ nodes and $m$ (often $m\ll n^2$) edges:
> if an SB-BM model has a computational cost $O(F(nd))$ in one iteration for some complexity factor $F(nd)$ with feature dimension $d$, then a corresponding TSB-BM model, due to the matrix multiplication $LX$, has a cost of $O(md + F(nd))$ if using the sparse computation; and, a cost of $O(n^2d+F(nd))$, if not. Modern GPUs really allow us to perform such large-scale matrix multiplications very much like allowing us to train deep neural networks.
> - In future work on model training, we aimed to be more inclusive on related work, where we cited recent advances (Deng 2024) that designed (partially) simulation-free training for SB models with a Gaussian final distribution. We wanted to point out this (among others) potential directions on training TSB models. While the other two references (Peluchetti 2023; Shi et al. 2023) used an alternative training method, they necessarily have comparable computational costs.
>
> **On normalizations of the graph Laplacian:**
> - We apologize for the unclarity here and have updated the normalization details in the appendix F.2.1. Specifically, for the synthetic and seismic experiments, we used the symmetric normalization, as well as for the newly added brain signal matching task. For the traffic flow and ocean current matching, we considered the normalized 1-Hodge Laplacian given by _Schaub et al. 2020_ for the former and a weighted Hodge Laplacian following _Chen et al. 2021b (Algorithm 1)_ for the latter.
> - We additionally tested the performance with different graph normalizations on the newly added brain fMRI data. Please see the results below.
>
> | Method     | $W_1, L_{\rm{sym}}^{\rm{normalized}}$     | $W_2, L_{\rm{sym}}^{\rm{normalized}}$    | $W_1, L_{\rm{RW}}$     | $W_2, L_{\rm{RW}}$    | $W_1, L_{\rm{comb}}$     | $W_2, L_{\rm{comb}}$    |
> |:----:|:----:|:----:|:----:|:----:| :----:|:----:|
> | TSB-BM     | $7.51 \pm 0.08$ | $5.51 \pm 0.06$ | $7.51 \pm 0.08$ | $5.52 \pm 0.06$ |$8.06 \pm 0.05$|$5.80 \pm 0.04$|
> | TSB-VE     | $7.59 \pm 0.05$ | $5.55 \pm 0.04$ | $7.62 \pm 0.09$ | $5.58 \pm 0.07$ | $9.21 \pm 0.06$|$6.62 \pm 0.05$|
> | TSB-VP     | $7.67 \pm 0.11$ | $5.64 \pm 0.09$ | $7.65 \pm 0.09$ | $5.62 \pm 0.06$ | $9.29 \pm 0.05$|$6.73 \pm 0.03$|
>
> ** Note that for $L_{\rm{comb}}$, we normalized it by the maximal eigenvalue for stability.
>
> We notice that using the symmetric normalized Laplacian has comparable performance with using the random walk one, and using the combinatorial one is worse than the other two. This is now included as an [ablation study on graph normalizations](https://openreview.net/pdf?id=WzCEiBILHu#page=41.69) on p41.
>
> We appreciate your feedback again and hope our responses have addressed your questions on experiments, the scalability of the methods and the graph normalizations.

---

> > ### Comment · Reviewer_fveF · 2024-11-27
> >
> > Thank you for addressing my concerns and answering my questions. I think the paper has been improved and I have revised my score higher.

---

> > > ### Author Response · Authors · 2024-11-27
> > >
> > > Dear reviewer,
> > >
> > > We're glad to hear that. Thank you for the questions and suggestions for improving our work. We appreciate your acknowledgement and updated rating. In case you have further questions, please feel free to leave a comment.

---

### Official Review · Reviewer_oR4y · 2024-11-03

**Soundness:** 3
**Presentation:** 3
**Contribution:** 3
**Rating:** 8
**Confidence:** 2

**Summary:**

This paper proposes to develop a method for solving the Schrödinger Bridge problem when there is an inherent topological structure to the underlying matching space.  Here, topological structure refers to discrete geometric structure, such as graphs and their higher order generalizations.  Theoretical guarantees are given and numerical demonstrations are given on both simulated and real data.

**Strengths:**

Overall, I found this paper to be a strong contribution.  I would have probably given/would have liked to give higher ratings but I am not very confident in some of the technical areas that this paper covers.

The experiments were good. They were carried out on a range of sizes where smaller ones were able to illustrate the impact of the proposed method but also showcase the computational ability of the method on larger networks.

Good overview on the limitations of the work, including future directions of research that can be carried to overcome them.

**Weaknesses:**

I think that some of the experiments could be more illustrative in why a topological approach is needed and what is missing if the underlying topology is ignored.

Also, other details on the computational aspects appear to be missing, such as runtime, complexity, convergence analyses, etc.

**Questions:**

- I am not familiar at all with the work of Deng et al. (2024) cited in in the introduction, but point clouds were cited as a Euclidean space.  Point clouds are in fact quite a challenging and general data structure (as a finite metric space) and are a key setting in which topological procedures are studied (especially in topological data analysis, which is subsequently cited in the introduction asa. basis for topological machine learning.  Wouldn't the approaches proposed in this work also apply to point clouds?
- Would it be possible to provide a performance comparison of the TSB to SB methods on point clouds?
- Would it be possible to have an example experiment that clearly shows what goes really wrong if the underlying topological structure is ignored and only Euclidean approaches are used?
- Check the paper for typos (e.g., "benefitial" on page 8).

---

> ### Author Response · Authors · 2024-11-19
>
> Dear reviewer, thank you for your positive feedback. Please see our response to your questions.
>
> **On why a topological approach is needed, and vs. Euclidean one**:
> We want to explain this from the following perspectives:
> - When a time stochastic process runs over a graph (for example), in addition to the knowledge (e.g., Markov property by Itô processes) over time dimension, a topological approach also utilizes the spatial statistics of the process, e.g., the covariance dependency between pairs of nodes, based on the topological structure.
> - Let us consider the processes driven by $\rm{TSHeat-BM}$ [cf. Example 1 in line 197]  and $\rm{BM}$ (Wiener process). Conceptually, the latter is a diffusion over time, while the former is a diffusion over both time and graph (space). To be more concise:
>   - In their transition densities from $t_1$ to $t_2$, the former has a cross-covariance $K_{t_1t_2}$ [cf. eq. 2] depending on the coupling of time and graph, while the latter has an only-time-dependent cross-covariance as $t_2-t_1$. The former really gives rise to a nonseparable graph-time process. So if the underlying topology is ignored, the process will not be able to capture the spatial dependency between nodes.
>   - Furthermore, we can see that the topological SB gives rise to an E-OT with a transport cost taking into account the topological information stored in $\Psi_t$ and $K_{t_1t_2}$, while classical SB returns an E-OT with a quadratic transport cost without any topological dependency.
>   - In the Gaussian case [eq. 5], the topological bridge really incorporates the topological information via $K_{t_1t_2}$ and $\Psi_t$.
>
> - Experimentally, on the ocean current (Figure 4) data, as well as the newly added brain signals (```Figure F.7``` on [p41](https://openreview.net/pdf?id=WzCEiBILHu#page=41)) and the single-cell one (```Figure. 3``` on [p9](https://openreview.net/pdf?id=WzCEiBILHu#page=9)), we observed that when the topology is ignored, the matching trajectories either get very noisy or remain largely unchanged. This is because the temporal statistics alone are difficult to transport one type of high-dim signals to another. Please refer to our  [_general response_](https://openreview.net/forum?id=WzCEiBILHu&noteId=7fX8aKiH6b) for details and results on the newly added tasks.
>
> **On point clouds tasks:**
> - We apologize for the confusion here. We meant to cite _Deng et al. 2024_ for time series and images. For point clouds, we meant they are often embedded/defined in terms of 2D or 3D coordinates, yet we agree with the reviewer on the challenging and general nature of point clouds tasks.
> - From a broad sense, our experiments are carried out on point clouds: the seismic magnitude supported on mesh points and the newly added fMRI signals supported on brain regions are scalar fields on point clouds, and the current velocity fields supported on ocean buoys are vector fields on point clouds. However, note that our setting considers the underlying network structure (built upon the point clouds) remains fixed, so the points are not moving and we focus on the signals associated to it.
> - For the typical point clouds tasks,
>     1. like object classification, the point clouds belonging to the same class of objects are possibly different, yet recently TDA-based methods have been applied based on the consistent topological features of the point clouds in the same class, which are rather different from our setting.
>     2. like object tracking, where the point clouds are moving over time, we initially thought our current work does not directly support that scenario because we did not want to have a topology tangled with time dimension. However, as suggested by the first reviewer [here](https://openreview.net/forum?id=WzCEiBILHu&noteId=GIKXfhIesn), if we have observations over the time dimension, we can build a big graph which connects all data points over the time span, and transport observations from one timepoint to another based on their indicator functions.
>
>         We did add this more point clouds-like application using the single-cell data. Please refer to this [_general response_](https://openreview.net/forum?id=WzCEiBILHu&noteId=7fX8aKiH6b) for the details.
>
> - So overall, our work can be applied to the point clouds tasks, though sometimes not trivial.
>
> **On computational aspects:**
> - Please see our  [_general response_](https://openreview.net/forum?id=WzCEiBILHu&noteId=7fX8aKiH6b) on this. We now included experimental complexity analyses in [appendix F.2.5 on p42](https://openreview.net/pdf?id=WzCEiBILHu#page=42). The main message is that TSB models have comparable complexity cost compared to SB ones.
>
> We hope our responses addressed your questions by adding additional experiments on point clouds and experimental analysis on complexity. We once again appreciate these suggestions and made updates in the paper accordingly.

---

### Official Review · Reviewer_ywBL · 2024-11-08

**Soundness:** 2
**Presentation:** 4
**Contribution:** 3
**Rating:** 8
**Confidence:** 3

**Summary:**

The manuscript is on the Schrödinger bridge problem for special topologies like graphs. The authors define the reference process akin to the heat diffusion on said topology. For specific boundary distributions, they present closed for solutions.

**Strengths:**

- The introduction and background are really a pleasure to read; the SBP is very well introduced a rigorously defined.
- The manuscript present strong theoretical contributions and a novel method to solve the SBP.

**Weaknesses:**

*Experiments.*
I believe the authors should conduct additional experiments to show the advantages and limitations of their method. There are many toy datasets or graph that could be used in addition to this one, for instance a Swiss roll (potentially rotated in higher dimensions) on which you can generate from a distribution or interpolate between distributions. I also think that reporting additional metrics for evaluation would be beneficial, for instance MMD, Wassertein etc, between predicted and ground truth distributions.

It is difficult to assess the improvement of the method on figure 2 or 3. Quantitative results along the lines of table 1 are much better. Since this data appears to have multiple timepoints, the authors could validate the quality of interpolation by training with holding out one timepoint and predicting it at inference. The same experiment, should be conducted on multiple datasets, e.g. datasets from [3,4].

*On solution complexity.*
It is mentioned that $\Psi_t$ can be computed efficiently, but this requires the eigendecomposition of L, which scales at O(n^3) for an $n \times n$ L. Did the authors try other approximation methods such as Chebyshev (e.g. used in [1]) or Cholesky (e.g. used in [2]) ? These methods scale much better for on a sparse graph. Additionally,, it would be beneficial to add experiments on complexity and training time with respect to the size of the Laplacian.

**Questions:**

Questions and minor comments.

- On line 146, and the definition of the kernel $\Sigma = \exp(-\kappa^2/2L)$ is L at the numerator or denominator ? If it is like a heat kernel, it should be at the numerator.
- On line 225 (and other sections), $\exp(-cLt)$ is a matrix exponential ?
- I may have missed it, but what is $\xi_1$ on line 248 ?
- In equations 3a and 3b, it looks like 3a goes from t=0 to t=1, and 3b goes from t=1 to t=0, but it is not clear from these equations. Do you also need to define the backward Wiener process in terms of the forward one ?

Potential typo:
- Line 133: "concerns"
- Line 141: "high-dim"
- Line 146 "ndoe"
- line 446: "borh"

[1] Huguet, G., Tong, A., Zapatero, M. R., Tape, C. J., Wolf, G., & Krishnaswamy, S. (2023, September). Geodesic sinkhorn for fast and accurate optimal transport on manifolds. In 2023 IEEE 33rd International Workshop on Machine Learning for Signal Processing (MLSP) (pp. 1-6). IEEE.

[2] J. Solomon, F. De Goes, G. Peyre,´ et al., “Convolutional
wasserstein distances: Efficient optimal transportation
on geometric domains,” ACM Transactions on Graphics
(ToG), 2015

[3] Charlotte Bunne, Laetitia Meng-Papaxanthos, Andreas Krause, and Marco Cuturi. Proximal Optimal Transport Modeling of Population Dynamics, February 2022. arXiv:2106.06345 [cs]. Cited
on page

[4] Alexander Tong, Kilian Fatras, Nikolay Malkin, Guillaume Huguet, Yanlei Zhang, Jarrid RectorBrooks, Guy Wolf, and Yoshua Bengio. Improving and generalizing flow-based generative models
with minibatch optimal transport, March 2024. arXiv:2302.00482

---

> ### Author Response · Authors · 2024-11-19
>
> Dear reviewer, thank you for taking the time to review our work. Please find our response to your questions:
>
> **On experiments:**
> - We added an additional dataset on human brain networks. Please refer to our [_general response_](https://openreview.net/forum?id=WzCEiBILHu&noteId=7fX8aKiH6b) for the details and results.
> - We added Swiss Roll point clouds to corroborate the theoretical results. Please refer to this [anonymized tutorial](https://anonymous.4open.science/r/topological_SB_matching-387D/GTSB_ClosedForm/readme.md) for the implementation.
> We initially did not use this because it was commonly used to transport discrete probability measures from one point clouds to another. This is different from our problem setting where we have a fixed topology and can be viewed as a limitation of our framework, as discussed in lines 518-526. For the same reason, our method cannot be directly applied to the tasks in [3,4] where the data points, together with the possible underlying topology, are moving over time. A possible (perhaps debatable) topology-based method might need a time-varying topology structure. While it is a promising future direction, the extension of topology-based SB models to time-varying topology can be nontrivial.
> - For figures 2 & 3, we forgot to cross-refer to the Wasserstein metrics in appendix Table F.4, which we have fixed. For this experiment, we only have initial and final samples.
> - We added Wasserstein distances for the seismic and traffic experiments in the appendix Table F.5. For convenience, we show them here:
>
> | Method  | Seismic $W_1$ | Seismic $W_2$ | Traffic $W_1$ | Traffic $W_2$ |
> |---------|---------------|---------------|---------------|---------------|
> | SB-BM   | 11.73 $\pm$ 0.05  | 8.29 $\pm$ 0.04   | 18.69 $\pm$ 0.02  | 13.36 $\pm$ 0.01  |
> | SB-VE   | 11.49 $\pm$ 0.04  | 8.13 $\pm$ 0.03   | 19.04 $\pm$ 0.02  | 13.61 $\pm$ 0.02  |
> | SB-VP   | 12.61 $\pm$ 0.06  | 8.92 $\pm$ 0.04   | 18.22 $\pm$ 0.03  | 13.02 $\pm$ 0.02  |
> | TSB-BM  | 9.01 $\pm$ 0.03   | 6.37 $\pm$ 0.03   | 10.57 $\pm$ 0.02  | 7.62 $\pm$ 0.01   |
> | TSB-VE  | 7.69 $\pm$ 0.04   | 5.44 $\pm$ 0.03   | 10.51 $\pm$ 0.02  | 7.58 $\pm$ 0.01   |
> | TSB-VP  | 8.40 $\pm$ 0.04   | 5.95 $\pm$ 0.03   | 9.92 $\pm$ 0.02   | 7.16 $\pm$ 0.01   |
>
>
> **On solution complexity:**
> - We did use approximation for this. In our practice, we have applied a fast eigensolver _LOBPCG_ for the 20k dimension computation, which is implemented in scipy.sparse, pytorch and other large-scale manifold learning libraries. We did this because this computation is required _only once_, and all other operators involved in Thms 6 & 7 can be computed efficiently when using certain amount of principal eigenbasis. In the synthetic experiments, we used scipy.expm, which is based on a Padé approximation.
> - We also included now the _Chebyshev polynomial approximation_ based on _Huguet et al. IEEE MLSP 2023_ and _Marcotte et al. ICASSP 2022_. We appreciate your suggestion and reckon this is feasible thanks to the closed-form Chebyshev coefficients for matrix exponentials (eq. 5 in Marcottee et al. 2022), despite the need for all $t\in[0,1]$.
> Please see the [anonymized tutorial](https://anonymous.4open.science/r/topological_SB_matching-387D/GTSB_ClosedForm/readme.md) for this implementation, where we also provided a running time comparison between the two ways of approximations.
> - We added experiments on complexity and training time on different sized swiss roll point clouds. Please refer to our [_general response_](https://openreview.net/forum?id=WzCEiBILHu&noteId=7fX8aKiH6b)  for the details.
>
> **On minor questions:**
>  1. It is at the numerator. To avoid confusion, we updated it more clearly as $\exp(\frac{-\kappa^2}{2}L)$.
>  2. Yes, it is a matrix exponential.
>  3. It is $\xi_{t=1}$ by taking time $t=1$, and $\xi_t$ is defined in eq. (cond. mean) at line 119.
>  4. Yes, we need the backward one. We updated the text to make it more clear: 'eq. (3a) runs forward and eq.(3b) runs backward in time with the backward Wiener process'.
>
> We appreciate your feedback on our work and hope that our responses addressed your concerns.
>
> ref: _Marcotte, et al. "Fast multiscale diffusion on graphs." ICASSP 2022._

---

> > ### Comment · Reviewer_ywBL · 2024-11-20
> >
> > I would like to thank the authors for the response and providing additional information and experiments.
> >
> > *On experiments akin to [3,4].* I understand that having a topology that evolves with time is not the focus of the manuscript. However, I wonder why it is not possible to create a topology that includes all timepoints? Say you have $k$ timepoints and sets of observations ${X_1, \dotsc, X_k}$, you can define a graph using all the data $\mathcal{X}:=\cup X_i$ (e.g. with a knn graph). Considering the transport of $X_1$ to $X_2$ can be done with a normalized indicator function on $\mathcal{X}$, i.e. one if the node is from $X_i$ zero otherwise
> > $\mu_i = \mathbb{1}\_{X_i}  /  \sum_k \mathbb{1}\_{X_i}(v_k)$. Then the SB problem is between $\mu_1$ and $\mu_2$ on $\mathcal{X}$.
> >
> > I appreciate the anonymized tutorial and new experiment on solution complexity.

---

> > > ### Author Response · Authors · 2024-11-21
> > >
> > > While we hope your other questions have been addressed, we want to thank you for the enlightening ideas. We didn't think like that before and had the rigid thought of needing a graph topology which does not entangle with time-dimension.
> > >
> > > We now took your suggestions and added this application on single-cell data. We took the dataset from [5] and looked at the example tutorial on handling single-cell data [TorchCFM: a Conditional Flow Marching library](https://github.com/atong01/conditional-flow-matching) as well as the related papers. Thanks to the preprocessing by [5] and the tutorial (we have to admit that we hadn't handled such data before), we could directly use the built connectivity matrix (graph). We then built the initial and final distributions based on your suggestion. We did get very interesting results based on the final sample of the learned forward transport. Please take a look at them in the [anonymized repo](https://anonymous.4open.science/r/topological_SB_matching-387D/TSBLearning/readme.md#single-cell-data-newly-added).
> > > The run time for each 2000 iterations for this data takes around 50 secs, allowing us to produce the results timely.
> > >
> > >
> > >
> > > [5] Tong, Alexander (2023), “Processed Single-cell RNA Time Series DAta”, Mendeley Data, V1, doi: 10.17632/hhny5ff7yj.1

---

> ### Comment · Reviewer_ywBL · 2024-11-21
>
> Thank you, I appreciate your willingness to try these new experiments and provide results so fast. For the interpolation evaluation, when you write “despite that we did not incorporate the intermediate observations in the training”, does the graph still includes all the data ? For example, if you are holding out timepoint `2`, is your graph constructed from timepoints 1,3,4,5 ? I think both can be interesting, but it should be clear in the details.

---

> > ### Author Response · Authors · 2024-11-21
> >
> > We appreciate your suggestion and it helped improve our work.
> >
> > Indeed, the graph includes all the data, so we should make this clear.. This does feel like we are using oracle knowledge.
> >
> > Here, at the first glimpse, there could be some difficulty without using the whole graph: if we do not have possible nodes indicating where the held-out observations can be, then our predicted final samples can never have values on these nodes. In the example, if we have timepoints except 2, then those nodes appeared at time 2 are likely not in the domain, so it's hard to produce some predictions. Please correct us if we were wrong. Hope it's not confusing.

---

> > > ### Comment · Reviewer_ywBL · 2024-11-21
> > >
> > > Thank you for the explanation. It would be helpful to explicitly clarify that the method relies on a graph constructed from all data points. Additionally, you might consider reporting the prediction accuracy of SB-BM trained on the entire dataset, alongside the current scores. Alternatively, if you are holding out timepoint 2, you could augment your dataset to construct the graph—for instance, by incorporating interpolations between timepoints 1 and 3. Essentially, generating candidates that your model will assign weights to. This is just a suggestion and may be outside the scope of your manuscript.

---

> ### Author Response · Authors · 2024-11-21
>
> Yes, we made this clear in the new revision, in both main text (lines 427-428) and appendix on data details. Meanwhile, we added the accuracy when training on the entire dataset (Table F.7).  A quick observation is that the final prediction accuracy can even reach 100%. However, the intermediate predications (interpolations) remains poor (still using indicators). Thank you also for the other suggestion. That is indeed a way to go when not using the entire dataset in this case. Yet, we agree that can be nontrivial and goes beyond this work.
>
> Overall, we hope we fully addressed your concerns on this work. It was quite pleasant to get your suggestions and work on applications outside our comfort zone. This greatly improved our work.

---

> > ### Comment · Reviewer_ywBL · 2024-11-21
> >
> > Thank you for addressing my questions and participating in the discussion. I don’t have other questions at the moment.

---

> > > ### Author Response · Authors · 2024-11-25
> > >
> > > Dear reviewer,
> > >
> > > Thank you very much for the engaging discussion, helpful suggestions on our work, and the updated rating! Please feel free to leave a comment in case you have any further questions or feedback until the end of the discussion period.

---

### Author Response · Authors · 2024-11-19
**General response**

We thank all four reviewers for their helpful comments and suggestions on our work! Below we address concerns shared by multiple reviewers.

### __1. on additional experiments__

We considered applications of our work in the following additional datasets to broaden the scope of experiments:
1. a dataset for fMRI signals matching on brain networks where we use two sets of fMRI signals representing different brain activities.
2. a single-cell dataset where we transport the initial observation to the observation at the final timepoint.

We refer reviewers to this [anonymized repo](https://anonymous.4open.science/r/topological_SB_matching-387D/TSBLearning/readme.md) for the details of experiment, together with the visualizations (and code pre-release). They are accordingly updated in the paper (please see below the [summary of revisions](https://openreview.net/forum?id=WzCEiBILHu&noteId=o7PEfrdNne)). For convenience, we provide the Wasserstein distance results here, which are consistent with our previous observations.

| Method  | $W_1$, single-cell  | $W_2$, single-cell | $W_1$, brain signals | $W_2$, brain signals    |
|---------|------------------|------------------|---------------------|---------------------|
| SB-BM   | 0.33 ± 0.01      | 0.40 ± 0.01      | 12.08 ± 0.08        | 8.58 ± 0.05         |
| SB-VE   | 0.33 ± 0.01      | 0.39 ± 0.01      | 17.46 ± 0.14        | 12.42 ± 0.09        |
| SB-VP   | 0.33 ± 0.01      | 0.40 ± 0.00      | 13.41 ± 0.05        | 9.54 ± 0.04         |
| TSB-BM  | 0.14 ± 0.03      | 0.28 ± 0.05      | 7.51 ± 0.08         | 5.51 ± 0.06         |
| TSB-VE  | 0.14 ± 0.02      | 0.27 ± 0.04      | 7.59 ± 0.05         | 5.55 ± 0.04         |
| TSB-VP  | 0.14 ± 0.01      | 0.22 ± 0.03      | 7.67 ± 0.11         | 5.64 ± 0.09         |


### __2. on complexity__
There are two kinds of complexity costs discussed in this work. Please see our analysis of each below.
1. TSB-based model learning complexity: Learning TSB models is comparable to learning SB models. Compared to the latter, TSB models only have an additional cost due to the extra drift term of topological convolution $H_t X_t$, in the TSDE. As discussed in lines 383-387, p8, this is a recurrent operation of $LX_t$, admitting an efficient computation due to the typically sparse structure of $L$. Modern GPUs excel at parallel computation, allowing us to perform large-scale matrix multiplications. Here, we show, on different-sized Swiss Roll graphs, the training time and memory of TSB models versus SB models. The comparison is done in a single training stage with 2000 iterations and runs on a single NVIDIA RTX 3080 GPU. A straightforward plot comparison is provided now in [appendix F.2.5](https://openreview.net/pdf?id=WzCEiBILHu#page=42).

|Graph Size|Time SB-VE (s)|Time TSB-VE (s)|Time Increase|Memory SB-VE (MiB)|Memory TSB-VE (MiB)|Memory Increase|
|:-:|:-:|:-:|:-:|:-:|:-:|:-:|
|100|46.79|47.21|0.90%|504|504|0%|
|200|46.28|46.68|0.86%|518|518|0%|
|1,000|46.44|48.31|4.03%|606|618|1.98%|
|2,000|47.51|49.00|3.14%|818|836|2.20%|
|10,000|68.93|74.27|7.75%|2618|3432|23.72%|
|20,000|109.96|135.63|15.30%|4676|6756|44.48%|
|20,000,sparse|109.38|112.23|2.61%|4678|5140|9.88%|
|40,000,sparse|204.42|229.29|12.17%|8928|9734|9.03%|

   - Here, _sparse_ denotes the implementation of the graph Laplacian using ```torch.tensor.to_sparse```. We see that: 1) in the moderately large scale ($< 10,000$) region, the training time and memory of TSB models are _negligibly_ higher than SB models; 2) while this difference becomes larger as dimension further increases, both the training time and memory are significantly reduced by exploiting the sparsity in the graph topology and close to those of SB models.

   - We also compared the two models across all the datasets in table F.8 on p42: the _additional memory_ and the _increase in training time_ introduced by TSB remain ```below 4%```.

2. Closed-form GTSB solution complexity: This concerns the computations of eqs. 5-6, which involve several matrix operations. We recognize they can be converted as scalar operations in the eigenspectrum. For this, we needed to do eigen-decomposition. However, note that this is a _one-time-only_ pre-computation. In our practice for dimension of 20k, we applied a fast eigensolver, LOBPCG, which is implemented in pytorch, scipy.sparse and other large-scale learning libraries. We again measured the wall clock time for the full eigenbasis computation: it takes 868.08 secs on one NVIDIA rtx 3080 GPU using ```torch.linalg.eig``` without using sparse computations. For smaller-scaled cases, we can apply the scalable Pade approximations for matrix exponentials. Suggested by the first reviewer, we also implemented Chebyshev polynomials based approximation.

---

> ### Author Response · Authors · 2024-11-21
> **Summary of revisions**
>
> **Summary of revisions in the paper:**
>
> _main text_:
> 1. We added introductions of the newly added two datasets (lines 420-431 on p8), and the predicted results visualized in ```Figs. 2 & 3``` in the ```experiments``` section.
> 2. We removed one ocean current figure, because it was already included in the appendix, and kept Fig. 4.
> 3. We replaced the NLL results (were already in the appendix on p40) with the 1-Wasserstein metrics across all datasets in ```Table 1```.
>
> _appendix_:
>
> 1. For the newly added brain and single-cell datasets, we included their details in appendix F.2.1 on p38, and the full experimental results in Table F.5 on p41, as well as visualizations in Figs. F. 7 & 8.
> 2. For the computational aspects of TSB models compared to SB models, we added a subsection appendix F.2.5 on p42 which includes the complexity comparisons for different-sized graphs (Fig. F.9) and across different datasets (Table F.8).
>
> We fixed a few typos in the paper and did a few small changes as suggested by reviewers to improve the presentation.

---

### Meta-Review · Area_Chair_VZ2h · 2024-12-20

**Metareview:**

The authors consider the Schrödinger bridge (SB) problem for topological domains such as graphs, simplicial complexes. The authors leverages topology-aware stochastic dynamics, e.g., topological heat diffusion to set the reference process. For Gaussian boundary distributions, the authors derive closed-form topological SB. The authors use topological neural network to parameterize the forward-backward topological dynamics for general cases. The authors show advantages of the proposed method on several experiments. The Reviewers agree that it is a good submission for ICLR'2025. We urge the authors to incorporate the Reviewers' comments and discussion in the rebuttal in the update version, especially about time complexity of the proposed method, and code release for reproduce as mentioned in the Authors' rebuttal.

**Additional Comments On Reviewer Discussion:**

The authors clarify the time complexity during the rebuttal, especially on how to handle large graph, e.g., on eigen decomposition, matrix multiplication.

---

### Decision · Program_Chairs · 2025-01-22

Accept (Spotlight)